# Evaluation of record low ozone values over the Arctic in boreal spring 2020

Martin Dameris[1], Diego G. Loyola[2], Matthias Nützel[1], Melanie Coldewey-Egbers[2], Christophe Lerot[3], Fabian Romahn[2], Michel van Roozendael[3]

[1]Deutsches Zentrum für Luft- und Raumfahrt, Institut für Physik der Atmosphäre, Oberpfaffenhofen, Weßling, Germany.
[2]Deutsches Zentrum für Luft- und Raumfahrt, Institut für Methodik der Fernerkundung, Oberpfaffenhofen, Weßling, Germany.
[3]Royal Belgian Institute for Space Aeronomy, Uccle, Belgium.

*Correspondence to*: Martin Dameris (martin.dameris@dlr.de)

**Abstract.** Ozone data derived from the TROPOMI sensor onboard the Sentinel-5 Precursor satellite show an exceptional ozone hole-like feature in the polar region of the Northern hemisphere (Arctic) in spring 2020. Minimum total ozone column values around or below 220 Dobson units (DU) were seen over the Arctic for 5 weeks in March and early April 2020. Usually the persistence of such low total ozone column values has only been observed in the polar Southern hemisphere (Antarctic) in spring, but not over the Arctic. The ozone hole-like pattern was caused by a particularly stable polar vortex in the stratosphere with a persistent cold stratosphere at higher latitudes, a prerequisite for ozone depletion through heterogeneous chemistry. Based on the ERA5 reanalysis from ECMWF, the Northern winter 2019/2020 (from December to March) showed minimum polar cap temperatures consistently below 195 K around 20 km altitude, which enabled enhanced formation of polar stratospheric clouds. The special situation in spring 2020 is compared and discussed in context with two other Northern hemisphere spring seasons, namely in 1997 and 2011 that were also showing relatively low total ozone column values. However, during these years total ozone columns below 220 DU over several consecutive days were not observed. The similarities and differences of the atmospheric conditions of these three events and possible explanations are presented and discussed. It becomes apparent that the monthly mean of the minimum total ozone column value for March 2020 (221 DU) was clearly below the respective values found in March 1997 (267 DU) and 2011 (252 DU), which emphasizes the noteworthiness of the evolution of the polar stratospheric ozone layer in Northern hemisphere spring 2020. A comparison with a typical ozone hole over the Antarctic (here 2016) indicates that although the Arctic spring 2020 situation is remarkable with total ozone column values around or below 220 DU observed over a larger area (up to 0.9 million km$^2$), the Antarctic ozone hole shows total ozone columns typically below 150 DU over a much larger area (in the order of 20 million km$^2$).

## 1 Introduction

Today's operating satellite instruments produce a reliable picture of the Earth's atmosphere and its chemical composition. These instruments monitor, for example, the evolution of the stratospheric ozone layer (e.g., Loyola et al., 2009), which is important for life on Earth. Unusually low ozone values can occur especially in polar regions if chemical and dynamical processes are interacting in a specific way that allow for strong ozone depletion and hamper meridional transport of ozone rich air from lower latitudes (e.g., Solomon, 1999; Solomon et al., 2014).

The largest concentrations of atmospheric ozone are found in the stratosphere, in the ozone layer, with about 90% of ozone abundance being located at an altitude between 15 and 30 km (e.g., Langematz, 2019). The Dobson unit (DU) – named after Gordon Dobson (1889-1976), who devised the first instrument for measuring atmospheric ozone content – is used to describe the total amount of ozone found in the atmosphere above a specific location. Typically, in the Antarctic an ozone hole is defined as the area where the total ozone column (TOC) decreases to values of less than 220 DU (e.g., WMO, 2018). In the Southern hemisphere polar region (Antarctic) a TOC below 220 DU is about 30% under the climatological mean ozone value in austral spring, which was determined for the years before 1980 (e.g., Chapter 4 in WMO, 1999; Chapter 3 in WMO, 2003). Climatological mean TOCs averaged over the Northern polar region (Arctic) in boreal spring are higher (~400-450 DU; e.g., Dameris, 2010). Therefore, the decrease of TOC below 220 DU during this period indicates a reduction of total ozone in the order of 50%. We note that Northern hemisphere (NH) winters with reduced wave activity are related to reduced transport of ozone into the stratospheric polar vortex and to stronger ozone depletion in the lower stratosphere because of lower temperatures (e.g. Tegtmeier et al., 2008).

In the cold lower polar stratosphere in winter, polar stratospheric clouds (PSCs) are formed during the polar night. PSCs develop at temperatures below 195K or 188K at 50 hPa (see for instance Figure 4-1 of Chapter 4 in WMO, 2018) when nitric acid trihydrate crystals (NAT: $HNO_3 \cdot 3H_2O$; so-called NAT-PSC) or ICE-PSC are formed, respectively. The South polar lower stratosphere cools significantly more in winter (June–August) than the North polar stratosphere (December–February). PSC particles allow heterogeneous reactions to take place on their surfaces, which enable halogen compounds (for instance chlorine) to be released from reservoir compounds (e.g., $ClONO_2$, HCl) and then be converted to an active form. When the sun returns in the polar spring, active molecules, such as $Cl_2$ or HOCl, are converted into reactive Cl and ClO, and ozone depletion begins. More details can be found in the review article by Solomon (1999).

Due to the prohibition of the production and usage of ozone depleting substances (among others CFCs: chlorofluorocarbons) in response to the international activities to protect the ozone layer (Montreal Protocol: multilateral environmental agreement of the United Nations, signed in 1987, and its amendments) atmospheric concentrations of these chemical substances (particularly CFCs) and their products have been reduced over the last 20 years by about 15% (Chapter 1 in WMO, 2018). Nevertheless, the current atmospheric content of CFCs is still enhanced with respect to 1980s values as CFCs have lifetimes of several decades (SPARC, 2013). Consequently, the chlorine concentration in the stratosphere is still high. Based on the

current scientific understanding, the chlorine content is expected to reach pre-CFC-era conditions (i.e. levels similar to the ones before 1980) around the middle of this century, and therefore we can expect a return to pre-CFC values of the ozone layer in the next 30 to 40 years (see Chapters 3 and 4 in WMO, 2018).

Notwithstanding the Montreal Protocol and the projected recovery of the ozone layer, very low temperatures in the polar lower stratosphere in any particular year can lead to heavy ozone depletion in early spring, not only in the Southern hemisphere (SH), but also in the NH. For instance, as shown in Figure 1, in March and early April 2020 very low TOC values were measured in the Arctic although the stratospheric chlorine content in 2020 is known to be clearly lower than in previous years (Chapter 1 in WMO, 2018).

The dynamical conditions of the stratosphere as observed in NH spring 2020 were unusual, showing a stable polar stratospheric vortex with low temperatures as we will see in the upcoming analysis. Two other specific situations have been noted in the literature, indicating comparable dynamical conditions in the Northern stratosphere in spring: 1997 (e.g., Coy et al., 1997; Manney et al., 1997; Lefèvre et al., 1998; Hansen and Chipperfield, 1999) and 2011 (e.g., Manney et al., 2011; Sinnhuber et al., 2011; Kuttippurath et al., 2012; Hommel et al., 2014). However TOC below 220 DU have not been observed in these two years. Although the dynamical conditions this winter and spring 2019/2020 were unusual, atmospheric researchers have expected the possible occurrence of such conditions because they are in the natural range of stratospheric dynamical fluctuations in Northern winter and early spring (e.g., Langematz et al., 2014). The importance of stratospheric dynamics causing low TOC has been discussed in detail in the last decades (e.g., Chapters 4 and 12 in WMO, 1999; Solomon, 1999; Chapter 3 in WMO, 2003; Rex et al., 2004; Tilmes et al., 2006; Kivi et al., 2007; Tegtmeier et al., 2008; Harris et al., 2010; Chapter 3 in WMO, 2014).

Considering the dynamical conditions, it was not unexpected to measure low TOC values within the polar vortex in NH spring 2020. However, as indicated in Figure 1, it is still noteworthy, that the TOC values were below the typical ozone hole threshold of 220 DU for about 5 weeks, despite the reduced chlorine content in the stratosphere. The occurrence of TOC values below 220 DU in March 2020 derived from satellite instrument measurements is confirmed by ground-based measurements at different NH stations, in particular at stations in Canada (e.g., Alert, Eureka, and Resolute). The ozone data are available, for instance at http://www.temis.nl/uvradiation/UVarchive/stations_uv.html (van Geffen et al., 2017). Additional ozone sonde profiles are discussed in detail by Wohltmann et al. (2020).

This study provides a description of the recent dynamical situation in northern winter and spring 2019/2020, which led for the first time to TOC values below 220 DU in larger areas of the polar vortex for an extended time period over the Arctic. It allows an evaluation of the current situation by the comparison with similar dynamical conditions in Arctic spring of other years, but which did not show such low TOC values below 220 DU over the polar NH in spring. Further, we also demonstrate that the observed Arctic low TOC values in spring 2020 are far away from the usually observed Antarctic ozone hole.

In the next Section (Sect. 2) the data sets used for our analyses are introduced including a short description of the performed data processing. In Section 3 the special situation in NH winter and spring 2019/2020 is presented in detail and in Section 4 it is compared with two NH winter and spring periods, namely 1996/1997 and 2010/2011, where similar polar stratospheric conditions – including low TOC values – have been observed. In addition, the observations in Arctic winter and spring 2019/2020 are compared with a typical Antarctic ozone hole as detected in 2016 and with the small Antarctic ozone hole observed in 2019. The discussion of results and the conclusions are presented in Section 5 and Section 6, respectively.

## 2 Data and data processing

Meteorological data

In this study the presented dynamical analyses are based on meteorological data derived from ECMWF's most recent atmospheric reanalysis, ERA5 (Hersbach et al., 2019b; 2020). For our investigations the ERA5 temperature and wind data was downloaded at the provided 0.25°x0.25° resolution. Daily mean data are prepared for the presentations of the respective meteorological situations. They are produced using hourly data on pressure levels (Hersbach et al., 2018) and using the CDO (climate data operators; Schulzweida, 2019) command "daymean" to produce daily means from the hourly data. Monthly mean values are obtained from the monthly mean data at pressure levels (Hersbach et al., 2019a). In addition, hourly potential vorticity (PV) data on isentropes, which are aggregated in the same way to daily data, were obtained from the full ERA5 data set (Copernicus Climate Change Service (C3S), 2017) and regridded to a regular latitude longitude grid (roughly 0.28°x 0.28°). The focus is laid on stratospheric zonal winds, polar temperatures and potential vorticity (PV). ERA5 (raw) data is publicly available. For details see the data availability section.

Ozone data

Ozone data from July 2019 to April 2020 from the TROPOMI sensor onboard the EU/ESA Copernicus Sentinel-5 Precursor satellite are scientifically used for the first time in combination with the long-term ozone data set from the European satellite data record GOME-type Total Ozone Essential Climate Variable (GTO-ECV) from July 1995 to June 2019 (Coldewey-Egbers et al., 2015). The publicly available (Level 2) TOC for July 2019 to April 2020 are derived from the TROPOMI sensor using the GODFIT algorithm (Lerot et al., 2014). The estimated mean bias of the TROPOMI total ozone compared with ground-based measurements is less than ±1 % with a mean standard deviation of up to ±1.6-2.5 % (Garane et al., 2019). An initial comparison of TOCs from TROPOMI and the Ozone Monitoring Instrument (OMI) onboard the NASA Aura spacecraft indicated that TROPOMI TOCs are slightly smaller (~-1%) than OMI TOCs. A similar difference is thus expected w.r.t. GTO-ECV since OMI provides the reference basis for the combined record (see next paragraph). The TROPOMI TOC images presented first in this study are based on daily mean data regridded to 1°x1° resolution to facilitate the comparison with the GTO-ECV data. For details see the data availability section.

GTO-ECV has been developed in the framework of the European Space Agency's Climate Change Initiative ozone project and is based on observations from the satellite sensors GOME/ERS-2, SCIAMACHY/ENVISAT, OMI/Aura, and GOME-2/MetOp covering the time period from July 1995 to June 2019 (Coldewey-Egbers et al., 2015). As for TROPOMI the retrieval algorithm GODFIT (Lerot et al., 2014) is used to derive TOCs from the measurements of the individual satellite sensors. Before the separate data records are merged into one single product, adjustments are applied in order to minimize possible inter-sensor biases and/or drifts. If not accounted for such discrepancies can introduce unwanted discontinuities or artificial trends in the combined record. Due to its notable temporal stability OMI was selected to serve as a reference for the other instruments. They are then adjusted in terms of a correction that depends on latitude and time. The agreement between GTO-ECV and ground-based observations is 0.5%-1.5% peak-to-peak amplitude with a negligible long-term drift in the NH (Garane et al., 2018) and the difference between GTO-ECV and an "adjusted" TOC data set based on reanalysis data is between $-0.5\pm1.7$ % and $-1.0\pm1.1$ % (for details see Coldewey-Egbers et al., 2020). In particular the excellent temporal stability makes the GTO-ECV data record suitable and useful for applications related to long-term investigations of the ozone layer. In this study we use the daily mean data product at 1°x1° resolution to analyse minimum ozone columns in the NH polar region during the past 24 years. During polar night the used satellite sensors cannot provide measurements. For instance, in December, north of about 70°N no observations are available. With returning sunlight the coverage in the northern high latitude regions improves and global coverage is resumed around March 20.

## 3 Situation in Northern winter and spring 2019/2020

In the Arctic winter and early spring 2019/2020 the stratospheric polar vortex turned out to be persistent with strong zonal winds from mid-December until early April. In connection with Figure 1, Figure 2 presents the potential vorticity (PV) field of ERA5 in the NH at the isentropic surface of 475 K (around 20 km altitude), representing the dynamic state of the lower stratosphere with respect to the position and strength of the polar vortex. The region of strong PV-gradients, which is represented here by the contour line of 36 PV units (e.g., Wohltmann et al., 2020), indicates the edge of the polar vortex. The figure illustrates that the polar vortex is still stable in March and early April, and that the position of the polar vortex is in accordance with the region of low TOC values (Figure 1). In Figure 3, our analysis of ERA5 data at 60°N, 10 hPa (about 30 km altitude) shows strong zonal mean zonal wind speeds (magenta line and dots in the figure), which are high with respect to the monthly mean values for the time period from 1979/1980 to 2019/2020 (see grey dots in the figure). This finding is very much in line with a similar analysis by Lawrence et al. (2020), which was using the respective Modern-Era Retrospective analysis for Research and Applications version 2 (MERRA-2) data. In addition, Lawrence et al. (2020) showed that the polar vortex was generally stronger than usual (with respect to the climatological mean) in the polar stratosphere from November to April. Further, their analysis showed the height dependence of the zonal mean zonal wind anomalies and it was found that the anomalies in NH spring are most pronounced around 10 hPa (and above). Smaller dynamical fluctuations were detected in winter 2019/2020, which were caused by planetary wave activity (not discussed in detail here; see for instance the variations

of meridional heat and momentum fluxes at mid-latitudes, 100 hPa, which can be found at https://acd-ext.gsfc.nasa.gov/Data_services/met/ann_data.html or https://ozonewatch.gsfc.nasa.gov/; see Newman et al. (2001) for details on the relation of these quantities to the wave activity). No relevant warmings of the polar stratosphere were observed (see below) and the polar vortex was mostly undisturbed and showed a circular shape, except for the period from mid-January to beginning of February 2020, as hinted in Figure 3. The results are further supported by Lawrence et al. (2020). In Figure 4 the ERA5 monthly mean zonal winds derived for the NH in January, February and March in 2020 are clearly indicating a persistent strong polar vortex, with maximum zonal wind speeds at 10 hPa of up to 118 ms$^{-1}$ in January.

The dynamical conditions in winter 2019/2020 with low planetary wave activity result in a strong radiative cooling of the polar lower stratosphere during polar night, especially in January, February and March, which causes a strong polar vortex as response. In the following, our analyses of lower stratosphere temperatures are focusing on the 50 hPa pressure level (about 20 km altitude), which is the height range of vital importance for ozone depletion. Figure 5 shows that the monthly mean temperatures in January, February and March 2020 were very low in comparison with the respective mean values calculated for the last 4 decades (1979/1980-2019/2020). In March 2020 the calculated maximum temperature difference with respect to the long-term mean was −23.8 K. In Figure 6 (magenta line) minimum polar temperatures below 195 K at 50 hPa (i.e. the Cl activation threshold at this altitude, see for instance Figure 4-1 of Chapter 4 in WMO, 2018) are detected in the polar cap region (50°-90°N) from the beginning of December until end of March. Further analyses of the temperature field at 50 hPa indicate large areas below 195 K. This result is again in agreement with the analyses by Lawrence et al. (2020), which were based on MERRA-2, and by Wohltmann et al. (2020), which were based on ERA5. As indicated in Figure 7 (magenta line with dots in the figure) the maximum daily mean area of temperatures below 195 K is $13 \cdot 10^{12}$ m$^2$, which is found end of January. At the end of March, the daily cumulative area below 195 K, i.e. the sum of the daily areas below 195 K up to the respective date, results in about $920 \cdot 10^{12}$ m$^2$. This led to conditions allowing the formation of polar stratospheric clouds (PSC) of type I (Nitric Acid Trihydrate (NAT) particles) at 50 hPa for about 3.5 months (see Figures 6 and 7). Our results were supported by Lawrence et al. (2020) and Wohltmann et al. (2020), who were analyzing among others the volume of PSCs (see also Manney et al., 2020). When the sun rises in spring, sunlight delivers the energy required for starting a chemical depletion process of ozone (e.g., Solomon, 1999; Dameris, 2010). Stratospheric ozone can then be destroyed by heterogeneous chemical reactions due to the still enhanced atmospheric chlorine content (caused by CFC emissions in last decades). In spring 2020 record low Arctic TOC values below 220 DU developed within the boundaries of the stable polar vortex for eight continuous days from March 12 to 19 (see Figure 1 and also the magenta line in Figure 8). A region of significantly reduced TOC values with an ozone hole-like pattern was observed over the polar cap from the beginning of March until early April 2020 (Figure 1).

In Figures 3 and 6 exemplarily corresponding values of mean zonal winds and minimum polar cap temperatures in the Antarctic are shown. In particular, a typical, undisturbed Southern hemisphere situation in 2016 (red lines) and the situation in 2019 (purple lines) with a dynamically disturbed spring season are shown. It becomes evident that in winter the zonal mean zonal

winds (at 60°S, 10 hPa) are stronger (by about 30 ms$^{-1}$) and the minimum temperatures (polar cap, 50 hPa) are much lower (about 10 K).

The temporal evolution of minimum TROPOMI TOC values north of 50°N from July 2019 until April 2020 is presented in Figure 8 (magenta line) and compared with historical values from the GOME-type Total Ozone Essential Climate Variable (GTO-ECV) data record (see Section 2 for details). In winter 2019/2020 ozone values were most of the time slightly below mean conditions until the end of February with respect to mean minimum TOC values (Figure 8, magenta line vs. thick black line). But there were several short-term deviations towards even lower TOC, during so-called ozone mini-hole events (e.g., Millán and Manney, 2017). The most noteworthy examples occurred in early December 2019 (Dec 3 and Dec 4), beginning of January 2020 (Jan 4 and Jan 5, and Jan 7 and Jan 8), and the end of January (Jan 25 to Jan 27). Ozone mini-holes are synoptic-scale features (with a high-pressure system in the troposphere below the stratospheric polar vortex, i.e. a low-pressure area) with significantly reduced TOC values. It is well understood that ozone mini-holes are primarily resulting from dynamical processes (e.g., Millán and Manney, 2017). The positions of the mini-holes correlate well with minima of potential vorticity near the tropopause (Peters et al., 1995; James and Peters, 2002). Hoinka et al. (1996) found that about 50% of short-term TOC fluctuations in the NH can be explained by variations of the tropopause pressure. Furthermore, Steinbrecht et al. (1998) showed that an increase of tropopause height by one kilometer is connected with a reduction of TOC by 16 DU. Figure 8 illustrates that such mini-hole events occur regularly (the lower light grey line) during Northern winter. Very commonly the ozone mini-holes are created in the Northern Atlantic region and then they often drift eastward towards Northern Europe within a few days (James, 1998). This was also the case for the three examples seen in winter 2019/2020 with minimum TOC found over Northern Europe (not shown). Since the polar vortex existed already in late November and early December 2019 with lower than usual TOC, for instance the ozone mini-hole on December 3 and 4 showed very low TOC values (170 DU; Figure 8) at 65°N, north-east of UK and west of Scandinavia.

Because the polar vortex was stable and strong since the beginning of the winter 2019/2020, an ozone hole-like pattern with reduced TOC values inside the polar vortex and higher TOC values outside was observed from January 2020 onwards, but with TOC values clearly above 220 DU inside the vortex. As indicated in Figure 4, the stable polar vortex with persistent strong zonal winds and strong PV gradients (not shown) prevented the meridional transport of ozone rich air from lower latitudes towards the Northern polar region. Indeed, Lawrence et al. (2020) showed that the undisturbed polar vortex acted as a strong transport barrier. Among others, this is indicated by ozone sonde and ground-based measurements at different NH stations with lower TOC values in the inner part of the polar vortex and higher TOC values outside. Respective ozone data are available at http://www.temis.nl/uvradiation/UVarchive/stations_uv.html (van Geffen et al., 2017), and also available at https://woudc.org/data/explore.php, and at https://www.ndacc.org (see also Wohltmann et al., 2020). After mid-February the ozone hole-like feature can be identified also in the TROPOMI TOC values, which is indicated by a strong horizontal gradient in the vicinity of the polar jet with strongest zonal winds.

Remarkable deviations from normal Arctic conditions could be found starting in early March 2020 until early April, when low TOC values in the North polar region were detected (magenta line in Figure 8): the long period of unusually low TOC started in early March 2020, falling below 220 DU for the first time on March 2, and continued with similarly low TOC – including a period of 8 consecutive days with minimum TOCs below 220 DU – until April 7. For the first time TOC values near or below 220 DU unrelated to ozone mini-hole events were observed for a period of about 5 weeks corresponding to new record low values for this time of the year. The maximum area with TOC below 220 DU was determined with 0.9 million km$^2$ ($= 0.9 \cdot 10^{12}$ m$^2$), which was detected on March 12 (Figure 1). This is in the order of 4% of the polar vortex determined for the isentropic surface at 475 K with respect to the 36 PV unit contour (i.e. the edge of the polar vortex; e.g., Wohltmann et al., 2020). For March 12, the size of the polar vortex is 21.75 million km$^2$ (Figure 2). In comparison with corresponding values of a typical ozone hole in the Antarctic (here 2016; red line in Figure 8; Figure 9, bottom left) the area of low TOC (below 220 DU) is much smaller and the minimum TOC are clearly higher. The Antarctic ozone hole in spring 2016 showed minimum total ozone columns clearly below 150 DU and TOC values below 220 DU were found for a period of about 4 months. The maximum area of the ozone hole was in the order of 20 million km$^2$ (Figure 9 bottom left). Even the maximum area of record low TOC values (below 220 DU) detected in the Arctic in spring 2020 (Figure 9, top right), is just about 10% of the exceptionally small Antarctic ozone hole observed in Southern hemisphere (SH) spring 2019 (see Wargan et al. (2020), who reported an ozone hole area of roughly 5-10 million km$^2$ during September and October 2019; see also Figure 9, bottom right).

## 4 Situations in Northern winter and spring 1996/1997 and 2010/2011

There are two other prominent spring periods in the NH, which showed similarly stable and cold stratospheric polar vortices in spring. In particular, comparable dynamical conditions in the Northern stratosphere were observed in February and March 1997 (e.g., Coy et al., 1997; Manney et al., 1997; Lefèvre et al., 1998; Hansen and Chipperfield, 1999) and 2011 (e.g., Manney et al., 2011; Sinnhuber et al., 2011; Kuttippurath et al., 2012; Hommel et al., 2014).

We note that the characteristics of the polar vortex - which in turn have direct or indirect consequences for the TOC, either by chemical ozone depletion or (meridional) transport of air masses - vary in different Northern winters (e.g., Tegtmeier et al., 2008). With respect to the intercomparison of various winter/spring seasons (here 1996/1997, 2010/2011, and 2019/2020), the analysis of the evolution of the polar vortex at 10 hPa and the minimum temperatures at 50 hPa are commonly used for the initial evaluation of the dynamical state of the stratosphere in (see e.g. Lawrence et al., 2020).

In Figure 3 the temporal evolution of the two stratospheric polar vortices in the NH in 1996/1997 (blue line) and 2010/2011 (green line) is indicated by the zonal mean zonal wind speed at 60°N, 10 hPa. In comparison with the dynamical situation in January, February and March 2020 (magenta line in Figure 3), the respective time periods in 1997 and 2011 showed also a persistent polar vortex with high zonal wind speeds, which reached values of up to more than 50 ms$^{-1}$. These values are higher than the long-term mean values, which show an increase up to 40 ms$^{-1}$ until the beginning of January and a decrease afterward

(see also Figure 1 in Lee and Butler, 2020). While the temporal evolution of the dynamical situation in Arctic spring 2011 was very similar to the one in 2020 with a persistent polar vortex and high zonal wind speeds until mid-March, the period of strong zonal winds in 1997 continued until April. The polar vortex in December 1996 was weak and therefore polar temperatures were relatively high (higher than 195 K; see below). The evolution of the winter vortices in December 2010 and 2019 are
similar reaching zonal wind speeds of about 40 ms$^{-1}$ in mid-December. Similar values were shown by Lawrence et al. (2020) who were looking at MERRA-2 data. In this context, Manney et al. (2011) showed that in 2011 the meridional transport was weak at the edge of the polar vortex (i.e. a strong barrier) throughout the winter, i.e. less ozone was transported from lower latitudes to higher latitudes, whereas the meridional transport was enhanced in 1997 because the polar vortex was weaker in December and January.

In all three years the respective February and March dynamical conditions led to low stratospheric temperatures in the polar cap region (50°N-90°N). In Figure 6 the temporal evolution (based on daily values) of the detected minimum temperatures at 50 hPa is shown. In February and in March the minimum temperatures were below 195 K in all three years, the threshold temperature value for the formation of polar stratospheric clouds (NAT-PSC). The determined minimum temperatures in December 2019 and January 2020 were mostly lower at 50 hPa compared to the minimum temperature values detected in
December/January 1996/1997 and December/January 2010/2011. The minimum values of the monthly mean temperatures are given in Table 1, indicating the characteristic of low temperatures in December 2019 and January and February 2020 (see also the colored dots in Figure 6).

**Table 1: Minimum temperatures (in Kelvin) of the polar cap region (50°N-90°N) at 50 hPa (about 20 km altitude) based on the**
**monthly mean temperatures from ERA5 in December, January, February and March of 1996/1997, 2010/2011, 2019/2020 and the long-term mean values (1979/1980-2019/2020), respectively.**

| Min. temp. 50 hPa | December | January | February | March |
|---|---|---|---|---|
| 1996/1997 | 201.3 | 196.5 | 191.8 | 192.7 |
| 2010/2011 | 195.6 | 194.2 | 191.2 | 194.7 |
| 2019/2020 | 194.3 | 190.7 | 190.8 | 194.6 |
| Long-term means (1979/1980-2019/2020) | 197.0 | 195.6 | 199.5 | 205.5 |

A series of studies were published in the past highlighting the differences of the two winters 1996/1997 to 2010/2011, especially regarding polar chemical processes (e.g., Manney et al., 2011; Kuttippurath et al., 2012; Chapter 3 in WMO, 2014). It turned out that in spring 2011 very strong chemical ozone loss occurred (Manney et al., 2011). In spring 1997 the chemical ozone loss was only moderate (Manney et al., 1997; Tegtmeier et al., 2008). The most important reason, which was mentioned

in these studies, was the late development of the polar vortex and late drop of temperatures below PSC thresholds in winter 1996/1997 (see also Table 1 and Figures 6 and 7). Recent studies found that the chemical ozone loss in spring 2020 was stronger than in spring 2011 (e.g., Manney et al., 2020; Wohltmann et al., 2020; Grooß and Müller, 2020).

As demonstrated in Figure 7, the daily areas with temperatures below 195 K at the 50 hPa pressure level are obviously larger in 2019/2020 (magenta line) than in 1996/1997 (blue line) and 2010/2011 (green line). In particular, the cumulative areas are

markedly different: whereas in 2019/2020 the cumulative area was about $920 \cdot 10^{12}$ m$^2$, in 1996/1997 it was about $370 \cdot 10^{12}$ m$^2$ and in 2010/2011 it was about $650 \cdot 10^{12}$ m$^2$. Furthermore, in the last week of January 2020 the temperatures at 50 hPa went below 188 K (magenta line in Figure 6), the typical ice PSC threshold (PSC type 2, ICE-PSC; see for instance Figure 4-1 of Chapter 4 in WMO, 2018). The maximum daily area with temperatures below 188 K was determined with $2.8 \cdot 10^{12}$ m$^2$ on January 30, and the cumulated area reached its maximum of $18 \cdot 10^{12}$ m$^2$ on March 2, 2020. While the threshold for ICE-PSC

was not reached in 1996/1997, in 2010/2011 the cumulative area with temperatures below 188 K was estimated with $4.3 \cdot 10^{12}$ m$^2$.

To summarize, in all three years the temperatures in the lower stratosphere in February and March were in a similar range, showing colder conditions than usual. Our analyses show that December 2019 and January 2020 were also clearly colder than the long-term mean conditions (Table 1). The minimum temperatures in the lower stratosphere in December/January

2019/2020 were lower than in December/January 1996/1997 and 2010/2011. It is worth mentioning that December 1996 was much warmer than December 2010. The winter 2019/2020 showed a larger area below the formation temperature of PSCs than in the other two Northern winters for an extended period of time (see Figure 7). Having permanent presence of polar stratospheric clouds over about four months enabled more efficient chlorine activation. In addition, they supported strong denitrification of the lower stratosphere by irreversible removal of total reactive nitrogen (NO$_y$), especially HNO$_3$, due to

heterogeneous reaction on the surface of PSCs followed by sedimentation of PSC particles (Fahey et al., 1990). This ultimately enabled a longer than usual period of chemical ozone depletion (e.g., Fahey et al., 1990; Rex et al., 1999, Pommereau et al., 2018). The occurrence of denitrification for the winter 2019/2020 was detected by Manney et al. (2020), who analyzed the data of the space-borne MLS instrument, indicating that denitrification was much stronger in 2020 than in 2011. Further, in Manney et al. (2011) it was shown that denitrification was clearly stronger in 2011 than in 1997.

The temporal evolution of minimum TOC values north of 50°N between July 1996 and June 1997 (blue line in Figure 8) and between July 2010 and June 2011 (green line in Figure 8) indicates normal or slightly enhanced ozone values until February with respect to the long-term mean value (thick black line), which is based on satellite observations from 1995 to 2019. In February 1997 and February 2011, TOC maps from the Northern polar region were both showing typical features of a stronger

polar vortex with lower TOC values within the vortex and relatively high TOC values in the collar region of the polar vortex (is not shown here). Around the beginning of March 1997 and March 2011 the TOC values were declining and low TOC values were detected in both years until early April. An important point to be mentioned is that in spring 1997 the dynamical conditions led to frequent ozone mini-holes (Coy et al., 1997) and to a higher tropopause (Manney et al., 2011) that obviously

contributed to lower TOC values via dynamical processes and which were then followed by a typical chemical ozone loss (Manney et al., 2011).

Figure 9 (top row) is showing the TOC monthly means for March 1997, 2011, and 2020. Three ozone hole-like patterns with low TCO values over the polar cap can be seen. The lowest Arctic TOC values are clearly detected in boreal spring 2020. In spring 1997 and 2011 TOC values below 220 DU were not detected over larger areas and over several consecutive days. The

monthly mean minimum TOC value for March 2020, which is 221 DU, is much lower compared to the monthly mean minimum TOCs for March 1997 (267 DU) and for March 2011 (252 DU). The temporal evolution of minimum Antarctic TOC values south of 50°S presented in Figure 8 (the red line for 2016 and the purple line for 2019) and the TOC monthly means for October 2016 and 2019 in Figure 9 (bottom row) show that the size and strength of the Antarctic ozone hole is much larger than the corresponding values detected for the Arctic in spring 2020.

**5 Discussion**

The Arctic winter and early spring TOC variability in the recent decades is in large parts reflecting the natural fluctuations of the stratospheric dynamics of the Northern hemisphere (NH) during this period (Chapter 4 in WMO, 2018). The same can be stated about the Southern hemisphere (SH), where in most cases the interannual fluctuations of the strength of the Antarctic ozone hole can be explained by different dynamical conditions of the stratosphere (e.g., Chapter 4 in WMO, 2018). Dynamical

conditions of the Northern stratosphere at higher latitudes in winter can range from a very disturbed polar vortex (i.e. by strong planetary wave activity), which leads to high stratospheric temperatures, to conditions with a persistent stable polar vortex (i.e. by low planetary wave activity), which creates low stratospheric temperatures.

Therefore, on the one hand, it is possible to find extreme situations with strong mixing of air masses in the polar regions (for instance during major stratospheric warmings) in combination with reduced chemically induced ozone depletion, which leads

to enhanced TOC values. On the other hand, situations with significantly suppressed meridional air mass exchange and transport into the polar vortex area can be found in combination with enhanced ozone depletion by heterogeneous chemical processes inside the polar vortex, which causes a clear reduction of TOC. The latter was the case in winter and spring 2019/2020. Our comparative analysis of 2019/2020 with respect to the last four decades (i.e. the length of the currently available ERA5 data set) indicates an extraordinary dynamical situation with a persistent strong and cold polar vortex over the

complete winter season. There is some evidence that a similar dynamical event did not happen in the period from 1955 to 1980, i.e. before the starting point of our dynamical analyses based on ERA5. An analysis of the historical data set was provided

by the Stratospheric Research Group at FU Berlin. In Labitzke and Naujokat (2000) it was stated that "The spring of 1997 was the coldest within our series of 45 winters."; the Berlin time series is ranging from 1955 to 2000. Among others, they compared the monthly mean North Pole temperatures at 30 hPa in all years and found that February 1997 (190 K) and March 1997 (194 K) were clearly colder than in all other years. The second coldest spring was detected in 1967 (February: 195 K; March: 201 K). January 1997 turned out to be normal with respect to the climatological mean value. In combination with our research results this suggests that the dynamical situation of the winter and spring 2019/2020 is outstanding since the beginning of monitoring the stratosphere in the 1950s. Although the historical Berlin data set does not have the same quality as for instance ERA5, they are suitable for a qualitative evaluation of the respective dynamical situations of Northern winter and spring seasons. Lawrence et al. (2020) looked at the Japanese Meteorological Agency's 55-year reanalysis (JRA-55), which goes back to the winter 1958/1959. Based on their investigations of the zonal mean wind at 60°N, 10 hPa, they found that the winter 2019/2020 ranked on the third position. The two winters of 1966/1967 (i.e. in the Berlin analysis it ranked second after 2019/2020) and 1975/1976 (i.e. in the Berlin analysis it indicated as a cold winter, but not extraordinary) turned out to be the strongest on record with respect to the zonal mean wind (mean from December to March). This slightly different classification of record years indicates that it depends on the chosen meteorological variable of analysis as well as the altitude and latitude region. Further it is clear that the data before 1980 (pre-satellite-era) are more uncertain. Nevertheless, it is obvious that the winter 2019/2020 is one of the coldest with a strong and stable polar vortex in the last 65 years.

We note that the stratospheric dynamical conditions were completely different in the Northern winter 2018/2019 (brown line in Figure 3) compared to 2019/2020 (magenta line in Figure 3) showing a sudden major stratospheric warming event, which started in late December 2018. In the first half of January 2019 the direction of the mean zonal wind (60°N, 10 hPa) changed from westerlies to easterlies. This strong disturbance of the polar vortex by planetary waves led to a pronounced warming of the lower stratosphere (e.g., Lee and Butler, 2020), indicating minimum temperatures in the polar cap region at 50 hPa, which were clearly above the threshold for the formation of NAT-PSC (195 K) for the complete winter season including early spring (brown line in Figure 6). Consequently, comparatively high TOC values (around the long-term mean) in the Arctic region were found from late winter to early spring (not shown).

There were two other examples of the importance of stratospheric dynamics regarding the development of low TOC values in SH spring in 2002 and 2019. In September 2002 a sudden major stratospheric warming in connection with a breakdown of the polar vortex was detected, which led to a split of the ozone hole (Sinnhuber et al., 2003; Allen et al., 2003; Hoppel et al., 2003; Stolarski et al., 2005). The other example happened in September 2019. The polar vortex was significantly disturbed (Wargan et al., 2020). Figure 3 (purple line) shows that the zonal mean west wind speed at 60°S, 10 hPa dropped from about 80 ms$^{-1}$ to 10 ms$^{-1}$. A strong minor polar stratospheric warming (roughly 30 K at 80°S, 50 hPa in about two weeks, not shown) happened over the Antarctic in the first half of September (see e.g., Lim et al., 2020). Figure 6 (purple line) indicates an increase of the minimum temperature in the polar cap (50° to 90°S) at 50 hPa in September of about 20 K. Afterwards the stratospheric polar vortex was weak (see purple line in Figure 3) leading to much warmer conditions in the polar region, with unusual high

stratospheric temperatures for this time of the year. After mid-September minimum polar cap temperatures at 50 hPa were consistently higher than 195 K (see purple line in Figure 6). The minimum TOC values in the Antarctic were noticeably higher in 2019 than in previous years (Wargan et al., 2020; see the purple line in Figure 8; see also https://public.wmo.int/en/media/news/antarctic-ozone-hole-smallest-record; October 24, 2019). Nevertheless, it is obvious that the small ozone hole in Antarctic spring 2019 is still much larger than the ozone hole-like feature with record low TOC values detected in Arctic spring 2020 (see also Figure 9).

The Arctic observations in winter and spring 2019/2020 are consistent with our expectation that Arctic ozone reductions in spring are largest after stratospheric winters with a stable, circular polar vortex in connection with low polar lower stratosphere temperatures (Chapter 4 in WMO, 2018). However, as can be seen in Figure 6, the temperatures in the Antarctic are considerably lower than in the Arctic (even for 2019/2020) and that the period of low temperatures is much longer in the Antarctic. As a result, although the dynamic features are similar, the record low Arctic TOC values are much higher than the observed TOC in the Antarctic, also in the case of the small ozone hole in Antarctic spring 2019, as indicated in Figure 8.

The dynamical conditions in the Arctic stratosphere in February and March 2020 were similar to the other two exceptional stratospheric dynamics situations in early spring 1997 and 2011. Further, all three years showed low TOCs in March (Figure 8). Noteworthy about March 2020 is that minimum TOC values were below 220 DU for several days, although the stratospheric chlorine content was clearly higher in 1997 (about 15%) and slightly higher in 2011 (Chapter 1 in WMO, 2018). Our comparisons show that especially in December 2019 and January 2020 the temperatures in the lower stratosphere were lower than in the two other years discussed here. In 2019/2020 the minimum polar cap temperatures at 50 hPa were all the time below 195 K (the threshold for formation of NAT-PSC) in December, January, February and most of March. In this context Manney et al. (2020) identified an activation of chlorine already in the beginning of December. Our analyses show that the daily areas allowing for the formation of PSC at 50 hPa were clearly larger in 2019/2020 in comparison with the winters 1996/1997 and 2010/2011. This finding is in line with the results presented by Lawrence et al. (2020). The ERA5 data set also indicates minimum temperature values in winter 2019/2020, which were slightly above or below 188 K for a week (in particular at the end of January 2020), providing conditions for the formation of ICE-PSCs (see also Manney et al., 2020). It is obvious that stratospheric temperatures, especially in December and January, compared to 1996/1997 and 2010/2011 were much lower in 2019/2020.

Since the polar vortex in winter and spring 2019/2020 provided continuous conditions for the formation of PSCs, significant denitrification of the stratosphere occurred, i.e., a permanent removal of total reactive nitrogen (NO$_y$, primarily HNO$_3$) by the sedimentation of NAT-PSC particles was found (Manney et al., 2020). In this case, this contributed to the five weeks period of significant TOC reduction by an extended phase of active stratospheric chlorine. However, the observed minimum TOC values in March 2020 with new low TOC records for the NH polar cap were pointing to substantial ozone depletion in spring. Here we note again that 2020 was also starting at lower base values of TOC (inside the polar vortex; see Figure 8), which was especially caused by reduced meridional transport of ozone from lower to higher latitudes due to the stable polar vortex during

this winter. This also contributed to the fact that the spring TOC values in the Arctic region in 2020 were clearly lower than those found in 1997 and 2011.

The situation with record low stratospheric ozone values over the Arctic in 2020 is not an unequivocal result of climate change. The dynamical situations in February and March of 1997 and 2011 in comparison with 2020 were similar. Beyond that the cold stratosphere in December 2019 and January 2020 as a single event does not point towards climate change due to increasing greenhouse gas concentrations. The NH winter 2019/2020 is a perfect showcase that a Northern winter with less planetary wave activity, and therefore a strong and stable vortex with low temperatures is possible. Only if similar conditions would happen more regularly in the next years, then this could be a sign of climate change. Although the stratosphere is more or less cooling steadily due to increasing greenhouse gas concentrations (Maycock et al., 2018; Steiner et al., 2020), consequences for stratospheric dynamics particularly in winter and ozone depletion in spring are still under debate (e.g., Bednarz et al., 2016; Ivy et al., 2016; Pommereau et al., 2018). For instance, the empirical quantification of the relation between winter-spring loss of Arctic ozone and changes in stratospheric climate by Rex et al. (2004) showed the possibility that cold (Northern) winters are getting colder in future. The investigations by Wohltmann et al. (2020) seem to support this hypothesis. It is possible that the cooling of the (lower) stratosphere could delay the recovery of the ozone layer (Pommereau et al., 2018). But this statement is in contradiction with results derived from Chemistry-Climate model predictions (e.g., Dhomse et al., 2018) indicating that climate change in the NH will accelerate stratospheric ozone recovery instead of delaying it (see also Chapters 3 and 4 in WMO, 2018). Since the changes of stratospheric temperature are affected not only by radiative cooling due to enhanced greenhouse gas concentrations, but also by respective atmospheric circulation changes (e.g., Langematz et al., 2014), the quantitative determination of net effect on ozone is still a challenge. Furthermore, more than 20 years ago model calculations by Waibel et al. (1999) showed that higher degrees of Arctic denitrification in future, related to stratospheric cooling by enhanced greenhouse gas concentrations, could lead to larger seasonal ozone depletion despite the projected decline in inorganic chlorine.

Finally, based on our current knowledge we deem it unlikely that the observed enhanced CFC-11 emissions in recent years (Montzka et al., 2018) have significantly influenced the ozone depletion in the NH in 2020 (Dameris et al., 2019; Fleming et al., 2020; Keeble et al., 2020).

## 6 Conclusions

This study presents a consistent description of the Northern winter and spring season 2019/2020 considering the dynamical situation of the stratosphere and the evolution of the ozone layer in the Arctic region. Record low TOC values were detected in the vicinity or below 220 DU over a larger area (up to $0.9$ million $km^2$) and for a longer time period (about five weeks). The 2019/2020 situation is compared with other years, which showed similar stratospheric dynamics in spring. We have used the most recent datasets for the preparation of presented analyses, (i) meteorological data from ERA5 and (ii) total ozone column

data sets, i.e. GTO-ECV (based on the European satellite sensors GOME/ERS-2, SCIAMACHY/ENVISAT, OMI/Aura, and GOME-2/MetOp) in combination with TROPOMI onboard Sentinel-5P. Although the detected Arctic ozone hole-like feature is much smaller in comparison with a typical Antarctic ozone hole – which is in the order of about 20 to 25 million km$^2$ (from early September until mid-October) and TOC values below 220 DU are seen for up to about four months (WMO, 2018) – it is

an extraordinary event because TOC values below 220 DU were not observed before over a period of 5 weeks. The results of our study pointed out that the persistent strong polar vortex in 2019/2020 (from mid-December to early April) led to particularly cold stratospheric conditions for the complete winter and early spring season, supporting the process of ozone depletion through heterogeneous chemistry. We have discussed the special dynamical situation in winter 2019/2020, which is relevant for this significant reduction of the TOC in spring 2020, despite the observed decrease in the stratospheric chlorine content

over the last 2 decades.

We note that currently numerous studies, which are based on observational, reanalyses and modelling data are being, or have been prepared to draw a holistic picture of this special winter season. In particular, a couple of these studies have been (e.g., Manney et al., 2020; Wohltmann et al., 2020) or are about to be published (e.g., Lawrence et al., 2020; Grooß and Müller, 2020) in a Geophysical Research Letters and Journal of Geophysical Research - Atmospheres special issue regarding this topic.

If the regulations of the Montreal Protocol regarding the prohibition of CFCs are implemented strictly one can expect a full recovery of the ozone layer including the polar regions by the middle of this century (Chapters 3 and 4 in WMO, 2018). In recent years, the beginning of ozone recovery was already detected (e.g., Solomon et al., 2016; Weber et al., 2018). However, in the upcoming decades ozone holes will still occur in the SH. But also, in the NH under appropriate dynamical conditions with a stable polar vortex yielding a strong cooling of the polar lower stratosphere regions of persistent low TOC as found in

NH spring 2020 might develop. The recovery of the ozone layer and its interactions with climate change must be carefully documented, as discussed for instance by Dameris and Loyola (2011). The data available from the different monitoring instruments enable well founded scientific explanations of special ozone features. For instance, these data help to explain the recent evolution of the ozone layer, in particular the occurrence of the record low total ozone values in Arctic spring 2020. This capability is crucial to allow an evaluation of specific events in the light of the Montreal Protocol.

*Data availability.* Meteorological data is based on ERA5 from ECMWF (https://cds.climate.copernicus.eu/#!/search?text=ERA5&type=dataset), which is available at the Climate Data Store (CDS). This work contains modified Copernicus Climate Change Service information (Copernicus Climate Change Service (C3S),

2017; Hersbach et al., 2018; 2019a). Neither the European Commission nor ECMWF is responsible for any use that may be made of the Copernicus information or data it contains. In particular, subsets, i.e. wind and temperature data, from the pressure level data sets of monthly averaged data (Hersbach et al., 2019a) and hourly reanalysis data (Hersbach et al., 2018) have been

used. Further, hourly PV data on isentropes from the full ERA5 data set (Copernicus Climate Change Service (C3S), 2017) have been exploited. We thank the ECMWF for producing ERA5 data and making it available through the CDS. The used data contains modified Copernicus Climate Change Service information, in particular with respect to Figures 2, 3, 4, 5, 6, and 7 and Table 1. Please note, that the data used here may also contain "preliminary" ERA-5 data (cf. Hersbach et al., 2020).

The GTO-ECV Climate Research Data Package (ESA CCI, 2020) is available at http://cci.esa.int/ozone/ (last access: April 6, 2020), detailed information about this data record can be found at https://atmos.eoc.dlr.de/gto-ecv (last access: April 6, 2020). This data source is used here, in particular with respect to the preparation of Figures 8 and 9.

The (Level 2) TROPOMI total ozone column data (TROPOMI OFFL TOC; Copernicus Sentinel-5P, 2018) are available at https://s5phub.copernicus.eu/ (last access: May 18, 2020) and https://s5pexp.copernicus.eu/ (last access: May 18, 2020). This
paper contains modified Copernicus Sentinel-5 Precursor data processed by DLR/BIRA/ESA. This data is used here, in particular with respect to Figures 1, 8, and 9.

*Author contributions.* MD structured and composed the paper. MD, DGL, MCE and MN jointly analyzed the different data sets and compiled the results including the preparation of the figures. MD, DGL, MCE and MN contributed to the writing of
the manuscript. MCE, DGL, CL, and MvR generated the GTO-ECV data in the ESA project Ozone_cci+ and the EU/ECMWF project C3S_312b. CL, FR, DGL and MvR are responsible for the TROPOMI TOC Level 2 data in the ESA project S5P-MPC.

*Competing interests.* The authors declare that they have no conflict of interest.

*Acknowledgements.* First, we would like to thank Birgit Hassler for an internal review of the first draft of the manuscript. The
work for this study was supported under the umbrella of the DLR-project MABAK (Innovative Methoden zur Analyse und Bewertung von Veränderungen der Atmosphäre und des Klimasystems). The work described in this paper has also received funding from the ESA-projects "Ozone_cci" and "Ozone_cci+" (as part of the ESA Climate Change Initiative (CCI) program) and the Initiative and Networking Fund of the Helmholtz Association through the "Advanced Earth System Modelling Capacity (ESM)" project. The NCAR Command Language (NCL, 2018) was used for data analysis and to create some of the
figures   in   this   study.   NCL   is   developed   by   UCAR/NCAR/CISL/TDD   and   is   available   online   at https://doi.org/10.5065/D6WD3XH5. CDO (climate data operators; Schulzweida, 2019) was employed for processing the data.

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

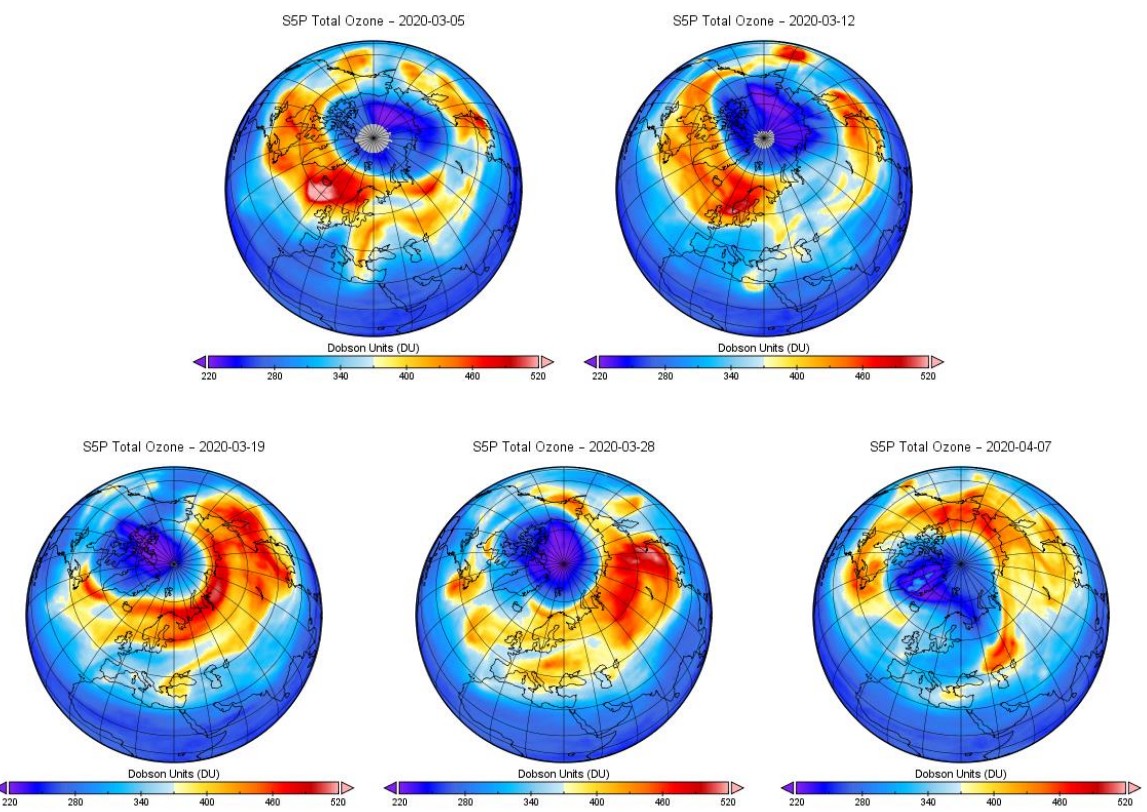

**Figure 1: Total ozone column over the Northern hemisphere on March 5, 12, 19, 28 and April 7, 2020 measured by the TROPOMI instrument onboard the Sentinel-5 Precursor (S5P) satellite. Color scale is showing Dobson units (DU). The area with total ozone values below 220 DU is denoted with the dark purple color.**

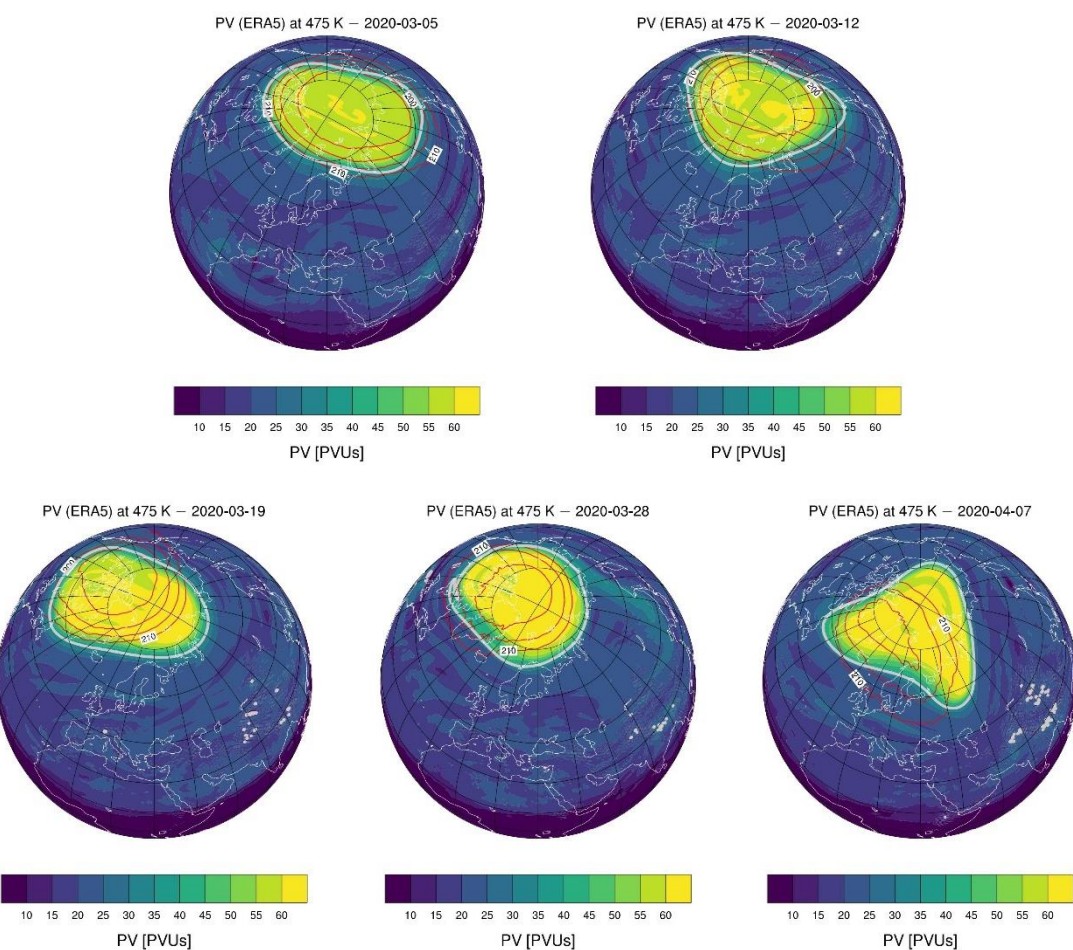

**Figure 2: Potential vorticity (PV) on the 475 K isentropic surface over the Northern hemisphere on March 5, 12, 19, 28 and April 7, 2020 based on ERA5. Color scale is showing potential vorticity units (PVUs). 1 PVU is $10^{-6}$ m$^{-2}$ s$^{-1}$ K kg$^{-1}$. The grey contour is highlighting the 36 PVU, which describes the position of the polar vortex edge. Red contour lines are indicating the polar temperature field with values ranging from 210 K to 195 K; the distance of isolines is 5 K.**

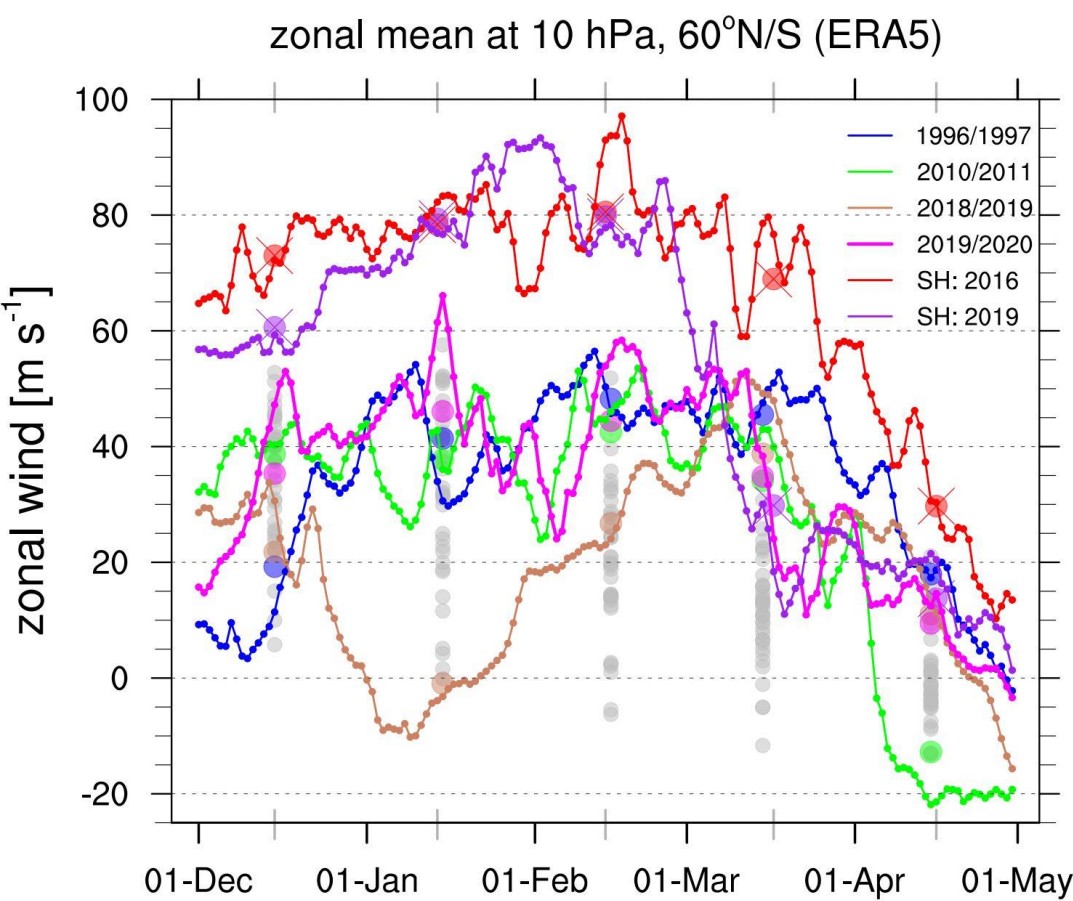

**Figure 3: Daily (lines) and monthly (dots; in the center of each month) mean zonal mean zonal wind (in m s⁻¹) at 10 hPa (about 30 km altitude): the Northern hemisphere winters 1996/1997, 2010/2011, 2018/2019 and 2019/2020 at 60°N from December 1 to April 30 based on ERA5 data are displayed in blue, green, brown and magenta (lines and dots); the Southern hemisphere winters 2016 and 2019 at 60°S from June 1 to November 1 (attention: the respective data are shifted by six months) based on ERA5 data are displayed in red and purple (lines and dots with crosses). Additional monthly means for the Northern hemisphere winters from 1979/1980 to 2019/2020 are shown as grey dots in the center of each month. For simplicity, the leap day in 2020 (February 29) was neglected in the daily time series.**

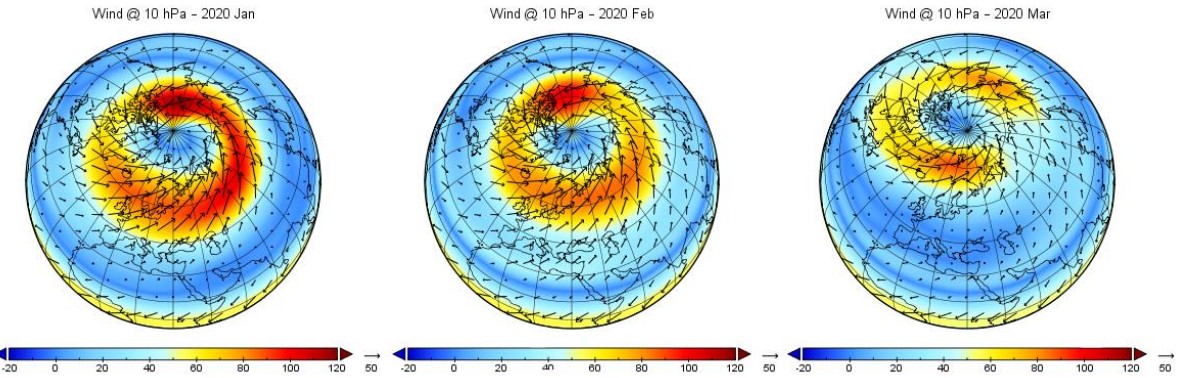

**Figure 4: ERA5 monthly mean wind at 10 hPa (about 30 km altitude) showing a strong vortex in the Northern polar region with speeds of up to 118 m s⁻¹, 103 m s⁻¹, and 89 m s⁻¹ for January to March 2020 respectively.**

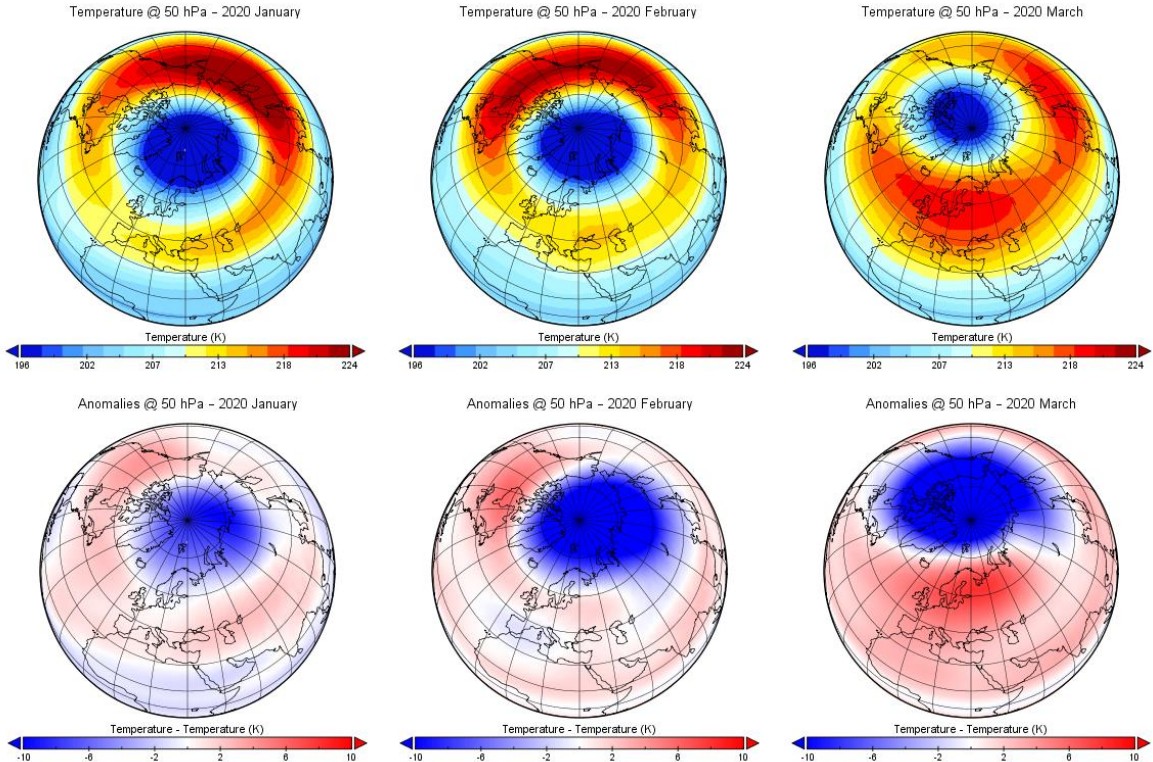

5  **Figure 5: ERA5 monthly mean temperature at 50 hPa (about 20 km altitude) for January to March (columns 1 to 3) for year 2020 (top row) and the corresponding temperature anomalies (lower row) with respect to the average from 1979-2019 showing negative differences of up to -9.93 K in January, -18.44 K in February, and -23.83 K in March 2020.**

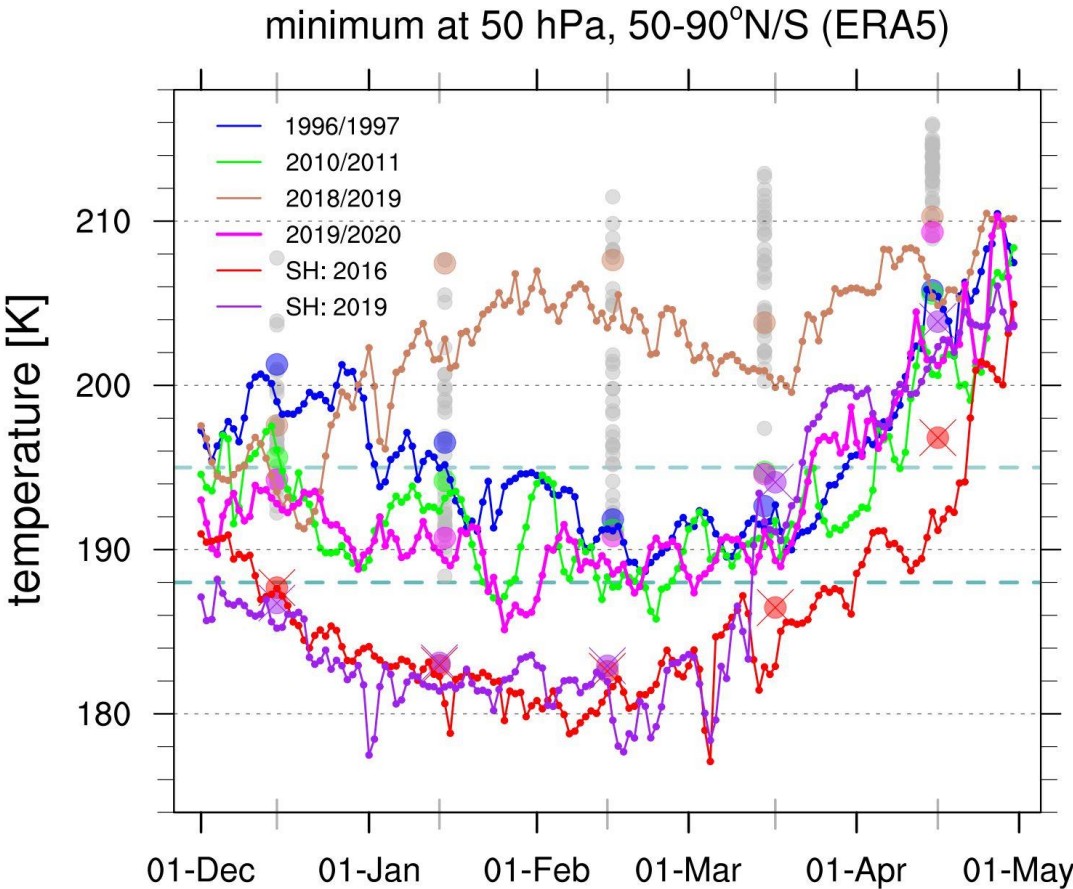

**Figure 6: Minimum daily (lines) and monthly (dots; in the center of each month) mean temperatures (in K) at 50 hPa (about 20 km altitude): the Northern hemisphere winters 1996/1997, 2010/2011, 2018/2019 and 2019/2020 for 50°N-90°N from December 1 to April 30 based on ERA5 data are displayed in blue, green, brown and magenta (lines and dots); the Southern hemisphere winters 2016 and 2019 for 50°N-90°N from June 1 to November 1 (attention: the respective data are shifted by six months) based on ERA5 data are displayed in red and purple (lines and dots with crosses). Additionally, the minima of the monthly mean temperature data for the Northern hemisphere winters from 1979/1980 to 2019/2020 are shown as grey dots in the center of each month. For simplicity, the leap day in 2020 (February 29) was neglected in the daily time series. The dark green broken horizontal lines at 195 K and 188 K marked the thresholds for the formation of NAT-PSC and ICE-PSC, respectively (see text).**

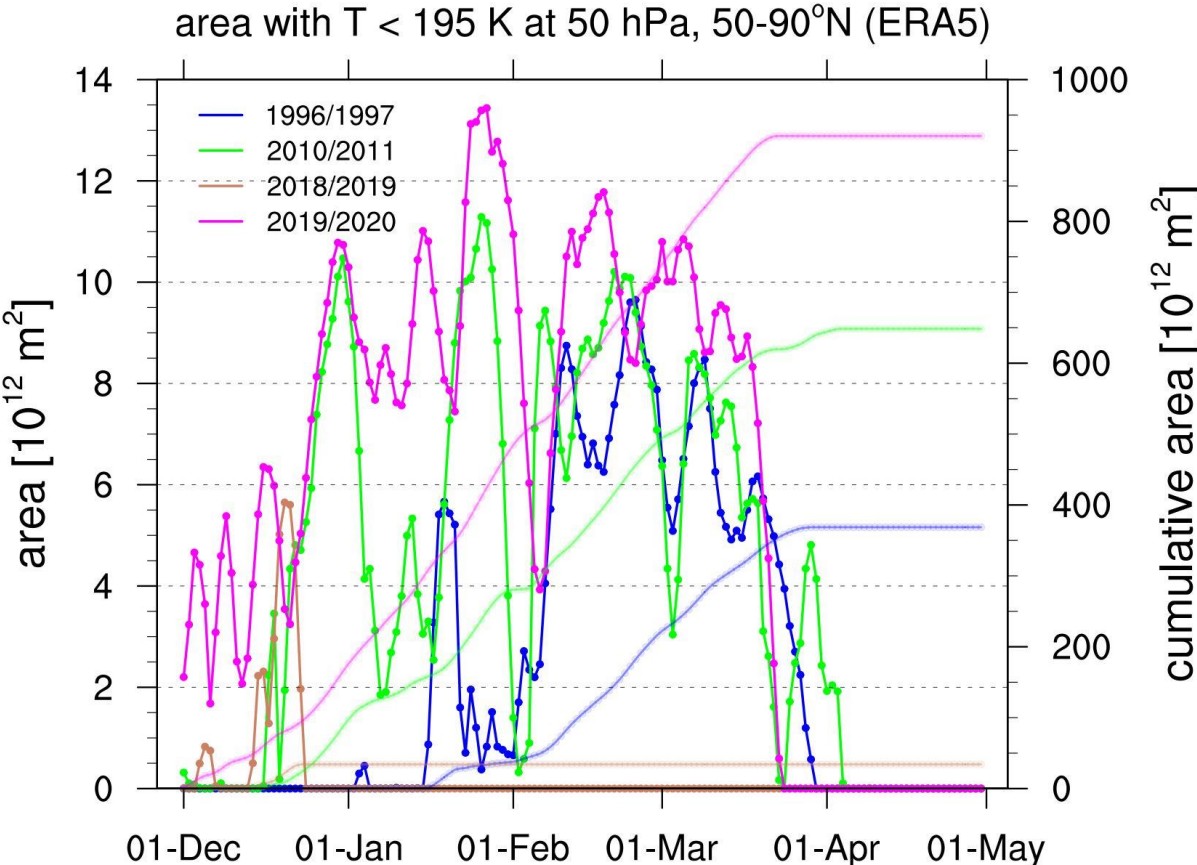

**Figure 7: Daily area (in $10^{12}$ m²) with temperature less than 195 K at 50 hPa (about 20 km altitude) in the region 50°N-90°N from December 1 to April 30 based on ERA5 data (solid lines). Daily cumulative values are indicated as faint lines. The Northern hemisphere winters 1996/1997, 2010/2011, 2018/2019 and 2019/2020 are displayed as blue, green, brown and magenta lines, respectively. For simplicity, the leap day in 2020 (February 29) was neglected in the daily time series.**

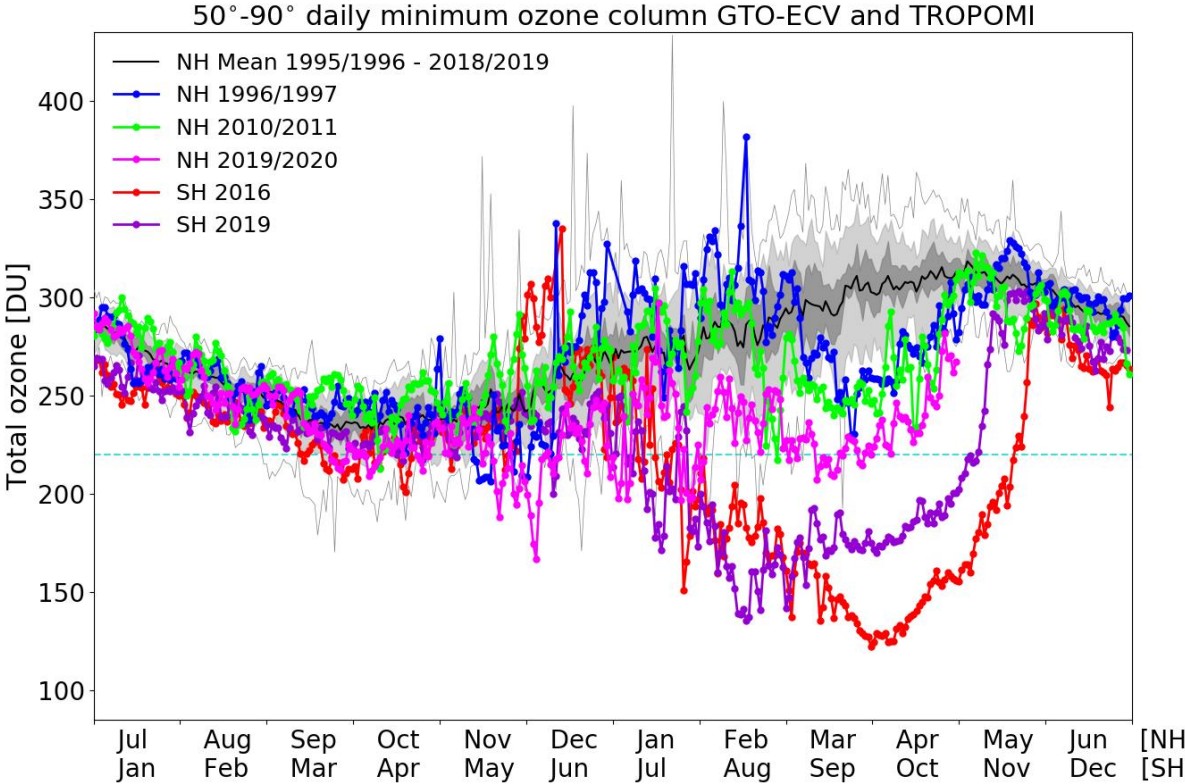

**Figure 8: Annual cycle of the minimum total column ozone values (in Dobson Units, DU) in the Northern polar region between 50°N and 90°N and in the Southern polar region between 50°S and 90°S derived from the European satellite data record GOME-type Total Ozone Essential Climate Variable (GTO-ECV) from July 1995 to June 2019 and TROPOMI data from July 2019 to April 2020. The thick black line shows the GTO-ECV mean annual cycle in the Northern polar region with lowest ozone values in fall season (October, November) and highest ozone values in late spring (April, May). The thin black lines indicate the maximum and minimum values for the complete time period of satellite measurements starting in 1995. The light grey shading denotes the 10th percentile and the 90th percentile, and the dark grey shading denotes the 30th percentile and the 70th percentile, respectively. The magenta line shows the minimum values for the TROPOMI total ozone in the 2019/2020 cycle. The blue and green lines show the minimum values for the total ozone in the years 1996/1997 and 2010/2011, respectively. Note the conspicuous phase of persistent low ozone values below 220 DU (dashed cyan line) in March and April 2020 with new record values for this time of the year. For comparison, the annual cycle of the minimum total column ozone values in the Southern polar region is shown in the years 2016 (red line) and 2019 (purple line). Attention: the respective data for the Southern hemisphere are shifted by six months.**

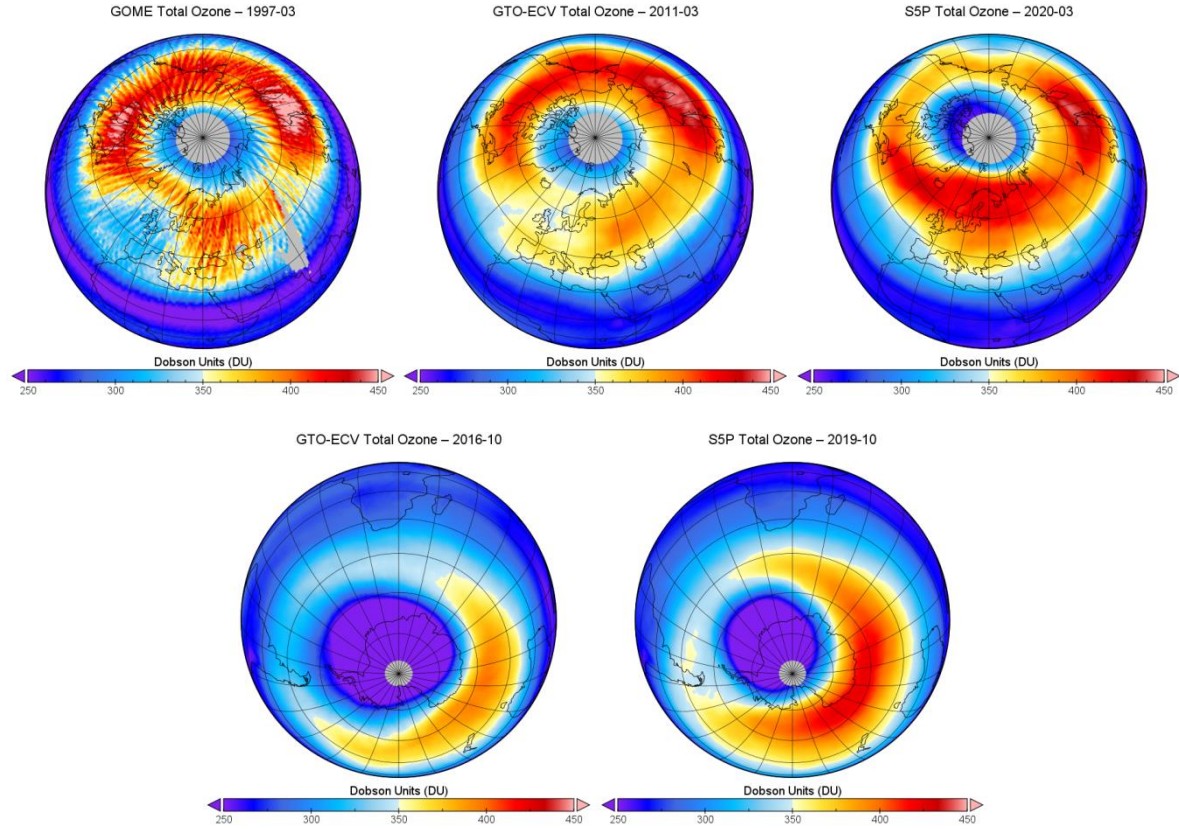

**Figure 9: Monthly mean total ozone columns over the Northern hemisphere in March 1997 (top left), 2011 (top middle), 2020 (top right), and the Southern hemisphere in October 2016 (bottom left), and 2019 (bottom right). The plot for March 1997 is based on GOME/ERS-2 data with a limited spatial sampling, which induces the orbit structures on the monthly mean values; the plots from 2011 and 2016 are based on GTO-ECV; and the plots from 2019 and 2020 are based on TROPOMI/S5P.**