# Peer review of "Evaluation of record low ozone values over the Arctic in boreal spring 2020"

_Atmospheric Chemistry and Physics, 2020_

## Referee Comment (RC1) · Ingo Wohltmann (Referee) · 29 Jul 2020

**Important remark**

There is a joint special issue of JGR and GRL on the 2019/2020 winter. Several of the papers submitted to the special issue show a considerable overlap to your study and are relevant for your manuscript.

These studies have been publicly available as preprints since several weeks (on ESSOAr, the equivalent of the discussions stage of ACP for the AGU journals) and a list of the planned and submitted papers has been publicly available for several months. The first study relevant for your paper (Manney et al.) has already been accepted and

published.

The call for the special issue has been widely circulated in the community. I would have assumed that you are aware of it, but your manuscript does not make this impression. I think it is mandatory that you discuss, cite and acknowledge the relevant papers from the special issue. Some of them are still under review, but will very probably appear before acceptance of your paper.

The list of planned and submitted papers and the current status of the special issue are available at `https://docs.google.com/document/d/1QqGblz_CXZ2eQRtv0-KY4f4XcGjTCIDkhVWLx8e4uTM/edit`

Here are the links to the preprints of the studies relevant for your manuscript:

- Manney et al.: `https://agupubs.onlinelibrary.wiley.com/doi/10.1029/2020GL089063`

- Lawrence et al.: `https://www.essoar.org/doi/10.1002/essoar.10503356.1`

- Wohltmann et al.: `https://www.essoar.org/doi/abs/10.1002/essoar.10503518.1`

- Grooß et al.: `https://www.essoar.org/doi/abs/10.1002/essoar.10503569.1`

I make some suggestions in the specific comments where to cite these papers, but this is certainly not a complete list. Please cite them where necessary.

**General**

In general, I found most parts of the manuscript to be scientifically sound and would recommend publication. While there are some issues (see specific comments), I don't think that there is anything which cannot not be resolved.

I have to admit that I had more of a problem with the wording and readability in some places. On the positive side, I did not find a single typo. But on the other hand, wording was quite awkward in some places, and sometimes the text could have been less confusing and better organized, it felt a little bit rushed in places. I have the impression that asking a native speaker to go through the text would help in many places.

**Major comment**

Unfortunately, however, I have a major comment on a less scientific issue. The general wording and tone of the paper are quite sensationalist in title, abstract and conclusions (or as a colleague who is a native speaker put it: "attention-grabbing"). I don't find the wording appropriate in several places. I don't think that you do yourself or the stratospheric community a favor with that. You need to phrase your manuscript more carefully. In addition, things are sometimes not put into perspective, which may lead the reader to draw the wrong conclusions.

If you imply conclusions here that are at least debatable and at the same time choose a manuscript title that will attract the interest of the public or press, I think this could be problematic. In particular:

- Remove "first" from the title. This is not a contest, but a scientific paper. In addition, it is just not correct. There are papers under review in the upcoming GRL/JGR special issue that have overlap in content to what you write here, and

the first paper has already been accepted.

- I find the repeated and prominent use of the phrase "ozone hole" highly problematic. In the abstract and title alone, it appears 7 times. This raises expectations and may imply conclusions for some readers which are not really backed by the facts or are at least debatable. Given that the phrase "ozone hole" has played a prominent role in the public discussion in the last decades, many people will have a certain understanding of the phrase which sticks in their minds, and we as a community should be careful what we write (in our own interest).

  I feel obliged to go a little bit more into detail why I think that prominently stating that the winter 2019/2020 showed an "ozone hole" is problematic. I think we probably agree that the winter 2019/2020 was exceptional. It was the coldest stratospheric Arctic winter on record and showed the lowest ozone columns and concentrations ever observed in the Arctic, which were comparable to typical values in the Antarctic ozone hole locally and for limited time periods.

  Having said this, it still was far removed from the usual conditions in the Antarctic ozone hole.

  – First, the area with Dobson values below 220 DU was much, much smaller than the typical area of the Antarctic ozone hole. First of all, the area of the Arctic vortex is typically smaller than the area of the Antarctic vortex (Manney et al., 2011, gives a number of 60 % for 2011). Then, the area with values below 220 DU covers only a small part of the vortex. This can easily be seen in your Figure 1. According to the numbers you give in the conclusions, even at maximum, the area was less than 5 % of the Antarctic ozone hole (0.9 million $km^2$ compared to 20 million $km^2$).

    In the end, this is a little bit of a problem with the standard definition of the ozone hole as the area below 220 DU. This definition does not take into account the area covered by the hole at all. But certainly nobody would call

it an ozone hole if the area of the hole would be only one square meter.

– Then, the vertical extent of the layer almost completely depleted in ozone
  in the Antarctic is much larger than the vertical extent of the depleted layer
  seen in 2019/2020 in the Arctic. Usually, ozone is depleted to near zero
  values in a large altitude range from about 350 K to more than 500 K in the
  Antarctic (e.g. Kuttippurath et al., doi:10.1038/s41612-018-0052-6). While
  the ozone profiles of 2020 show a pronounced minimum, very low values
  (below, say, 0.2–0.3 ppm) were restricted to a layer of a few 10 K depth
  around 450 K (see plot of a sonde measurement in Ny-Alesund on 27 March,
  measurement is the blue line).

  This also is a weakness of the 220 DU definition. In the Antarctic, values will
  usually fall far below 220 DU, while they only scratch 220 DU in the Arctic in
  2020 (please see my comment to Figure 7 how to improve on this).

– While the lowest mixing ratios reached in 2019/2020 were comparable to
  mixing ratios that can often be observed in the Antarctic ozone hole (0.1–
  0.2 ppm), they did not reach the near zero minimum values (0.01 ppm) that
  are typical for the Antarctic ozone hole.

– And last but not least, it is also the time period. The Antarctic ozone hole
  lasts several months, while the time period with very low ozone values in
  2019/2020 was at most 5 weeks or so.

Thus, I really would suggest to phrase things more carefully, e.g. to speak of
an "ozone minimum" or "values comparable to values observed in the Antarctic
ozone hole" and so on. Please see the specific comments for the places where I
think you have to phrase things more carefully.

• Sometimes, it seems that you would like to push the reader into a certain direc-
  tion by omitting information in strategical places. While this means that you don't
  write anything scientifically wrong formally, you may push the reader to draw conclusions that are not correct. In particular, it would have been very easy to include Antarctic data in figures like Figure 1, 5, 6 and 7, and to discuss this data in the text to put things into perspective. I think it is really mandatory that you change the manuscript to be more balanced and to put things better into perspective.

**Specific comments**

- Page 1, lines 11–12: Please rephrase to something like "record low ozone column" or similar and avoid the term "ozone hole".

- Page 1, lines 12–14: The sentence would work equally well when you omit "A persistent ozone hole pattern". Just start with: "Minimum total ozone column values...". To make the "for the first time" in the sentence work better, maybe you could write "for more than 5 weeks" (or 4 weeks?) instead of "about 5 weeks".

- Page 1, line 14 (next sentence): Suggestion: "Usually such low total ozone column values have only been observed in the ozone hole in the polar Southern hemisphere (Antarctic) in spring over the last 4 decades, but not over the Arctic." Slight change in text, but larger change in meaning. But please state here in addition that column values will go far below 220 DU in the Antarctic to put things into perspective. It would also make sense to state the other differences which I have outlined in my general comment here (smaller area, vertical extent and time period).

- Page 1, line 16: Change to "The record low values were caused..."

- Page 1, line 16: A stable vortex does not enable a cold stratosphere. This confuses cause and effect, when there is no wave activity. When there is wave activity, it is a little bit more complicated, please see my comment to page 5, line 3 (sorry, wrote that comment first...). Please phrase that correctly.

- Page 1, line 20: "in the context of" is probably better.

- Page 1, line 20: Replace "ozone-hole like features" simply by "cold winters"

- Page 1, lines 27–28: I would delete this sentence. This is exactly what I would call "attention-grabbing", but it doesn't really transport information.

- Introduction: In a later comment (page 5, line 3–30), I suggest to add a short overview on how ozone depletion works somewhere in the introduction (PSCs, cold temperatures, chlorine activation, return of sunlight, . . . ) to be able to streamline the text in the later sections a bit.

- Introduction: In some places, I find the references a bit odd, while I miss others. E.g. Langematz, 2019 and Loyola et al., 2009, would not be the first ones that come to my mind. I would expect the review paper of Susan Solomon from 1999 somewhere (Rev. Geophys. 37, 275–316, 1999). A paper that is mandatory to cite in a study like this is in my opinion Solomon et al., "Fundamental differences between Arctic and Antarctic ozone depletion", 2014, doi:10.1073/pnas.1319307111 (it is by chance that it is Susan again). This is a review paper about exactly the topic you are talking about here, and it also contains some very critical remarks about using the term "ozone hole" for the Arctic.

- Page 2, lines 5–10: You don't need to explain the meaning of the term "ozone column". This is basic textbook knowledge.

- Page 2, line 7: Delete "so-called"

- Page 2, line 11: You need a citation for the 220 DU threshold (it is for example defined in the WMO report 2018, along with further references). This is more or less an "official" definition which most people agree on, and you will need some references for that. All of your following discussion depends on this definition.

- Page 2, line 14: This is misleading. Changes of ozone column by transport and changes of the column by chemistry are correlated (e.g. Tegtmeier et al., 2008, which you may want to cite here). A more dynamically active winter means both more transport of ozone into the vortex and less ozone depletion because of higher temperatures. Your sentence reads as if the difference between 450 DU and 220 DU would mainly be caused by chemistry. In fact, one of the fundamental differences between the Arctic and the Antarctic is that transport plays a large role in determining the Arctic ozone columns.

- Page 3, line 3: Perhaps replace "with respect to low TOC" by "in reaching low ozone columns"?

- Page 3, line 4–5: Some references from the paragraph starting page 10, line 27 would also fit, and see also my comment to this paragraph for even more references.

- Page 3, line 11: It would probably be good to mention the official databases for ozone sondes here, i.e. WOUDC and NDACC.

- Page 3, line 13: You need to phrase that more carefully: "which led for the first time to ozone values below 220 DU in larger parts of the vortex for an extended time period".

- Page 3, line 25: As far as I know, the nominal resolution of ERA5 is 0.28125 degrees. It does not really make sense to sample the data at a higher resolution (but does not hurt either).

- Page 3, line 24–28: You don't need to go into detail how you do a daily average. I trust you that you are able to do this correctly :-) In fact, you can replace anything between "For our investigations..." and the end of line 28 by "We use daily and monthly averages in the following". This is totally sufficient.

- Page 4, line 3 and line 8: Delete "first". This is really not relevant in the context of this paper. Again, this is no contest.

- Why is Figure 2 the first plot that you discuss in the paper? You should change the order of the plots, so that Figure 2 becomes Figure 1.

- Figure 2 and accompanying discussion: Lawrence et al. from the special issue contains similar figures and discussion. Please cite and discuss Lawrence et al. here.

- Page 4, line 25 and Figure 2: At first, I was a little bit confused by the plot because I didn't realize that the dots are monthly mean values and the lines are daily values, causing the colored dots not to lie exactly on the lines. Maybe there is some information overkill in the plot. One could replace the dots by a grey area showing the range of the daily values in all years for every day.

- Page 4, lines 26–28: I would suggest to delete the part in parentheses. Either discuss this explicitly in the paper, or leave it at "caused by planetary wave activity".

- Page 4, lines 28: Would be good to use the established terminology here (major warming, minor warming, sudden stratospheric warming...)

- Page 4, lines 29: The wording is a little bit awkward and hard to understand. I would try wording like "stable and undisturbed vortex", "circular shape", "not displaced from the pole" etc. "deteriorated" is not the correct word.

- Page 5, line 2: Figure 3 does not add any relevant information which is not contained in Figure 2. You could delete this figure without loss of information.

- Page 5, line 3: You confuse cause and effect here. First, in the absence of wave activity, the polar region gets colder than mid-latitudes in winter due to a lack of

sunlight and because of radiative cooling. Then, a pressure difference develops compared to mid-latitudes, which causes a geostrophic wind as response.

For a correct discussion, you would need to explain the mechanisms of the Brewer-Dobson circulation in more detail. Interannual differences in polar temperatures or vortex strength are caused by differences in momentum deposition by breaking waves (mainly in the mid-latitude stratosphere) which drive the BDC.

That means that both temperatures and vortex strength are correlated, but that this has a common underlying cause, and not that one causes the other.

- Figure 5 and 6 and accompanying discussion: Lawrence et al. and Wohltmann et al. from the special issue contain similar figures and discussion (e.g. Figure 1 Wohltmann et al. and Figure 11 Lawrence et al.). Please cite and discuss Lawrence et al. and Wohltmann et al.

- Figure 5 and 6: Please add typical Antarctic values in the figures and discuss them in the text. This will help very much to put things into perspective and I think this is mandatory.

- Figure 5: It is a little bit confusing that you use a polar cap area and not the vortex area here as the area where you look for the minimum. That only gives the desired result because of the comparatively high temperatures outside of the vortex. You are interested in the minimum temperatures inside the vortex here, because these are relevant for ozone depletion. It would be more consistent to base the plot on the vortex area (the plot would look almost identical probably).

- Figure 5: Can you really learn something from the minima of the monthly mean values? These will by definition always be higher than the daily minima. I don't really see that they provide any insight. I would suggest to remove them and to replace them by some grey area showing the range of the daily values over all years.

- Page 5, lines 11–12: The numbers for the vortex area are a little bit unintuitive. At least I don't have a really good judgement of them. An alternative would be to divide the values by the vortex area, which gives a value in percent, which is more easy to grasp. In addition, the unit for the cumulative area can't be correct. There must be some time unit missing (probably "days").

- Page 5, line 14: The citation seems odd. An obvious citation would have been the original study of Solomon et al., Nature, 321, 755, 1986. Or the WMO report or the 1999 Solomon review paper.

- Page 5, line 16–20: Discussion on "ozone hole". Please phrase that more carefully. You could state here that record low values have been reached and that their temporal extent and the covered area were unrivalled in other years (but please check that, this is just what I suppose is correct).

  It is certainly also ok to mention the 220 DU definition of the Antarctic ozone hole here, but I think it is mandatory here to discuss the smaller area, the limited vertical extent and the limited time period compared to the Antarctic ozone hole to put things into perspective.

- Page 5, line 20: As far as I can see, this is the first time you mentiom Figure 1. Please correct the order of the figures.

- Figure 1: Add contours for the 220 DU contour and the vortex edge. These are things that are really hard to see in a coloured contour plot. And the 220 DU contour is really central for the discussion in your paper.

- Page 5, paragraph lines 3–30: The readability of this paragraph suffers because of two issues in my opinion:

  First, you introduce 5 figures in this short paragraph, but only by mentioning them in parentheses. It would help immensely to insert a few sentences starting with "Figure xxx shows. . . " in this paragraph.

In addition, you try to explain how ozone depletion works in some half-sentences and in parentheses here and introduce things like Cl activation and NAT clouds. I think you could improve your manuscript a lot by adding a short paragraph in the introduction explaining the basics of ozone depletion in a few sentences (cold temperatures, PSCs, return of sunlight, chlorine activation, CFCs, ...). Then, you could refer to that later, and the text would read much more fluently.

You have tried to explain even more basic things in this manuscript, like the definition of ozone column or the averaging, so it seems odd that you don't explain some things in the introduction which would really help to streamline the text in your paper.

- Figure 6: To make things more consistent, you could show the range of values of all other years in grey, as in Figure 2, 5 and 7.

- Page 6, line 9: Why is this expected from January on, and why is this expected with values above 220 DU? I don't understand your reasoning here. Maybe it would be better to simple write "is observed"?

- Page 6, line 12: See comment page 3, line 11.

- Page 6, line 21: Please compare the area below 220 DU to the area of the vortex here. This would really be important and interesting information which helps to put things into perspective.

- Page 6, line 22: Again, the units are not quite correct (at least formally). A unit of "days" is missing.

- Figure 7 (and accompanying text): What really would add a lot of value to your paper with relatively small effort would be to show typical values from the Antarctic in the same plot and to discuss the differences between Arctic and Antarctic in the text. Please add Antarctic values to the figure to put things into perspective.

- Figure 7: The same comment as for Figure 5 applies: It would be better to base the figure on the vortex area and not on a polar cap area.

- Page 6, line 27: There is a lot of literature on the 2010/2011 winter. Maybe it wouldn't hurt to cite one or two studies more here. Suggestions: Sinnhuber et al., 2011, doi:10.1029/2011GL049784, Hommel et al., 2014, doi:10.5194/acp-14-3247-2014, Strahan et al., 2013, doi:10.1002/jgrd.50181, Kuttippurath et al., 2012, doi:10.5194/acp-12-7073-2012, . . .

- Table 1: I could live well without the table. The information can already be found in the plots.

- Page 8, line 9: The wording is awkward. Suggestion: "The winter 2019/2020 showed a larger volume below the formation temperature of PSCs than other winters for an extended period of time."

- Page 8, line 11–15: This is visible in the HNO3 measurements of MLS. Please cite the Manney et al. paper from the special issue here. You don't need to speculate.

- Page 9, line 12–14: Can you get a little bit more quantitative here? What is the quantity you are looking at here? Vortex mean temperatures at some level? In the moment, these sentences do not convey enough information to be useful (I am well aware that the uncertainties are large and will only allow a qualitative statement).

- Page 9, line 25–32: And of course, there is the ozone hole split of 2002.

- Page 10, paragraph 1–6: The statements in this paragraph are problematic and don't really tell the truth because you omit information. You don't mention that the temperatures in the Antarctic are considerably lower than in the Arctic (even for 2019/2020) and that the period of low temperatures is much longer in the

Antarctic. If you would have added this information to your plots (e.g. Figure 5 and 6), this would be quite obvious.

It is also not true that this would result in ozone depletion rates that are comparatively strong. The depletion rates will also depend on the amount of ClOx, which in turn e.g. depends on the amount of descent in the vortex. In fact, this is a rather complicated topic. The crucial difference between the Arctic and the Antarctic which leads to very low ozone values is the much longer time period with ozone depletion. Since ozone loss is usually in saturation in the southern hemisphere, some details on the path to zero ozone don't really matter here.

- Page 10, line 10: Please give numbers for the chlorine content. This would show that the differences are not that large (I assume 10 % to 20 % difference?) and that it is not too surprising that differences in temperature are the main driving factor. But in principle, you are right, this is worth mentioning. Maybe some additional discussion along the lines above is appropriate.

- Page 10, line 17: In fact, a dehydration event can clearly be seen in the MLS H2O measurements. Please cite Manney et al. from the special issue here.

- Page 10, lines 18–22: You repeat what you already have said on page 8. Please delete. See also the comment to page 8, lines 11–15.

- Page 10, line 24: See comment page 10, line 10.

- Page 10, line 27. Please rephrase "atypical ozone hole" to something like "record low ozone values".

- Page 10, line 29 and line 32: I totally agree that you never can tell climate change from a single year and that you have to look how this evolves in the future in the context of a longer time series. On the other hand, you can have a look into the winters observed so far and don't need to wait for the future to have a

long timeseries. And the year 2019/2020 does add information to this timeseries, since it was the coldest Arctic winter observed so far and the coldest winters have become colder in the last years quite consistently, at least according to some metrics and meteorological data sets. You could cite Wohltmann et al., Figure 1, from the special issue for a figure illustrating this quite well. But in general, I agree with you.

- Paragraph Page 10, line 27 to page 11, line 10: Some additional recent relevant studies on this topic: Ivy et al., 2016, doi:10.1175/jcli-d-15-0503.1, Rieder and Polvani, 2013, doi:10.1002/grl.50835, Butchart et al., 2010, doi:10.1175/2010JCLI3404.1, Bednarz et al., 2016, doi:10.5194/acp-16-12159-2016. These contain a lot of interesting discussion on this somewhat controversial topic (Are coldest winters getting colder etc.), which you may want to add here.

- Page 11, line 2: I think an important point to mention here is that the other important driver of changes in stratospheric temperature (apart from changes in radiative cooling by greenhouse gases) are changes in the strength of the Brewer-Dobson circulation (e.g. Langematz et al., 2014 and many more), and the BDC in turn is affected by climate change.

- Page 11, line 11–14: This paragraph seems a little bit out of context.

- Page 11, line 16–17: The sentence seems overly complicated and wording is a little bit awkward. Suggestion: "This study presents a description of the Northern winter and spring season 2019/2020 considering the. . . " ("regarding" seems not to be the correct English word to me, I think "considering" is what you mean).

- Page 11, line 17–19, 23–26: Please phrase more carefully. It is ok to discuss that the observations were below the 220 DU threshold for a longer period of time, but please also discuss the differences to the Antarctic ozone hole in area, vertical

extent and duration that I have mentioned in my general comment right after lines 17–19. I am very happy that you finally discuss some of this in lines 23–26, but please move it a few lines up and discuss a little more. I.e. put things into perspective. Please don't call it an "ozone hole". And don't claim that this year was the first occurence of an "ozone hole". This was already claimed by some people in 2011, and it didn't help in the discussion back then, too. In general, I don't find this chase for superlatives very helpful, and it does not help to advance our scientific understanding.

- Page 11, line 23–26: I am very happy that you finally discuss this here, but I hope you agree that it would have been necessary to mention this much much earlier (and more often) in your manuscript.

**Technical corrections**

- Page 3, line 18: You can delete "It must be noted that".

- Page 6, line 16: I don't think "outstanding" is the right word here. Maybe "prominent" or "remarkable"?

- Page 8, line 1 and 3: I would write "ice PSC" and not "ICE-PSC" (ice is not an abbreviation).

- Page 8, line 28: "Discussion" and not "Discussions"

- Page 9, line 17: "recall" is not the right word. Suggestion: "We note"

- Page 11, line 19: You very probably mean "compared" and not "confronted"?

- Page 11, line 20: I would say "which show" and not "which are showing"

- Page 11, line 20: I would say "the most recent datasets"

[Figure]

[Figure]

**na20200327**

Fig. 1. Sonde measurement in Ny-Alesund on 27 March, measurement is the blue line

---

## Short Comment (SC1) · 30 Jul 2020

Short reply to Ingo Wohltmann (reviewer)

Dear referee, dear Ingo,

Thank you very much for your detailed review and the specific comments regarding our manuscript. Your statements and suggestions are highly appreciated and will definitely help to improve our manuscript, especially with respect to the communication of our scientific results. We will consider them in the revised version of the paper. A detailed response to your comments will be submitted with the revised version. For now, we mainly reply to your "Important remark":

Our manuscript was prepared in May for submission to ACP and in the meantime we

recognized the announcement of the JGR/GRL special issue on the winter 2019/2020. Since ACP is our favored scientific journal we decided nevertheless to submit the manuscript to ACP on June 9, 2020. Due to a major misunderstanding this draft of the paper was not accepted for ACPD. After a longer period of clarification (due to out of office periods and various delayed responses) regarding the misunderstanding, the paper was resubmitted with consent of the Editor on July 21, 2020. Unfortunately, despite our efforts and the clarification of the misunderstanding, we were told that it is not possible to keep the first registration number and hence the submission date shows July 21, 2020. None of the listed manuscripts (in the beginning of your report) have been accepted before June 09, 2020 and only one of the papers (Manney et al.) was available on this day. In fact, if I am correct, the Manney et al. study was first published online on this very date. Because of this, we have not been able to refer to the papers of the JGR/GRL special issue during the time of writing our paper.

In the meantime we also have seen these manuscripts in the special issue of JGR/GRL and we will definitely refer to these papers in the revised version of our paper when appropriate. We think that our paper shows interesting points based on our analyses (in particular using the TROPOMI data), which are in line with the results presented in the mentioned papers, e.g. by Manney et al. or by yourself.

In summary: (1) we will consider the points of your detailed review, (2) we will refer to the (accepted) papers of the JGR/GRL special issue, and (3) we will set the results of these papers into context with our findings in a revised version of our manuscript.

Best regards, Martin

---

## Referee Comment (RC2) · Anonymous Referee #2 · 20 Aug 2020

Dameris et al. present a documentation and evaluation of the evolution of observed total column ozone during the winter 2019/20, contrasting it with two other winters with anomalously cold, stable polar vortices, as well as climatological Arctic column ozone. This study makes an important contribution to the study of Arctic ozone, and it is timely given it evaluates the most recent winter. I found the manuscript to be well written, the analysis clear and the figures well presented. I feel the paper fits the scope of ACP and would recommend publication of the manuscript after the authors address the comments below.

General comments:

I wonder if it is fair to refer to the Artic winter 2019/20 as an 'ozone hole'. While the authors are clearly correct in stating that total column ozone falls below the 220 DU

[Figure]

threshold, typically used to define the edge of the ozone hole, it should be remembered that in the Antarctic column ozone values typically fall far below 220 DU, and for a timescale measured in months. Can the authors use further common metrics (ozone mass deficit, minimum column ozone, etc) in their evaluation to give a better understanding of the column ozone evolution? Additionally, I feel it would be beneficial if the authors included data from the Antarctic in their timeseries plots, so the reader can get an impression of the how the Arctic winter 2019/20 compares to what is more generally considered an ozone hole. The authors state that these low ozone values cover a large area (0.9 million km2), but that is a tiny fraction of the area covered by the Antarctic vortex. I feel that either the authors should refrain from using the term 'ozone hole' or to place this term into context by comparing it with the Antarctic ozone hole and state explicitly that it is much smaller and shorter lived than the

Further, a lot of emphasis is placed on the idea that the winter 2019/10 was the first instance of column ozone falling below the 220 DU threshold. However, the authors' Figure 7 shows that there are repeated instances of column ozone below 220 DU in the thin black line. While the authors refer to these as mini-holes, and explain the role in dynamics in their formation, I feel a distinction should be made between these and the 2019/20 winter – is it fair to say that this winter constitutes and ozone hole because these events are longer lived? While the winter 2019/20 is certainly atypical, it is wrong, based on this figure, to say, as the authors do on P11L17-18, that it is the first time these values have been observed. And if the qualify here is that they occur over a 'large area', does a new definition for an ozone hole need to include some measure of the areal extent?

I miss in the introduction some general information on the processes involved in polar ozone depletion. While these processes are mentioned later in the manuscript, a paragraph in the introduction detailing the polar vortex, cold polar lower stratospheric temperatures, PSC formation, heterogeneous chemistry, and subsequent catalytic ozone depletion upon return of sunlight to the polar vortex would significantly aid the reader.

Additionally, I would like to see more information on how the Arctic and Antarctic differ: increased wave activity in the Arctic, the fact that the Artic vortex is often displaced from the pole, which can affect the amount of sunlight that can reach the vortex, the relative importance of chemical depletion vs transport.

The authors focus on the large-scale meteorological conditions within the winter 2019/20 Arctic polar vortex, particularly the area below the 195K threshold as a metric for PSC occurrence and chlorine activation. Can they say anything about local conditions, particularly for example the role of orographic gravity waves during the winter of 2019/20 and the impacts of these on local temperatures?

The Harris et al. (2010) paper cited in the manuscript highlights linearity between PSC occurrence and ozone depletion. Similarly, Hommel et al. (2014: Chemical ozone loss and ozone mini-hole event during the Arctic winter 2010/2011 as observed by SCIAMACHY and GOME-2) highlight linearity between total column ozone change at 100 hPa eddy heat flux. Are the authors able to say something about if the winter 2019/20 falls on these linear relationships identified in past studies? Or does this extreme winter violate the relationships identified in other studies?

Some key references are missing from the manuscript, with many instances of only one, recent citation given during key discussion. I would encourage the authors to expand upon the literature already cited in the manuscript.

Specific comments:

P2L19: 'Nevertheless, the current atmospheric content of CFCs is still enhanced. . .'. It would be beneficial to explicitly state a date here, i.e. '...still enhanced with respect to 1980s values. . .'

P2L22: Care should be taken when using a term such as full recovery. While several studies show that column ozone is projected to return to 1980s values by the middle of the century, is that really full recovery? Some of this signal is driven by stratospheric

cooling resulting from increased CO2 mixing ratios, and is separate to recovery driven by reduction in ODSs. I would prefer the authors say something about ozone return to historic values, which is an important part of the recovery story, rather than 'full recovery'.

P5L3: Is 'strong cooling' correct here, or are the cold temperatures a result of reduced warming? Can the authors say anything about the radiative and dynamical processes operating within the polar lower stratosphere? This thought is also applicable to P7L7.

P5L21: The analysis here focuses on column ozone values north of 50°N. However, Figure 1 of the manuscript shows that the Arctic vortex is not symmetrical about the pole, and so this average includes considerable amounts of column ozone from outside the Arctic vortex. Is it possible to plot vortex averaged column ozone instead, and so separate out the low values from inside the vortex from the high values outside?

P6L22: 'The daily accumulated ozone hole area in March and April was estimated with 4 million km2' – how does this value compare to that for September and October of a typical year in the Antarctic? I suspect the Antarctic value is many times larger. If so, is this a useful metric – I feel it may be misleading if not placed into context.

P10L1-6: Care should be taken here attributing all of the low column ozone values to chemical depletion. The authors discuss the importance of dynamics in the preceding paragraphs in preconditioning the polar vortex, but the phase 'ozone depletion rates' to me describes ozone loss through catalytic reactions, whereas in actuality the low column ozone is driven in part by chemistry and in part by reduced transport of ozone to the polar cap. This is obvious from your Figure 7, as column ozone increases from December to May, and this is not driven by chemistry.

Technical:

P2L26: Check use of 'Exemplarily'

P7L22: replace 'cumulated' with 'cumulative' – also other instances throughout the

manuscript.

P11L19: remove 'a' from 'about a five weeks'

The x-axis label for all timeseries plots says 'time [days]', which I would expect to be a set of numbers, but the plot shows date on the x-axis. Please revise.

———————————————

---

## Short Comment (SC2) · 20 Aug 2020

**On the use of the phrase "Arctic ozone hole"**

In this manuscript, the phrase "Arctic ozone hole" is used many times. Even though one reviewer Ingo Wohltmann already pointed out that this is problematic, we would like to come back to this point.

There is no rigorous scientific definition of an "ozone hole". Commonly, the ozone hole size is used as the area where the total ozone colum is below 220 Dobson units (DU). This value was chosen because it is lower than values reached in the Antarctic prior to the appearance in the early 1980s (when chemical ozone destruction via man-

made compounds was first large enough that it dominated the evolution of ozone in the Antarctic spring) of what is now commonly called the Antarctic ozone hole, and because it does tend to approximately follow the vortex edge in the Antarctic (Newman et al., 2004). It is not at all clear, given very different conditions and background ozone levels, that these considerations would be appropriate for the Arctic.

It is true that for the first time, Arctic ozone columns were depleted to below 220 DU. In the manuscript, an "ozone hole area" of 0.9 Mio km$^2$ is reported. This is well below the Antarctic ozone hole areas of 20-30 Mio km$^2$. Ingo Wohltmann has nicely summarized the arguments that do not need to be repeated here.

A specific meaningless concept is the "daily accumulated ozone hole area". If we did get this correctly, it would be the sum of daily "ozone hole areas". If this concept was used for the Antarctic ozone hole, it would reveal an area of about 3 times the total surface of the earth.

Beyond the scientific arguments, there is also the responsibility of science not to transport sensation but to convey correct understandable information to the general public. Many people without scientific background in this area would see only the phrase and draw the wrong conclusion that in the conditions in the Northern hemisphere are comparable to those in the Antarctic ozone hole. We have seen that in press articles already. We therefore request that the authors reconsider their choice to call the severe Arctic ozone depletion in 2020 an "Arctic ozone hole" and revise the paper accordingly.

**Jens-Uwe Grooß and Gloria Manney**

**Reference**

Newman, P. A., Kawa, S. R., and Nash, E. R.: On the size of the Antarctic ozone hole, Geophys. Res. Lett., 31, L21104, 10.1029/2004GL020596, 2004.

---

## Short Comment (SC3) · 23 Aug 2020

**Short (well, rather long actually!) comment on "First description and classification of the ozone hole over the Arctic in boreal spring 2020" by Dameris et al.**

Gloria L Manney, manney@nwra.com

Other comments have already discussed the overly-casual and ill-defined use of the term "Arctic ozone hole" and I believe that subject has been covered well already (though my primary scientific comment below will touch on it in the context of comparison of specific winters).

Other comments on / reviews of this paper have also already discussed the claim of "First" and the lack of citations of other papers in review and published on this exceptional winter, and I agree overall with their remarks. I do have additional "philosophical" comments on this subject:

I am happy to see numerous papers submitted on the exceptional 2019/2020 stratospheric winter, and made publicly available as preprints, regardless of what journal they are published in (the special issue is simply a vehicle to encourage and facilitate publication of as comprehensive work as possible on this winter) -- this includes, in particular, seeing the full range of datasets and models that are available studied, which points to what I see (and the authors mention in their brief reply to Ingo's review) as a strength of this paper, namely, the use of TROPOMI data along with the GOME-type Total Ozone Essential Climate Variable (GTO-ECV) to describe the evolution of total ozone in relation to previous winters. I believe we as a scientific community will learn the most (always our ultimate goal) about this winter if we all study and cite each other's work, and discuss / collaborate with each other in analysis and modeling of what we see in the observations. The move towards open access to preprints of submitted papers -- strongly supported by both AGU (in making preprints available on ESSOAr immediately after submission) and EGU (in journals with open review such as ACP/ACPD) -- allows us to do this much more effectively than in the past. Thus:

(1) As one with possibly a strong claim to leading the "first" peer-reviewed paper on Arctic ozone loss during 2019/2020 (the Manney et al GRL paper was submitted in late May, available on ESSOAr by early June, first published in GRL online on 17 July), I firmly believe that **none** of us should be claiming to be first when all of us have been working since it became apparent that this past Arctic winter was going to be exceptional to produce and disseminate the best science describing / explaining the observations!

(2) In this collaborative spirit, I hope that not only will the authors of this paper cite other submitted papers that are openly available wherever it is appropriate to support or show consistency with the material in this paper, regardless of whether they have received final acceptance or not, but also that those papers (e.g., for the GRL/JGR special issue) that are still at a stage prior to having completed their final revision will in turn cite this preprint wherever it is appropriate to support or show consistency with their results.

These are, as I said above, philosophical rather than scientific views, so I can only ask the authors (of this preprint and others!) to ponder them and make revisions according to their judgement of the merit of these points.

**Major Scientific Comments:**

The biggest issue that has not been raised in other comments / reviews at the time I'm writing this, and that I believe must be addressed before peer-reviewed publication, is the comparisons of 2019/2020 with 2010/2011 and 1996/1997, and the failure to communicate the very large differences in polar processing and ozone loss in 1996/1997 compared to the other winters studied.  In the context of comparing superficially similar springtime lower stratospheric vortex conditions in 1996/1997 and 2010/2011, the very large differences in polar chemical processing in those two winters have been extensively highlighted, first in detailed discussion in the supplementary information (SI) of Manney et al. (2011, Nature), and in numerous later publications culminating in a detailed summary / synthesis in the WMO 2014 Scientific Assessment of Ozone Depletion (section 3.2.3.3), which provides further references.   In short, for numerous reasons (very late lower stratospheric, LS, vortex development and late drop of temperatures below PSC thresholds, smaller altitude region of low temperatures, weaker LS vortex throughout the winter, little/no denitrification, etc), chemical ozone loss was much less in 1997 than in 2011 (and hence than in 2020, which saw as much or more chemical loss as in 2011, eg, Manney et al, 2020; Wohltmann, et al, 2020; Grooß  and Müller, 2020).  Moreover, dynamical conditions led to frequent ozone mini-holes (e.g., Coy et al., 1997) and higher tropopause altitudes (e.g., Manney et al, 2011, Nature, SI) in spring 1997 that contributed to lower column ozone via dynamical processes than followed other winters with comparable chemical ozone loss.  This is an important distinction that it is essential to address for the comparisons in this manuscript to provide accurate information on the similarities (a few) and differences (many) between 1997 and the other two winters considered.  Statements such as (to pick only one example, page 10, lines 8-9) "...all three years showed particularly strong ozone depletion..." are scientifically inaccurate.  This also folds in with the inadvisability of lightly using the term "Arctic ozone hole", as 1997 is a classic case of a situation that looked superficially similar to the Arctic winters, 2011 and 2020, with the most chemical ozone loss and in some ways "Antarctic-like" conditions (see WMO 2014, Section 3.2.3.2; Manney et al, 2020; Wohltmann et al, 2020), but which in fact had chemical processing that was in no way comparable to that in the Antarctic.

I also have concerns with the description of the dynamical conditions in relation to previous winters.  The dynamical situation is described using only 10hPa zonal mean winds and 50hPa temperatures.
Zonal mean winds in the middle stratosphere (10hPa as opposed to the levels around 50hPa where chemical processing maximizes) are virtually irrelevant to the state of the lower stratospheric vortex, because:
(1) Vortex strength, size, and geometry vary strongly with altitude in different ways in different winters -- we have seen winters (such as 2010/2011) where the vortex was exceptionally strong in the lower stratosphere but not in the middle stratosphere, and winters (such as 1997) where the vortex was for much of the winter fairly typical in the middle stratosphere but exceptionally weak in the lower stratosphere.

(2) The Arctic vortex is rarely close to symmetric or pole-centered, even in the coldest and/or most dynamically quiescent winters (see, e.g., Figure 1 in Manney et al, 2020, or any of numerous other publications in the past ~20 years), and its size, shape and position vary dramatically both intraseasonally and interannually.  Thus, zonal means, even were they examined at altitudes in the range where LS polar processing occurs rather than at 10hPa, provide very little information on characteristics of the polar vortex such as size, location, and strength.

Similarly, 50hPa minimum temperatures north of 50N and area of T < 195K at 50hPa, while very relevant to polar chemical processing, are by themselves inadequate to characterize the potential for chemical ozone loss in the LS vortex because:

(1) The vertical structure/location/extent of the region with temperatures conducive to PSCs varies strongly interannually and within seasons; this is one of the reasons why one of the most useful measures of polar processing / ozone loss potential (both day-to-day and as a measure of total ozone loss potential in a given winter) is V_psc, the area below the PSC or chlorine activation threshold integrated over all lower stratospheric levels.

(2) Because the LS vortex varies strongly in size, shape, and position, while the high-latitude minimum associated with the polar vortex is usually north of 50N in dynamically quiet winters, this may not always be the case, and is certainly not always the case in winters with strong SSWs during the cold period (Dec--Feb).

Furthermore, in relation to column ozone and its relationship to the LS vortex and low temperatures, because low column ozone is strongly spatially correlated with low LS temperatures by dynamical processes (see, e.g., discussion and references in SI of Manney et al, 2011) and the region of low temperatures in the LS is often not well-correlated with the lower stratospheric vortex (see, e.g., Manney et al, 1996, GRL; Mann et al, 2002, JGR; SI of Manney et al, 2011; Lawrence et al, 2015, ACP; and references in those latter two works), in absence of strong chemical depletion, the shape / extent of the region of low column ozone is not expected to be correlated with the shape / extent of the polar vortex.  Therefore, in the maps of column ozone in Figure 8, we cannot judge whether the morphology of the low region is consistent with strong ozone loss unless we know how it relates to the morphology of the LS vortex.  E.g., one of the most commonly used metrics for this is a contour or contours of potential vorticity (PV) on an isentropic surface somewhere in the LS (somewhere between about 450 and 550K is typical, commensurate with the approximate levels where ozone contributes most to the column), with value(s) such that it is (they are) in the region of strong PV gradients bounding the vortex. (Similarly, the strength of PV gradients along the vortex edge is a common and valuable metric of vortex strength -- while maximum windspeeds at an appropriate level would also be informative of vortex strength, zonal mean winds are not.) PV on isentropic surfaces is readily available in all modern reanalyses including the ERA5 reanalysis used herein.

Please note that Lawrence et al (2020; as I write this in late August, nearing completion of minor revisions for JGR, and available on ESSOAr since mid-June) in their section 3.4 (submitted version) provide a detailed discussion of LS vortex strength and chemical processing potential in 2019/2020 in comparison with the record from 1979 through 2019 and in particular compare with 1997, 2011, and 2016 (2016 is of interest because, while low temperatures and chemical

processing ended much earlier, it still is the record cold winter in January and February, and had overall greater polar processing potential and more chemical loss than that in 1997; e.g., Manney and Lawrence, 2016, ACP; WMO 2019; and references therein), using widely accepted diagnostics / methods with uncertainties quantified.  Because this paper for the AGU Special Collection provides an overview of the dynamical conditions during the winter, it is designed to provide this "foundational" material in a thorough way so that other papers on this winter can start with that comprehensive description as background.   [While not as relevant to this paper and the 2019/2020 ozone loss, Lawrence et al (2020) also describe thoroughly in their Section 3.1 the evolution of 10hPa zonal mean zonal winds in 2019/2020 versus climatology (their Figure 1) and in the context of the time series since 1959 (their Figure 2).]
[The points above for which I have not provided references are well documented, e.g., in the past ~4 WMO Assessments and references on Arctic polar ozone and ozone loss therein.  The relevant aspects of the meteorological situation are also discussed in Lawrence et al (2020), Manney et al (2020), and/or Wohltmann et al (2020).]

**Other Scientific Comments / Questions:**

What do the authors mean by "classification" in the title?   The term is typically used for grouping and comparing things with similar characteristics, but it is unclear what sort of classification is being attempted herein.

Given the novelty of the ozone datasets used in this paper, I would like to see, especially in Section 2, more discussion of the TROPOMI data, the GTO-ECV data, and the relationships between them -- especially in conjunction with Figure 7, which compares time series from GTO-ECV in previous years with that from TROPOMMI in 2019/2020.  What are expected biases between the two datasets?  Are the discontinuities in the GTO-ECV data that might result in biases between some of the earlier years, or in non-physical trends?

Page 2, line 12: How is "the expected ozone value in austral spring" defined?  Do you mean the value that would be expected at that time of year with no chemical loss?  If so, how is that determined?

Page 2, line 25: Statements such as "...due to a strong and stable polar vortex in winter…" are too over-simplified, since there is (particularly in the much more dynamically active Arctic) no one-to-one relationship between vortex strength and temperature.

Page 3, lines 6-8:  What is this statement based on?  Given the similarity of LS temperature evolution to that in 2016 and later 2011 as the winter progressed, and the large chemical ozone loss and resulting low column in 2011 (coupled with the knowledge that dynamical variations that can reduce column ozone play a significant role even in the coldest Arctic winters, and the large interannual variability making winters as cold as or colder than 2011 very likely "sometime"), I see nothing unexpected about what happened in 2020!

Page 5, line 29:  I think "to large parts" is too weak here -- **no** chemical processing of any kind is needed to produce "mini-holes" as defined here.

Page 6, lines 8--9: There is no reason to expect this, since (as per above discussion in the major points) the strength and coldness of the vortex are not necessarily closely correlated, and "an ozone-hole like pattern" in the sense of chemical loss driving the ozone morphology would never be expected before March because chemical loss is limited when the polar regions are in darkness.

Page 6, lines 14--15:  But you don't even show the polar jet (which would have to be zonally resolved to show where the polar vortex was) in relation to the TOC, and show nothing about it at a level that is appropriate to determine where the LS vortex is.  See comment on lack of definition of polar vortex in major comments above.

Page 6, line 21:  If you accept that this value is an accurate reflection of the similarity of chemical loss in the 2020 Arctic to that in the Antarctic, it would be helpful to point out that this is about 10% of the **smallest** Antarctic ozone hole area (in 2019) on record, and less than 5% of typical Antarctic values (see, e.g., Figure 1 in Wargan et al, 2020, accepted for JGR, https://doi.org/10.1002/essoar.10503445.1; and references therein).

Page 9, lines 4--8:  Per previous comments, cold winters with substantial ozone loss can have weak polar vortices (e.g., 2004/2005), and not all winters with large ozone loss have unusually strong vortices.  "Cold" and "strong" are not synonymous in relation to the polar vortex.

Page 9, lines 17--24:  This paragraph seems irrelevant to this paper.  If indeed 2018/2019 needs to be mentioned, this could be done in a sentence by simply citing one or more of several papers that have been published on that winter (e.g., Butler, et al, QJRMS, 2020, and references therein).

Page 9, lines 24--32:  This paragraph seems largely irrelevant as well, and if the 2019 Antarctic ozone hole needs to be mentioned, that could be done by citing one or more of the several papers published on it (in particular the Wargan et al, 2020 paper mentioned above, which provides a detailed analysis of the dynamical and chemical mechanisms leading to the unusually small ozone hole in 2019; but there are also a couple of earlier references given therein).

Page 10, line 27 and line 30:  "From our point of view" is a statement that would appropriately preface an opinion, not a scientific statement.  If the statements following these can be backed up with evidence, there is no reason to use this language; if they cannot, they should not be made in a scientific paper.

Page 10, lines 13--14:  If this is intended to convey something beyond the point made in the previous sentence, a citation or some evidence should be given..

---

## Short Comment (SC4) · 24 Aug 2020

Dear Gloria,

thank you for your detailed comments regarding our manuscript. Your scientific remarks are appreciated and we will consider your points in the revised version of our paper.

A short reply to your points regarding the term "Arctic ozone hole" and with respect to the missing references in our manuscript, in particular those of the JGR/GRL special issue on the winter 2019/2020: we answered these two points in the short replies to Ingo Wohltmann's review and to the short comment given by you and Jens-Uwe Grooß. In summary, in the revised manuscript we will also compare the Arctic values with respect to corresponding Antarctic values, and we will no longer call it an Arctic

ozone hole. And, of course, the respective literature presented in the JGR/GRL special issue will be cited and the results will be discussed in the revised manuscript.

Your point (which was also raised by Ingo Wohltmann) regarding "the claim of being the First": It was not our intention to claim that we are the first who have detected such low total ozone values. What we would like to say in our submitted paper (and title) was that it is a general, initial overview of the winter/spring 2019/2020. So the choice of word was wrong. On the other hand we said that for the "first" time such (record) low values have been detected over the Arctic for such a long period. Maybe the formulations of the respective sentences were also misleading. In the revision of our paper we will change the wording and hopefully such misunderstandings will be avoided.

Best regards, Martin

---

## Author Comment (AC1) · 24 Aug 2020

Dear Gloria, dear Jens-Uwe,

for the first time record low total ozone columns around or below 220 DU were detected over a period of five weeks in spring 2020, which covered a larger area of the Northern polar region. The Arctic situation in March and early April was exceptional because it showed a persistent ozone hole-like pattern. In this case using the term "Arctic ozone hole" is probably a matter of opinion. But we got your (and also Ingo Wohltmann's) point! It was definitely not our intention to "transport sensation" or dramatize or overdraw the situation in Arctic spring 2020, which allow the reader to draw wrong conclusions with respect to Antarctic conditions. Therefore we will rephrase the

manuscript (including changing the title) accordingly. We will make it clearer that the corresponding total ozone column values for an Antarctic ozone hole are much lower and that the covered area is much larger, and that the Arctic conditions are not comparable with the Antarctic. The corresponding figures will include comparable values for the Antarctic and they will be discussed in detail.

Best regards, Martin Dameris (for the author team)

---

## Author Comment (AC2) · 19 Oct 2020

acp-2020-746 Reply to the review of Ingo Wohltmann by Dameris et al. Thank you very much for your detailed review and the specific comments regarding our manuscript. Your statements and suggestions are highly appreciated and have helped to improve our manuscript. We have considered them in the revised version of the paper. A detailed response to your comments is given below. Regarding your remark concerning the joint special issue of JGR and GRL on the 2019/2020 winter, we explained the circumstances in our short comment (July 30), which has led to the submitted draft of the manuscript. In the revised version of the manuscript we are referring to the (accepted) papers of the JGR/GRL special issue and we have set the results of these papers into context with our findings. In the following the points raised by the referee

are displayed in black and our responses are given in blue. General In general, I found most parts of the manuscript to be scientifically sound and would recommend publication. While there are some issues (see specific comments), I don't think that there is anything which cannot not be resolved. I have to admit that I had more of a problem with the wording and readability in some places. On the positive side, I did not find a single typo. But on the other hand, wording was quite awkward in some places, and sometimes the text could have been less confusing and better organized, it felt a little bit rushed in places. I have the impression that asking a native speaker to go through the text would help in many places. We thank the reviewer for this statement regarding our scientific results. We have tried to address the raised points of the reviewer and the choice of words has been selected more carefully in the revision, in particular with respect to the aspect of talking about an "ozone hole" over the Arctic. For instance the title has been changed appropriately. Moreover, specific passages from the text have been revised. Some of the figures have been slightly revised (labeling) and extended (e.g. regarding the Antarctic) to improve the readability of the manuscript and provide clearer scientific messages. A new figure (Figure 2) has been added.

Major comment Unfortunately, however, I have a major comment on a less scientific issue. The general wording and tone of the paper are quite sensationalist in title, abstract and conclusions (or as a colleague who is a native speaker put it: "attention-grabbing"). I don't find the wording appropriate in several places. I don't think that you do yourself or the stratospheric community a favor with that. You need to phrase your manuscript more carefully. In addition, things are sometimes not put into perspective, which may lead the reader to draw the wrong conclusions. If you imply conclusions here that are at least debatable and at the same time choose a manuscript title that will attract the interest of the public or press, I think this could be problematic. In particular: • Remove "first" from the title. This is not a contest, but a scientific paper. In addition, it is just not correct. There are papers under review in the upcoming GRL/JGR special issue that have overlap in content to what you write here, and the first paper has already been accepted. We understand the reviewers concern and therefore

we have changed the text. Among others, the title of the paper has been changed. As stated before (in our first short reply) our intention was not to be "the first" but to provide "an initial" overview of the situation in 2020. Further we also provide more context to avoid misunderstandings. Further papers of the GRL/JGR special issue, which are relevant to our manuscript, are addressed now and a paragraph pointing to the special issue has been added in the Conclusion.

• I find the repeated and prominent use of the phrase "ozone hole" highly problematic. In the abstract and title alone, it appears 7 times. This raises expectations and may imply conclusions for some readers which are not really backed by the facts or are at least debatable. Given that the phrase "ozone hole" has played a prominent role in the public discussion in the last decades, many people will have a certain understanding of the phrase which sticks in their minds, and we as a community should be careful what we write (in our own interest). I feel obliged to go a little bit more into detail why I think that prominently stating that the winter 2019/2020 showed an "ozone hole" is problematic. I think we probably agree that the winter 2019/2020 was exceptional. It was the coldest stratospheric Arctic winter on record and showed the lowest ozone columns and concentrations ever observed in the Arctic, which were comparable to typical values in the Antarctic ozone hole locally and for limited time periods. The Abstract and the title are appropriately revised. And, also in the following we have changed our wording regarding the "Arctic ozone hole". We removed it from the text entirely. Our intention was to clearly state that the situation was "exceptional" (as you put it) as (i) the TOC values were very low over a relatively long time period, and (ii) the shape of the region of low TOC looks "ozone-hole-like" (while we agree that the values are still much higher than in the SH). We tried to be more precise and to avoid possible misunderstandings. We are now comparing the spring situation in 2020 not only with spring 1997 and 2011, but also with a typical ozone hole over the Antarctic (2016) and with the small ozone hole in 2019. The "old" Figures 2, 5, 7, and 8 (now Figs. 3, 6, 8, and 9) have been extended by showing the respective values for the SH.

Having said this, it still was far removed from the usual conditions in the Antarctic ozone hole. – First, the area with Dobson values below 220 DU was much, much smaller than the typical area of the Antarctic ozone hole. First of all, the area of the Arctic vortex is typically smaller than the area of the Antarctic vortex (Manney et al., 2011, gives a number of 60 % for 2011). Then, the area with values below 220 DU covers only a small part of the vortex. This can easily be seen in your Figure 1. According to the numbers you give in the conclusions, even at maximum, the area was less than 5 % of the Antarctic ozone hole (0.9 million km2 compared to 20 million km2). In the end, this is a little bit of a problem with the standard definition of the ozone hole as the area below 220 DU. This definition does not take into account the area covered by the hole at all. But certainly nobody would call it an ozone hole if the area of the hole would be only one square meter. In context with the discussion of Figure 8 (now Fig. 9), we have now compared the size of the Antarctic ozone hole with the situation in spring 2020 over the Arctic. Also, at other places in the manuscript (in connection with "new" Figures 3, 6, and 8) the differences between the Antarctic and Arctic have been pointed out to allow for a direct comparison.

– Then, the vertical extent of the layer almost completely depleted in ozone in the Antarctic is much larger than the vertical extent of the depleted layer seen in 2019/2020 in the Arctic. Usually, ozone is depleted to near zero values in a large altitude range from about 350 K to more than 500 K in the Antarctic (e.g. Kuttippurath et al., doi:10.1038/s41612-018-0052-6). While the ozone profiles of 2020 show a pronounced minimum, very low values (below, say, 0.2–0.3 ppm) were restricted to a layer of a few 10 K depth around 450 K (see plot of a sonde measurement in Ny-Alesund on 27 March, measurement is the blue line). This also is a weakness of the 220 DU definition. In the Antarctic, values will usually fall far below 220 DU, while they only scratch 220 DU in the Arctic in 2020 (please see my comment to Figure 7 how to improve on this). Figure 7 (now Fig. 8) has been extended by adding the data of the two Antarctic ozone holes in 2016 (a typical one with respect to duration and strength) and 2019 (one of the smallest ozone holes). Some discussion has been added. Appropriate literature

has been considered, e.g. Wargan et al. (2020).

– While the lowest mixing ratios reached in 2019/2020 were comparable to mixing ratios that can often be observed in the Antarctic ozone hole (0.1– 0.2 ppm), they did not reach the near zero minimum values (0.01 ppm) that are typical for the Antarctic ozone hole. – And last but not least, it is also the time period. The Antarctic ozone hole lasts several months, while the time period with very low ozone values in 2019/2020 was at most 5 weeks or so. In context with the discussion of Figure 7 (now Fig. 8) , we have now compared the duration and shaping of the Antarctic ozone holes in 2016 and 2019 with the situation in spring 2020 over the Arctic.

Thus, I really would suggest to phrase things more carefully, e.g. to speak of an "ozone minimum" or "values comparable to values observed in the Antarctic ozone hole" and so on. Please see the specific comments for the places where I think you have to phrase things more carefully. • Sometimes, it seems that you would like to push the reader into a certain direction by omitting information in strategical places. While this means that you don't write anything scientifically wrong formally, you may push the reader to draw conclusions that are not correct. In particular, it would have been very easy to include Antarctic data in figures like Figure 1, 5, 6 and 7, and to discuss this data in the text to put things into perspective. I think it is really mandatory that you change the manuscript to be more balanced and to put things better into perspective. This was not our intention and hence we revised the manuscript accordingly (see previous comments). We hope that the changes in our manuscript are now in balance and that it is improved with respect to the discussion of the spring 2020 situation, especially in context with a typical Antarctic ozone hole.

Specific comments • Page 1, lines 11–12: Please rephrase to something like "record low ozone column" or similar and avoid the term "ozone hole". Has been changed.

• Page 1, lines 12–14: The sentence would work equally well when you omit "A persistent ozone hole pattern". Just start with: "Minimum total ozone column values. .

. ". To make the "for the first time" in the sentence work better, maybe you could write "for more than 5 weeks" (or 4 weeks?) instead of "about 5 weeks". Has been changed.

• Page 1, line 14 (next sentence): Suggestion: "Usually such low total ozone column values have only been observed in the ozone hole in the polar Southern hemisphere (Antarctic) in spring over the last 4 decades, but not over the Arctic." Slight change in text, but larger change in meaning. But please state here in addition that column values will go far below 220 DU in the Antarctic to put things into perspective. It would also make sense to state the other differences which I have outlined in my general comment here (smaller area, vertical extent and time period). The suggested slight change of the text is considered. In addition at the end of the Abstract a sentence is added to make clear the differences in the Arctic and Antarctic with respect to the low total ozone values and the differences in the respective area and time.

• Page 1, line 16: Change to "The record low values were caused. . . " Wording is slightly changed.

• Page 1, line 16: A stable vortex does not enable a cold stratosphere. This confuses cause and effect, when there is no wave activity. When there is wave activity, it is a little bit more complicated, please see my comment to page 5, line 3 (sorry, wrote that comment first. . . ). Please phrase that correctly. We agree with your statement. Wording is slightly changed.

• Page 1, line 20: "in the context of" is probably better. We keep "in context with". From our point it is the correct wording.

• Page 1, line 20: Replace "ozone-hole like features" simply by "cold winters" Wording is changed.

• Page 1, lines 27–28: I would delete this sentence. This is exactly what I would call "attention-grabbing", but it doesn't really transport information. This sentence has been deleted.

• Introduction: In a later comment (page 5, line 3–30), I suggest to add a short overview on how ozone depletion works somewhere in the introduction (PSCs, cold temperatures, chlorine activation, return of sunlight, . . . ) to be able to streamline the text in the later sections a bit. After the second paragraph in the Introduction section, a short paragraph about the processes of polar ozone depletion is added.

• Introduction: In some places, I find the references a bit odd, while I miss others. E.g. Langematz, 2019 and Loyola et al., 2009, would not be the first ones that come to my mind. I would expect the review paper of Susan Solomon from 1999 somewhere (Rev. Geophys. 37, 275–316, 1999). A paper that is mandatory to cite in a study like this is in my opinion Solomon et al., "Fundamental differences between Arctic and Antarctic ozone depletion", 2014, doi:10.1073/pnas.1319307111 (it is by chance that it is Susan again). This is a review paper about exactly the topic you are talking about here, and it also contains some very critical remarks about using the term "ozone hole" for the Arctic. Some additional references are included in the Introduction. The papers by Langematz (2019) and Loyola et al. (2009) are cited with respect to other points, i.e. the "definition" of the ozone layer and the importance of satellite measurements, which are monitoring the atmosphere; they are necessary prerequisites for a better understanding of atmospheric processes. But you are right: We are happy to add the papers by Solomon (1999 and 2014).

• Page 2, lines 5–10: You don't need to explain the meaning of the term "ozone column". This is basic textbook knowledge. OK, but finally we decided to keep this sentence.

• Page 2, line 7: Delete "so-called" Done.

• Page 2, line 11: You need a citation for the 220 DU threshold (it is for example defined in the WMO report 2018, along with further references). This is more or less an "official" definition which most people agree on, and you will need some references for that. All of your following discussion depends on this definition. The reference of

WMO (2018) is added now here.

• Page 2, line 14: This is misleading. Changes of ozone column by transport and changes of the column by chemistry are correlated (e.g. Tegtmeier et al., 2008, which you may want to cite here). A more dynamically active winter means both more transport of ozone into the vortex and less ozone depletion because of higher temperatures. Your sentence reads as if the difference between 450 DU and 220 DU would mainly be caused by chemistry. In fact, one of the fundamental differences between the Arctic and the Antarctic is that transport plays a large role in determining the Arctic ozone columns. Good point! The text is changed accordingly and the reference of Tegtmeier et al. (2008) is added.

• Page 3, line 3: Perhaps replace "with respect to low TOC" by "in reaching low ozone columns"? Text is slightly changed.

• Page 3, line 4–5: Some references from the paragraph starting page 10, line 27 would also fit, and see also my comment to this paragraph for even more references. Here we added only the references of Solomon (1999) and Tegtmeier et al. (2008), which are important ones. We think that also in the reference lists of the different WMO ozone assessments all relevant publications can be found.

• Page 3, line 11: It would probably be good to mention the official databases for ozone sondes here, i.e. WOUDC and NDACC. These ozone data sets of WOUDC and NDACC have been briefly mentioned now (later) in the manuscript and the Wohltmann et al. (2020) paper is cited in this connection.

• Page 3, line 13: You need to phrase that more carefully: "which led for the first time to ozone values below 220 DU in larger parts of the vortex for an extended time period". Thank you. The text is changed accordingly.

• Page 3, line 25: As far as I know, the nominal resolution of ERA5 is 0.28125 degrees. It does not really make sense to sample the data at a higher resolution (but

does not hurt either). Parts of the ERA5 data set are made available on the CDS at this resolution (cf. e.g. Hersbach et al., 2020). PV data on isentropes is not available on this grid, but on a reduced gaussian grid, which is then converted to the 0.28x0.28 degree resolution as stated by the reviewer.

• Page 3, line 24–28: You don't need to go into detail how you do a daily average. I trust you that you are able to do this correctly :-) In fact, you can replace anything between "For our investigations. . . " and the end of line 28 by "We use daily and monthly averages in the following". This is totally sufficient. We decided to keep this as is because this was one of the points raised by the Editor with respect to the first draft of the paper. Our intention is to make clear how we prepared the data for our analyses.

• Page 4, line 3 and line 8: Delete "first". This is really not relevant in the context of this paper. Again, this is no contest. A lot of work was necessary before such a consistent data set has been available for scientific purposes. And again, this was one of the questions raised by the Editor (concerning the first draft of the paper) with respect to the personal achievements of the author team.

• Why is Figure 2 the first plot that you discuss in the paper? You should change the order of the plots, so that Figure 2 becomes Figure 1. Figure 1 was already mentioned on page 3, line 8. Therefore we keep the order as is.

• Figure 2 and accompanying discussion: Lawrence et al. from the special issue contains similar figures and discussion. Please cite and discuss Lawrence et al. here. The paper by Lawrence et al. (2020) has been cited and discussed in this context.

• Page 4, line 25 and Figure 2: At first, I was a little bit confused by the plot because I didn't realize that the dots are monthly mean values and the lines are daily values, causing the colored dots not to lie exactly on the lines. Maybe there is some information overkill in the plot. One could replace the dots by a grey area showing the range of the daily values in all years for every day. We have slightly changed the figure captions. Hopefully it is clearer now that the dots indicate the monthly mean values.

• Page 4, lines 26–28: I would suggest to delete the part in parentheses. Either discuss this explicitly in the paper, or leave it at "caused by planetary wave activity". We think that referring to GSFC analyses is helpful. It contains additional information if someone is interested. We have slightly changed the sentence.

• Page 4, lines 28: Would be good to use the established terminology here (major warming, minor warming, sudden stratospheric warming...) Wording is changed.

• Page 4, lines 29: The wording is a little bit awkward and hard to understand. I would try wording like "stable and undisturbed vortex", "circular shape", "not displaced from the pole" etc. "deteriorated" is not the correct word. Text passage is slightly changed.

• Page 5, line 2: Figure 3 does not add any relevant information which is not contained in Figure 2. You could delete this figure without loss of information. We would like to keep this figure because it nicely indicates the persistent circular shape of the polar vortex.

• Page 5, line 3: You confuse cause and effect here. First, in the absence of wave activity, the polar region gets colder than mid-latitudes in winter due to a lack of sunlight and because of radiative cooling. Then, a pressure difference develops compared to mid-latitudes, which causes a geostrophic wind as response. For a correct discussion, you would need to explain the mechanisms of the Brewer-Dobson circulation in more detail. Interannual differences in polar temperatures or vortex strength are caused by differences in momentum deposition by breaking waves (mainly in the mid-latitude stratosphere) which drive the BDC. That means that both temperatures and vortex strength are correlated, but that this has a common underlying cause, and not that one causes the other. Based on your comment with respect to the Introduction, we have introduced the new paragraph and some additional information (e.g. Tegtmeier et al.) is given. We slightly changed the text on page 5. Regarding the mechanisms of the Brewer-Dobson circulation, we think that it is not necessary to explain the BDC in more

detail in this paper.

• Figure 5 and 6 and accompanying discussion: Lawrence et al. and Wohltmann et al. from the special issue contain similar figures and discussion (e.g. Figure 1 Wohltmann et al. and Figure 11 Lawrence et al.). Please cite and discuss Lawrence et al. and Wohltmann et al. The papers by Lawrence et al. (2020) and Wohltmann et al. (2020) have been cited and shortly discussed in this context.

• Figure 5 and 6: Please add typical Antarctic values in the figures and discuss them in the text. This will help very much to put things into perspective and I think this is mandatory. We have updated Figures 2 (now Fig. 3) and 5 (now Fig. 6), showing corresponding values for the Antarctic. We show the data for year 2016 as a typical situation (in particular with respect to a "normal" ozone hole situation; see also the revised Figure 9), and year 2019 with a stratospheric warming (not major) indicating one of the smallest Antarctic ozone holes (new Figure 9).

• Figure 5: It is a little bit confusing that you use a polar cap area and not the vortex area here as the area where you look for the minimum. That only gives the desired result because of the comparatively high temperatures outside of the vortex. You are interested in the minimum temperatures inside the vortex here, because these are relevant for ozone depletion. It would be more consistent to base the plot on the vortex area (the plot would look almost identical probably). Looking at the polar cap (50°-90°) is a standard diagnostic, which helps to identify the minimum temperature values on a solid foundation, in particular in undisturbed winters. And you are (as we assume too) right: the corresponding analysis concentrating on the polar vortex area would yield very similar results. Therefore we would like to keep the analysis as is.

• Figure 5: Can you really learn something from the minima of the monthly mean values? These will by definition always be higher than the daily minima. I don't really see that they provide any insight. I would suggest to remove them and to replace them by some grey area showing the range of the daily values over all years. The monthly mean

values (dots) allow getting an overview of the variability of the atmospheric system and you can classify the individual years (color dots). A figure with grey area based on daily values would yield to similar information. In addition this figure is related to Table 1.

Page 5, lines 11–12: The numbers for the vortex area are a little bit unintuitive. At least I don't have a really good judgement of them. An alternative would be to divide the values by the vortex area, which gives a value in percent, which is more easy to grasp. In addition, the unit for the cumulative area can't be correct. There must be some time unit missing (probably "days"). Here we only mean the sum of the vortex area and not the integral. Therefore, the units are correct. To make it clear, a half sentence has been added.

• Page 5, line 14: The citation seems odd. An obvious citation would have been the original study of Solomon et al., Nature, 321, 755, 1986. Or the WMO report or the 1999 Solomon review paper. We have added the Solomon (1999) paper.

• Page 5, line 16–20: Discussion on "ozone hole". Please phrase that more carefully. You could state here that record low values have been reached and that their temporal extent and the covered area were unrivalled in other years (but please check that, this is just what I suppose is correct). It is certainly also ok to mention the 220 DU definition of the Antarctic ozone hole here, but I think it is mandatory here to discuss the smaller area, the limited vertical extent and the limited time period compared to the Antarctic ozone hole to put things into perspective. The text is changed accordingly. Setting the areas of low Arctic TOC values in context with the respective Antarctic values is done later in the manuscript.

• Page 5, line 20: As far as I can see, this is the first time you mentiom Figure 1. Please correct the order of the figures. Figure 1 was mentioned first on page 3.

• Figure 1: Add contours for the 220 DU contour and the vortex edge. These are things that are really hard to see in a coloured contour plot. And the 220 DU contour is really central for the discussion in your paper. We think that it is not necessary. As said

in the figure caption: "The area with total ozone values below 220 DU are denoted with the dark purple color." This should be sufficient.

• Page 5, paragraph lines 3–30: The readability of this paragraph suffers because of two issues in my opinion: First, you introduce 5 figures in this short paragraph, but only by mentioning them in parentheses. It would help immensely to insert a few sentences starting with "Figure xxx shows. . . " in this paragraph. Text is changed.

In addition, you try to explain how ozone depletion works in some half-sentences and in parentheses here and introduce things like Cl activation and NAT clouds. I think you could improve your manuscript a lot by adding a short paragraph in the introduction explaining the basics of ozone depletion in a few sentences (cold temperatures, PSCs, return of sunlight, chlorine activation, CFCs, . . .). Then, you could refer to that later, and the text would read much more fluently. You have tried to explain even more basic things in this manuscript, like the definition of ozone column or the averaging, so it seems odd that you don't explain some things in the introduction which would really help to streamline the text in your paper. Such a short paragraph is now added in the Introduction. Then it should be clear in the manuscript.

• Figure 6: To make things more consistent, you could show the range of values of all other years in grey, as in Figure 2, 5 and 7. We would like to suggest keeping the figure as is. The aim of Figure 6 (now Fig. 7) is only to demonstrate the differences between these three Arctic winter situations.

• Page 6, line 9: Why is this expected from January on, and why is this expected with values above 220 DU? I don't understand your reasoning here. Maybe it would be better to simple write "is observed"? Wording is changed.

• Page 6, line 12: See comment page 3, line 11. As said already, WOUDC and NDACC have been shortly mentioned as an additional data source.

• Page 6, line 21: Please compare the area below 220 DU to the area of the vortex

here. This would really be important and interesting information which helps to put things into perspective. Text has been revised; the Arctic area with TOC below 220 DU is about 4% of the polar vortex area, which is determined for March 12. It is based on our new Figure 2 (PV). Some text has been added.

• Page 6, line 22: Again, the units are not quite correct (at least formally). A unit of "days" is missing. As said already, the units are correct.

• Figure 7 (and accompanying text): What really would add a lot of value to your paper with relatively small effort would be to show typical values from the Antarctic in the same plot and to discuss the differences between Arctic and Antarctic in the text. Please add Antarctic values to the figure to put things into perspective. As suggested, in the new Figure 8 (and also in the new Figure 9), we add respective ozone data of the Antarctic. As an example, for a typical ozone hole (i.e. with respect to the means of the ozone hole area and its variation with time) we choose the year 2016. In addition, we also display the corresponding data for the year 2019, showing a small ozone hole. Corresponding zonal winds and minimum temperatures are also added in Figures 2 (new Fig. 3) and 5 (new Fig. 6).

• Figure 7: The same comment as for Figure 5 applies: It would be better to base the figure on the vortex area and not on a polar cap area. As said, looking on the polar cap ($50°$ to $90°$) is a standard analysis, and we think that it leads to robust results. Looking at the respective polar vortex area would very likely yield a similar message. Therefore we would like to keep the figure as is.

• Page 6, line 27: There is a lot of literature on the 2010/2011 winter. Maybe it wouldn't hurt to cite one or two studies more here. Suggestions: Sinnhuber et al., 2011, doi:10.1029/2011GL049784, Hommel et al., 2014, doi:10.5194/acp-14-3247-2014, Strahan et al., 2013, doi:10.1002/jgrd.50181, Kuttippurath et al., 2012, doi:10.5194/acp-12-7073-2012, . . . Here, we add the reference of Kuttippurath et al. (2012).

• Table 1: I could live well without the table. The information can already be found in the plots. We would like to keep the table. It shows impressively (in our opinion) the differences of these three years.

• Page 8, line 9: The wording is awkward. Suggestion: "The winter 2019/2020 showed a larger volume below the formation temperature of PSCs than other winters for an extended period of time." Thank you for the suggestion. We changed it accordingly.

• Page 8, line 11–15: This is visible in the $HNO_3$ measurements of MLS. Please cite the Manney et al. paper from the special issue here. You don't need to speculate. Of course! We are happy to cite Manney et al. (2020) here. It is nice to see that our thoughts are confirmed.

• Page 9, line 12–14: Can you get a little bit more quantitative here? What is the quantity you are looking at here? Vortex mean temperatures at some level? In the moment, these sentences do not convey enough information to be useful (I am well aware that the uncertainties are large and will only allow a qualitative statement). Based on the information available in the SPARC newsletter article, we put in some more information in our manuscript. The statement by Labitzke and Naujokat here was very clear that the spring 1997 was the coldest in the Berlin time since from 1955 to 2000. We give a short comment on the uncertainties of the Berlin analysis.

• Page 9, line 25–32: And of course, there is the ozone hole split of 2002. Of course! We add some information about the split-event 2002 and refer to respective literature.

• Page 10, paragraph 1–6: The statements in this paragraph are problematic and don't really tell the truth because you omit information. You don't mention that the temperatures in the Antarctic are considerably lower than in the Arctic (even for 2019/2020) and that the period of low temperatures is much longer in the Antarctic. If you would have added this information to your plots (e.g. Figure 5 and 6), this would be quite obvious. It is also not true that this would result in ozone depletion rates that are comparatively strong. The depletion rates will also depend on the amount of ClOx, which

in turn e.g. depends on the amount of descent in the vortex. In fact, this is a rather complicated topic. The crucial difference between the Arctic and the Antarctic which leads to very low ozone values is the much longer time period with ozone depletion. Since ozone loss is usually in saturation in the southern hemisphere, some details on the path to zero ozone don't really matter here. You are fully right! The text is changed (some is deleted) and the respective information about the difference in Arctic and Antarctic temperature is discussed. Figure 5 (new Fig. 6) is revised by showing now also the respective Antarctic temperatures for 2016 and 2019.

• Page 10, line 10: Please give numbers for the chlorine content. This would show that the differences are not that large (I assume 10 % to 20 % difference?) and that it is not too surprising that differences in temperature are the main driving factor. But in principle, you are right, this is worth mentioning. Maybe some additional discussion along the lines above is appropriate. The change of the atmospheric chlorine content over the about last 20 years was already mentioned on page 2 (line 18). The number (15%) is given here again with the corresponding reference (Chapter 1 in WMO, 2018). And following your suggestion a sentence is added to highlight the importance of the low temperatures.

• Page 10, line 17: In fact, a dehydration event can clearly be seen in the MLS H2O measurements. Please cite Manney et al. from the special issue here. Done!

• Page 10, lines 18–22: You repeat what you already have said on page 8. Please delete. See also the comment to page 8, lines 11–15. We think that a short repetition here is helpful because we are now in the discussion section. The two sentences have been revised and Manney et al. (2020) is cited again. The sentence about our suspicion is deleted and the half sentence about 1997 and 2011 is also deleted.

• Page 10, line 24: See comment page 10, line 10. Done!

• Page 10, line 27. Please rephrase "atypical ozone hole" to something like "record low ozone values". Is changed!

• Page 10, line 29 and line 32: I totally agree that you never can tell climate change from a single year and that you have to look how this evolves in the future in the context of a longer time series. On the other hand, you can have a look into the winters observed so far and don't need to wait for the future to have a long timeseries. And the year 2019/2020 does add information to this timeseries, since it was the coldest Arctic winter observed so far and the coldest winters have become colder in the last years quite consistently, at least according to some metrics and meteorological data sets. You could cite Wohltmann et al., Figure 1, from the special issue for a figure illustrating this quite well. But in general, I agree with you. The paper by Wohltmann et al. (2020) has been cited here.

• Paragraph Page 10, line 27 to page 11, line 10: Some additional recent relevant studies on this topic: Ivy et al., 2016, doi:10.1175/jcli-d-15-0503.1, Rieder and Polvani, 2013, doi:10.1002/grl.50835, Butchart et al., 2010, doi:10.1175/2010JCLI3404.1, Bednarz et al., 2016, doi:10.5194/acp-16-12159-2016. These contain a lot of interesting discussion on this somewhat controversial topic (Are coldest winters getting colder etc.), which you may want to add here. References of Bednarz et al. and Ivy et al. are now included.

• Page 11, line 2: I think an important point to mention here is that the other important driver of changes in stratospheric temperature (apart from changes in radiative cooling by greenhouse gases) are changes in the strength of the BrewerDobson circulation (e.g. Langematz et al., 2014 and many more), and the BDC in turn is affected by climate change. Thank you for your point. For the future most climate models indicate a strengthening of the BDC. But comparisons of climate model (incl. CCM) results with analyses of observations since 1975 (e.g. Engel et al.) so far did not show a consistent picture for the past. We have now added a sentence indicating the possible role of an intensified atmospheric circulation in the future. Langematz et al. (2014) was already included in the manuscript, but this paper is now placed in better relationship.

• Page 11, line 11–14: This paragraph seems a little bit out of context. Yes, but we

got such questions about the enhanced CFC-11 emissions many times in connection with the record low Arctic ozone values. Therefore this small paragraph at the end of the discussion part is included. We hope that you do agree to that.

• Page 11, line 16–17: The sentence seems overly complicated and wording is a little bit awkward. Suggestion: "This study presents a description of the Northern winter and spring season 2019/2020 considering the. . . " "regarding" seems not to be the correct English word to me, I think "considering" is what you mean). Thank you! Your suggestion is accepted.

• Page 11, line 17–19, 23–26: Please phrase more carefully. It is ok to discuss that the observations were below the 220 DU threshold for a longer period of time, but please also discuss the differences to the Antarctic ozone hole in area, vertical extent and duration that I have mentioned in my general comment right after lines 17–19. I am very happy that you finally discuss some of this in lines 23–26, but please move it a few lines up and discuss a little more. I.e. put things into perspective. Please don't call it an "ozone hole". And don't claim that this year was the first occurence of an "ozone hole". This was already claimed by some people in 2011, and it didn't help in the discussion back then, too. In general, I don't find this chase for superlatives very helpful, and it does not help to advance our scientific understanding. This paragraph is slightly revised. The expression "ozone hole" is deleted.

• Page 11, line 23–26: I am very happy that you finally discuss this here, but I hope you agree that it would have been necessary to mention this much much earlier (and more often) in your manuscript. It is now discussed in connection with the Figures 2, 5, 7, and 8 (new Figs. 3, 6, 8, and 9).

Technical corrections • Page 3, line 18: You can delete "It must be noted that". Deleted!

• Page 6, line 16: I don't think "outstanding" is the right word here. Maybe "prominent" or "remarkable"? Changed!

• Page 8, line 1 and 3: I would write "ice PSC" and not "ICE-PSC" (ice is not an abbreviation). Changed!

• Page 8, line 28: "Discussion" and not "Discussions" Changed!

• Page 9, line 17: "recall" is not the right word. Suggestion: "We note" Changed!

• Page 11, line 19: You very probably mean "compared" and not "confronted"? Changed!

• Page 11, line 20: I would say "which show" and not "which are showing" Changed!

• Page 11, line 20: I would say "the most recent datasets" Changed!

Please also note the supplement to this comment:
https://acp.copernicus.org/preprints/acp-2020-746/acp-2020-746-AC2-supplement.pdf

---

## Author Comment (AC3) · 19 Oct 2020

acp-2020-746 Reply to the review of referee #2 by Dameris et al. Thank you very much for your detailed review and the specific comments regarding our manuscript. Your statements and suggestions are highly appreciated and have helped to improve our manuscript. We have considered them in the revised version of the paper. A detailed response to your comments is given below. In the following the points raised by the referee are displayed in black and our responses are given in blue. General comments I wonder if it is fair to refer to the Artic winter 2019/20 as an 'ozone hole'. While the authors are clearly correct in stating that total column ozone falls below the 220 DU threshold, typically used to define the edge of the ozone hole, it should be remembered that in the Antarctic column ozone values typically fall far below 220

DU, and for a timescale measured in months. Can the authors use further common metrics (ozone mass deficit, minimum column ozone, etc) in their evaluation to give a better understanding of the column ozone evolution? Additionally, I feel it would be beneficial if the authors included data from the Antarctic in their timeseries plots, so the reader can get an impression of the how the Arctic winter 2019/20 compares to what is more generally considered an ozone hole. The authors state that these low ozone values cover a large area (0.9 million km2), but that is a tiny fraction of the area covered by the Antarctic vortex. I feel that either the authors should refrain from using the term 'ozone hole' or to place this term into context by comparing it with the Antarctic ozone hole and state explicitly that it is much smaller and shorter lived than the In the revised manuscript we are now talking about record low ozone values in spring 2020 or we name it an ozone hole-like feature. The term "Arctic ozone hole" is avoided in the revision. In addition, the Arctic values are compared and discussed with corresponding Antarctic values. The Figures 2, 5, 7, and 8 (now Figs. 3, 6, 8, and 9) have been updated to allow for a direct comparison of Arctic and Antarctic values and quantities. In accordance the title is also changed. In Figure 7 (new Fig. 8) we are presenting the minimum TOC in the polar cap region (50°-90°). It is expected that the minimum TOC are detected in this latitudinal region, which covers also the inside of the polar vortex.

Further, a lot of emphasis is placed on the idea that the winter 2019/10 was the first instance of column ozone falling below the 220 DU threshold. However, the authors' Figure 7 shows that there are repeated instances of column ozone below 220 DU in the thin black line. While the authors refer to these as mini-holes, and explain the role in dynamics in their formation, I feel a distinction should be made between these and the 2019/20 winter – is it fair to say that this winter constitutes and ozone hole because these events are longer lived? While the winter 2019/20 is certainly atypical, it is wrong, based on this figure, to say, as the authors do on P11L17-18, that it is the first time these values have been observed. And if the qualify here is that they occur over a 'large area', does a new definition for an ozone hole need to include some

measure of the areal extent? You are right and especially the sentence ("For the first time . . .", P11 line 17-18) is misleading. We revised this sentence and also at other places in the manuscript; the statements are now hopefully clearer. In particular, we are now trying to avoid the impression that such low values (TOC) are observed for the first time. It is now clearly stated that it shows an ozone hole-like structure with such low values over a longer time period. We hope that it is now much clearer stated that such record low TOC values (below 220 DU) in the Arctic were detected over a period of five weeks in Arctic spring and that this is observed for the first time. And, it is now clearer stated that the TOC values are certainly higher than the respective values in Antarctic spring, and that the area of the Antarctic ozone hole is much larger in comparison with the area of low TOC values in Arctic spring 2020.

I miss in the introduction some general information on the processes involved in polar ozone depletion. While these processes are mentioned later in the manuscript, a paragraph in the introduction detailing the polar vortex, cold polar lower stratospheric temperatures, PSC formation, heterogeneous chemistry, and subsequent catalytic ozone depletion upon return of sunlight to the polar vortex would significantly aid the reader. Additionally, I would like to see more information on how the Arctic and Antarctic differ: increased wave activity in the Arctic, the fact that the Artic vortex is often displaced from the pole, which can affect the amount of sunlight that can reach the vortex, the relative importance of chemical depletion vs transport. A new paragraph is now included in the Introduction, which discusses briefly the involved processes regarding polar ozone depletion. Some more information (not only in the Introduction) about the differences between Northern and Southern winter conditions in the stratosphere is given, which is also related to the dynamics and the transport of air masses. Corresponding references are added.

The authors focus on the large-scale meteorological conditions within the winter 2019/20 Arctic polar vortex, particularly the area below the 195K threshold as a metric for PSC occurrence and chlorine activation. Can they say anything about local conditions, particularly for example the role of orographic gravity waves during the winter of 2019/20 and the impacts of these on local temperatures? This is a very interesting and important question. Yes, in our study we are focusing only on large-scale processes. You are right, possibly orographic gravity waves can affect local temperature and this definitely could impact the formation of PSCs, in particular in the Northern hemisphere. Unfortunately, looking in more detail on local effect is beyond the scope of this study.

The Harris et al. (2010) paper cited in the manuscript highlights linearity between PSC occurrence and ozone depletion. Similarly, Hommel et al. (2014: Chemical ozone loss and ozone mini-hole event during the Arctic winter 2010/2011 as observed by SCIAMACHY and GOME-2) highlight linearity between total column ozone change at 100 hPa eddy heat flux. Are the authors able to say something about if the winter 2019/20 falls on these linear relationships identified in past studies? Or does this extreme winter violate the relationships identified in other studies? So far, we did not carry out a more detailed analysis looking at the linearity between TOC and the change of the meridional heat flux at 100 hPa mid-latitudes as discussed briefly in the paper. Since the temporal evolution of the meridional heat flux in winter 2019/2020 indicates smaller values (variability) than usual (see GSFC webpage, which is mentioned in the manuscript) and the spring TOC values (see new Fig. 3) are low in the polar vortex, this assumption could hold. The same could be also true for the linear relationship between the occurrence of PSCs and ozone depletion. We have not analyzed the rate of chemical ozone depletion, but our analyses of conditions for the formation of PSC are hinting in this direction (see also the papers by Manney et al., 2020, Lawrence et al, 2020, and Wohltmann et al., 2020, which are considered and discussed in the revised manuscript). To our current understanding this was an expectable winter (with respect to the known dynamical conditions), leading to exceptional TOCs.

Some key references are missing from the manuscript, with many instances of only one, recent citation given during key discussion. I would encourage the authors to expand upon the literature already cited in the manuscript. We have added several

references in the revised manuscript. Some of them have been published very recently (see also our short replies to previous reviewer comments).

Specific comments: P2L19: 'Nevertheless, the current atmospheric content of CFCs is still enhanced. . .'. It would be beneficial to explicitly state a date here, i.e. '...still enhanced with respect to 1980s values. . .' "with respect to 1980s values" is included now.

P2L22: Care should be taken when using a term such as full recovery. While several studies show that column ozone is projected to return to 1980s values by the middle of the century, is that really full recovery? Some of this signal is driven by stratospheric cooling resulting from increased $CO_2$ mixing ratios, and is separate to recovery driven by reduction in ODSs. I would prefer the authors say something about ozone return to historic values, which is an important part of the recovery story, rather than 'full recovery'. Has been changed accordingly.

P5L3: Is 'strong cooling' correct here, or are the cold temperatures a result of reduced warming? Can the authors say anything about the radiative and dynamical processes operating within the polar lower stratosphere? This thought is also applicable to P7L7. The sentence has been changed. We state now clearer that the dynamical conditions in 2019/2020 with low planetary wave activity result in strong radiative cooling of the polar lower stratosphere during polar night, which causes a strong polar vortex. A more detailed discussion is given now about the importance of radiative cooling and reduced (meridional heat) transport of airmasses.

P5L21: The analysis here focuses on column ozone values north of 50°N. However, Figure 1 of the manuscript shows that the Arctic vortex is not symmetrical about the pole, and so this average includes considerable amounts of column ozone from outside the Arctic vortex. Is it possible to plot vortex averaged column ozone instead, and so separate out the low values from inside the vortex from the high values outside? In Figure 7 (new Fig. 8) we are looking at the daily minimum TOC values north of 50°N.

We are not looking at the mean TOC values in the polar cap region (50°-90°N). We are comparing the minimum values of the three Northern winters 1996/1997, 2010/2011, and 2019/2020. In the revised Figure 7 (now Fig. 8) we have added the seasonal evolution of the minimum TOCs over the polar cap of the two Antarctic years 2016 and 2019. In principle, we expect that a comparison of the averaged TOC of the polar vortex will provide a qualitatively information, which is similar to the minimum TOC. In addition, we have added a new Figure (Fig. 2 in the revision), which shows the respective PV values on the 475 K isentrope. Appropriate explanations are given in the revised manuscript.

P6L22: 'The daily accumulated ozone hole area in March and April was estimated with 4 million km2' – how does this value compare to that for September and October of a typical year in the Antarctic? I suspect the Antarctic value is many times larger. If so, is this a useful metric – I feel it may be misleading if not placed into context. We set the numbers of the Arctic and the Antarctic winter/spring seasons into context, now. This comparison clearly shows that the values for the Arctic are very much smaller than those found in the Antarctic. Thus, misinterpretation is now hopefully avoided! The numbers discussed here and shown in Figure 6 (now Fig. 7 in the revision) should only be compared for the NH winter/spring seasons and are supposed to facilitate the intercomparison of the Arctic situations discussed.

P10L1-6: Care should be taken here attributing all of the low column ozone values to chemical depletion. The authors discuss the importance of dynamics in the preceding paragraphs in preconditioning the polar vortex, but the phase 'ozone depletion rates' to me describes ozone loss through catalytic reactions, whereas in actuality the low column ozone is driven in part by chemistry and in part by reduced transport of ozone to the polar cap. This is obvious from your Figure 7, as column ozone increases from December to May, and this is not driven by chemistry. You are completely right! This paragraph has been revised.

Technical: P2L26: Check use of 'Exemplarily' Changed to "For instance".

P7L22: replace 'cumulated' with 'cumulative' – also other instances throughout the manuscript. Changed.

• Page 1, lines 27–28: I would delete this sentence. This is exactly what I would call "attention-grabbing", but it doesn't really transport information. This sentence has been deleted.

P11L19: remove 'a' from 'about a five weeks' Done.

The x-axis label for all timeseries plots says 'time [days]', which I would expect to be a set of numbers, but the plot shows date on the x-axis. Please revise. They have been changed accordingly in the new Figures 3, 6 and 7.

Please also note the supplement to this comment:
https://acp.copernicus.org/preprints/acp-2020-746/acp-2020-746-AC3-supplement.pdf

---

## Author Comment (AC4) · 19 Oct 2020

acp-2020-746 Reply to the short comment of Gloria L Manney by Dameris et al. Thank you for your interest regarding our paper and in particular for your detailed comments and the specific points regarding our manuscript. Your statements and suggestions are appreciated and have helped to improve our manuscript. We have considered your critical notes in the revised version of our paper. A detailed response to the comments is given below. In the following the points raised by GL Manney are displayed in black and our responses are given in blue.

Other comments have already discussed the overly-casual and ill-defined use of the term "Arctic ozone hole" and I believe that subject has been covered well already

(though my primary scientific comment below will touch on it in the context of comparison of specific winters). Other comments on / reviews of this paper have also already discussed the claim of "First" and the lack of citations of other papers in review and published on this exceptional winter, and I agree overall with their remarks. I do have additional "philosophical" comments on this subject: I am happy to see numerous papers submitted . . . These are, as I said above, philosophical rather than scientific views, so I can only ask the authors (of this preprint and others!) to ponder them and make revisions according to their judgement of the merit of these points. As mentioned in the replies to the referees, we have changed our choice of words, in particular with respect to the terms "Arctic ozone hole" and "First". We would like to mention again that first to us meant not "the first" but rather "an initial". We did not want to convey that the situation in 2020 in the Arctic is similar in size and duration to Antarctic ozone holes. To avoid any misunderstandings, we have formulated it more carefully in the revised version. The missing literature is now cited and discussed, and also the accepted papers of the special issue of JGR/GRL with respect to the winter 2019/2020. The submitted and not yet accepted papers, which are related to our work, are also mentioned. Further we added a short paragraph referring to the corresponding special issue in our Conclusion section.

Major Scientific Comments: The biggest issue that has not been raised in other comments / reviews at the time I'm writing this, and that I believe must be addressed before peer-reviewed publication, is the comparisons of 2019/2020 with 2010/2011 and 1996/1997, and the failure to communicate the very large differences in polar processing and ozone loss in 1996/1997 compared to the other winters studied. In the context of comparing superficially similar springtime lower stratospheric vortex conditions in 1996/1997 and 2010/2011, the very large differences in polar chemical processing in those two winters have been extensively highlighted, first in detailed discussion in the supplementary information (SI) of Manney et al. (2011, Nature), and in numerous later publications culminating in a detailed summary / synthesis in the WMO 2014 Scientific Assessment of Ozone Depletion (section 3.2.3.3), which provides further references. In

short, for numerous reasons (very late lower stratospheric, LS, vortex development and late drop of temperatures below PSC thresholds, smaller altitude region of low temperatures, weaker LS vortex throughout the winter, little/no denitrification, etc), chemical ozone loss was much less in 1997 than in 2011 (and hence than in 2020, which saw as much or more chemical loss as in 2011, eg, Manney et al, 2020; Wohltmann, et al, 2020; Grooß and Müller, 2020). We have added some statements and corresponding references in the revised version, which clearer point out the larger differences of chemical ozone loss of the three winter-spring seasons 96/97, 10/11, and 19/20.

Moreover, dynamical conditions led to frequent ozone mini-holes (e.g., Coy et al., 1997) and higher tropopause altitudes (e.g., Manney et al, 2011, Nature, SI) in spring 1997 that contributed to lower column ozone via dynamical processes than followed other winters with comparable chemical ozone loss. This is an important distinction that it is essential to address for the comparisons in this manuscript to provide accurate information on the similarities (a few) and differences (many) between 1997 and the other two winters considered. Statements such as (to pick only one example, page 10, lines 8-9) "...all three years showed particularly strong ozone depletion..." are scientifically inaccurate. This also folds in with the inadvisability of lightly using the term "Arctic ozone hole", as 1997 is a classic case of a situation that looked superficially similar to the Arctic winters, 2011 and 2020, with the most chemical ozone loss and in some ways "Antarctic-like" conditions (see WMO 2014, Section 3.2.3.2; Manney et al, 2020; Wohltmann et al, 2020), but which in fact had chemical processing that was in no way comparable to that in the Antarctic. Thank you for the comment that in spring 97 ozone mini-holes are frequent and clearly affected the TOC. We added this point (including the corresponding literature). With respect to the mentioned statements: You are fully right! The used mode of expression was misleading as "ozone depletion" is likely to be understood as "chemical ozone depletion". We have formulated it more carefully in the revised manuscript. (Our intention was to say that in all three years particularly low TOCs were observed in March.)

I also have concerns with the description of the dynamical conditions in relation to previous winters. The dynamical situation is described using only 10hPa zonal mean winds and 50hPa temperatures. Zonal mean winds in the middle stratosphere (10hPa as opposed to the levels around 50hPa where chemical processing maximizes) are virtually irrelevant to the state of the lower stratospheric vortex, because: (1) Vortex strength, size, and geometry vary strongly with altitude in different ways in different winters – we have seen winters (such as 2010/2011) where the vortex was exceptionally strong in the lower stratosphere but not in the middle stratosphere, and winters (such as 1997) where the vortex was for much of the winter fairly typical in the middle stratosphere but exceptionally weak in the lower stratosphere. (2) The Arctic vortex is rarely close to symmetric or pole-centered, even in the coldest and/or most dynamically quiescent winters (see, e.g., Figure 1 in Manney et al, 2020, or any of numerous other publications in the past ∼20 years), and its size, shape and position vary dramatically both intraseasonally and interannually. Thus, zonal means, even were they examined at altitudes in the range where LS polar processing occurs rather than at 10hPa, provide very little information on characteristics of the polar vortex such as size, location, and strength. To get an overview of the dynamic state of the stratosphere in specific winters it is usual to analyze the temporal evolution of the zonal mean wind at $60°$ (10 hPa or 30 hPa) and the zonal mean temperature in polar regions (e.g. at $80°$, 30 hPa or 50 hPa) or alternatively the minimum temperature in the polar cap region ($50°$ to $90°$, 50 hPa). Both are providing a good overall view of the dynamic situation. For our considerations, in Figures 2 and 5 (now the new Figs. 3 and 6) we identify obvious differences in the seasonal behavior in the individual winters (NH and SH), which support our interpretation of the different years. In our discussion of results, we are concentrating on these pressure levels. Lawrence et al. (2020) also focused on the zonal mean zonal wind at $60°$-$65°$N, 10 hPa (their Fig 1) and the minimum temperatures poleward of $40°$N at 50 hPa (their Fig 11). But you are right that we have to point out more clearly the height dependence of the polar vortex and its changes. We have mentioned it in the revised version, also by citing the paper by Lawrence et al. (2020). In addition,

[Figure]

Figure 2 (new) shows the PV on 475 K with respect to Figure 1, indicating the strength and position of the polar vortex.

Similarly, 50hPa minimum temperatures north of 50N and area of T < 195K at 50hPa, while very relevant to polar chemical processing, are by themselves inadequate to characterize the potential for chemical ozone loss in the LS vortex because: (1) The vertical structure/location/extent of the region with temperatures conducive to PSCs varies strongly interannually and within seasons; this is one of the reasons why one of the most useful measures of polar processing / ozone loss potential (both day-to-day and as a measure of total ozone loss potential in a given winter) is V_psc, the area below the PSC or chlorine activation threshold integrated over all lower stratospheric levels. (2) Because the LS vortex varies strongly in size, shape, and position, while the high-latitude minimum associated with the polar vortex is usually north of 50N in dynamically quiet winters, this may not always be the case, and is certainly not always the case in winters with strong SSWs during the cold period (Dec–Feb). With respect to Figure 6 (new Fig. 7) and the corresponding paragraphs, we have decided to keep our analysis only on the 50 hPa level. The results of this altitude range are representative. They are mostly qualitatively in line with the results of nearby layers (for instance 30 hPa; not shown in the paper). Our findings are in qualitative agreement with the V_psc analyses in the lower stratosphere. Therefore, we have now cited the papers by Wohltmann et al. (2020) and Lawrence et al. (2020). The text in the paper is changed accordingly. In addition, we have mentioned the importance of the characteristics of the polar vortex and that it varies with height and in different Northern winters.

Furthermore, in relation to column ozone and its relationship to the LS vortex and low temperatures, because low column ozone is strongly spatially correlated with low LS temperatures by dynamical processes (see, e.g., discussion and references in SI of Manney et al, 2011) and the region of low temperatures in the LS is often not well-correlated with the lower stratospheric vortex (see, e.g., Manney et al, 1996, GRL; Mann et al, 2002, JGR; SI of Manney et al, 2011; Lawrence et al, 2015, ACP; and references in those latter two works), in absence of strong chemical depletion, the shape / extent of the region of low column ozone is not expected to be correlated with the shape / extent of the polar vortex. Therefore, in the maps of column ozone in Figure 8, we cannot judge whether the morphology of the low region is consistent with strong ozone loss unless we know how it relates to the morphology of the LS vortex. E.g., one of the most commonly used metrics for this is a contour or contours of potential vorticity (PV) on an isentropic surface somewhere in the LS (somewhere between about 450 and 550K is typical, commensurate with the approximate levels where ozone contributes most to the column), with value(s) such that it is (they are) in the region of strong PV gradients bounding the vortex. (Similarly, the strength of PV gradients along the vortex edge is a common and valuable metric of vortex strength – while maximum windspeeds at an appropriate level would also be informative of vortex strength, zonal mean winds are not.) PV on isentropic surfaces is readily available in all modern reanalyses including the ERA5 reanalysis used herein. Please note that Lawrence et al (2020; as I write this in late August, nearing completion of minor revisions for JGR, and available on ESSOAr since mid-June) in their section 3.4 (submitted version) provide a detailed discussion of LS vortex strength and chemical processing potential in 2019/2020 in comparison with the record from 1979 through 2019 and in particular compare with 1997, 2011, and 2016 (2016 is of interest because, while low temperatures and chemical processing ended much earlier, it still is the record cold winter in January and February, and had overall greater polar processing potential and more chemical loss than that in 1997; e.g., Manney and Lawrence, 2016, ACP; WMO 2019; and references therein), using widely accepted diagnostics / methods with uncertainties quantified. Because this paper for the AGU Special Collection provides an overview of the dynamical conditions during the winter, it is designed to provide this "foundational" material in a thorough way so that other papers on this winter can start with that comprehensive description as background. [While not as relevant to this paper and the 2019/2020 ozone loss, Lawrence et al (2020) also describe thoroughly in their Section 3.1 the evolution of 10hPa zonal mean zonal winds in 2019/2020 versus

climatology (their Figure 1) and in the context of the time series since 1959 (their Figure 2).] [The points above for which I have not provided references are well documented, e.g., in the past ∼4 WMO Assessments and references on Arctic polar ozone and ozone loss therein. The relevant aspects of the meteorological situation are also discussed in Lawrence et al (2020), Manney et al (2020), and/or Wohltmann et al (2020).] The aim of Figure 8 (new Fig. 9) is to indicate the differences of the TOC values with respect to the monthly means of the spring months (March and October). It should demonstrate that March 2020 showed the smallest TOC in comparison to March 1997 and March 2011 and that the Arctic region shows significantly lower TOC. You are right, based on our Figure 8 (now 9) we cannot decide whether the morphology of the region of low TOC is consistent with strong ozone loss unless we know how it relates to the morphology of the polar vortex in the lower stratosphere. Therefore, we checked the contours of PV at 475 K and 530 K. In Figure 2, we are now showing the results for the 475 K level (530 K shows basically the same result) with the same dates as given in Figure 1 (TOC). In more detail we have discussed now the strength of the polar vortex in the lower stratosphere and the chemical processing potential in 2019/2020. The corresponding papers are cited.

Other Scientific Comments / Questions: What do the authors mean by "classification" in the title? The term is typically used for grouping and comparing things with similar characteristics, but it is unclear what sort of classification is being attempted herein. Title has been changed, also due to the other comments and statements. The choice of word was not adequate, "comparison" would have been better.

Given the novelty of the ozone datasets used in this paper, I would like to see, especially in Section 2, more discussion of the TROPOMI data, the GTO-ECV data, and the relationships between them – especially in conjunction with Figure 7, which compares time series from GTO-ECV in previous years with that from TROPOMMI in 2019/2020. What are expected biases between the two datasets? Are the discontinuities in the GTO-ECV data that might result in biases between some of the earlier years, or in

non-physical trends? Most information is provided by Coldewey-Egbers et al. (2015; 2020). Some more explanations have been added in the revised manuscript. Besides, in the revision we have now also mentioned the results of an initial comparison of TOCs from TROPOMI and OMI (not published so far); it indicates that TROPOMI TOCs are slightly smaller (about -1%) than OMI TOCs.

Page 2, line 12: How is "the expected ozone value in austral spring" defined? Do you mean the value that would be expected at that time of year with no chemical loss? If so, how is that determined? Yes, it is the long-term (climatological) mean value, which is based on observations before 1980. The text is slightly changed ("climatological mean ozone value in austral spring, which was determined for the years before 1980 (. . .").

Page 2, line 25: Statements such as "...due to a strong and stable polar vortex in winter. . ." are too over-simplified, since there is (particularly in the much more dynamically active Arctic) no one-to-one relationship between vortex strength and temperature. We agree that the statement here is oversimplified! The half sentence has been deleted to avoid misunderstanding.

Page 3, lines 6-8: What is this statement based on? Given the similarity of LS temperature evolution to that in 2016 and later 2011 as the winter progressed, and the large chemical ozone loss and resulting low column in 2011 (coupled with the knowledge that dynamical variations that can reduce column ozone play a significant role even in the coldest Arctic winters, and the large interannual variability making winters as cold as or colder than 2011 very likely "sometime"), I see nothing unexpected about what happened in 2020! We agree, that the dynamical conditions found in NH winter/spring 2019/2020 are not unexpected. However, having on the one side a significantly reduced stratospheric chlorine content of about 15% (compared to the years around 2000) and on the other side detecting new record low TOC (below 220 DU) in the Arctic polar stratosphere over a longer period is still noteworthy. The text has been changed accordingly. Page 5, line 29: I think "to large parts" is too weak here – no chemical processing of any kind is needed to produce "mini-holes" as defined here.

We deleted the half-sentence containing this phrase as this is already addressed in the next sentence by referring to Millán and Manney (2017).

Page 6, lines 8–9: There is no reason to expect this, since (as per above discussion in the major points) the strength and coldness of the vortex are not necessarily closely correlated, and "an ozone-hole like pattern" in the sense of chemical loss driving the ozone morphology would never be expected before March because chemical loss is limited when the polar regions are in darkness. You are right! The word "expected" has been changed to "observed".

Page 6, lines 14–15: But you don't even show the polar jet (which would have to be zonally resolved to show where the polar vortex was) in relation to the TOC, and show nothing about it at a level that is appropriate to determine where the LS vortex is. See comment on lack of definition of polar vortex in major comments above. In Figure 3 (now new Fig. 4) we show the ERA5 monthly mean horizontal wind fields for January, February and March 2020. Maximum wind speeds are given. The figure indicates the persistence of the polar vortex (here at 10 hPa) in late winter and early spring 2020. In addition, the new Figure 2 shows PV at 475 K for particular days in March and early April (same dates as in Figure 1 showing TOC, to allow for a direct comparison).

Page 6, line 21: If you accept that this value is an accurate reflection of the similarity of chemical loss in the 2020 Arctic to that in the Antarctic, it would be helpful to point out that this is about 10% of the smallest Antarctic ozone hole area (in 2019) on record, and less than 5% of typical Antarctic values (see, e.g., Figure 1 in Wargan et al, 2020, accepted article online for JGR, https://doi.org/10.1002/essoar.10503445.1; and references therein). Good point! We discuss briefly the comparison with the Antarctic ozone holes in 2016 and 2019, and we will bring up this point again later in the discussion. See the updated Figures 3, 6, 8, and 9.

Page 9, lines 4–8: Per previous comments, cold winters with substantial ozone loss can have weak polar vortices (e.g., 2004/2005), and not all winters with large ozone

loss have unusually strong vortices. "Cold" and "strong" are not synonymous in relation to the polar vortex. OK, we got your point and we agree in principle, but what we have written in these two sentences is not wrong. Therefore, we would like to keep this part as is.

Page 9, lines 17–24: This paragraph seems irrelevant to this paper. If indeed 2018/2019 needs to be mentioned, this could be done in a sentence by simply citing one or more of several papers that have been published on that winter (e.g., Butler, et al, QJRMS, 2020, and references therein). We would like to keep this short paragraph as is because it shows nicely the differences. We already cited the paper by Lee and Butler (2020; published in Weather 10.1002/wea.3643) in this context. From our point of view this paper fits better than Butler et al. (2020) in QJRMS.

Page 9, lines 24–32: This paragraph seems largely irrelevant as well, and if the 2019 Antarctic ozone hole needs to be mentioned, that could be done by citing one or more of the several papers published on it (in particular the Wargan et al, 2020 paper mentioned above, which provides a detailed analysis of the dynamical and chemical mechanisms leading to the unusually small ozone hole in 2019; but there are also a couple of earlier references given therein). We do not think that this comparison is irrelevant. With respect to our reply above (regarding the Wargan et al. (2020) paper, we slightly revised this paragraph.

Page 10, line 27 and line 30: "From our point of view" is a statement that would appropriately preface an opinion, not a scientific statement. If the statements following these can be backed up with evidence, there is no reason to use this language; if they cannot, they should not be made in a scientific paper. The turn of phrase is deleted.

Page 10, lines 13–14: If this is intended to convey something beyond the point made in the previous sentence, a citation or some evidence should be given.. We assume that page 11 is meant here. The sentence is deleted.

Please also note the supplement to this comment:
https://acp.copernicus.org/preprints/acp-2020-746/acp-2020-746-AC4-supplement.pdf

---

## Referee Report (RR1)

Unfortunately, I think that this manuscript needs another revision. The "Major comment" and the comments under "General" were not fully addressed. In addition, the number of specific comments (new ones and old ones) adds up to more than 6 pages. I have also added 10 pages of suggestions to improve language, grammar etc.

**Title:**

I am much happier with the title now. However, while I am not a native speaker, I have the impression a native speaker would not have phrased it like this. Just "Record low ozone values over the Arctic in boreal spring 2020" sounds better to me. Or may be "assessment of"? Or "Record low […] spring 2020 compared to other winters"?

**General:**

**"I have to admit that I had more of a problem with the wording and readability in some places. Wording was quite awkward in some places, and sometimes the text could have been less confusing and better organized, it felt a little bit rushed in places. I have the impression that asking a native speaker to go through the text would help in many places."**

I have the impression that you misunderstood my first comment in the "General" section or confused that with my comment under the section "Major comment". This comment was not about phrasing things carefully and the use of the term "ozone hole". It was about readability, conciseness, correct English language and grammar, awkward phrasing etc. Unfortunately, that means that this comment was not addressed adequately.

Maybe I did not express myself clearly enough. But in fact, your manuscript is very hard to read. I would try to improve language, grammar etc. in the manuscript in your own interest.

Even though I am not a native speaker, I can tell that there are many places of awkward writing, or not using the correct English phrases and words. In addition, the text is often very convoluted and confusing, and contains many repetitions and filler words. You probably could shorten the manuscript considerably without loss of information. As a result, you would get a more concise and readable text.

I have read the manuscript again and have compiled a long list of concrete suggestions how you could change the text (see last section **"New comments part 2: language"**). But this list is certainly not exhaustive.

**Reply to my major comment:**

I appreciate your effort to phrase the script more carefully and I am happy that you acknowledge my major comment in your reply. The manuscript has certainly improved.

Unfortunately, you contradict what you write in your reply as early as in the first sentence of your revised manuscript. You have rephrased "atypical ozone hole feature" just to "exceptional ozone hole-like feature". I am sorry to say that, but for the majority of the readers (and certainly for me) that will mean too much the same as before.

May be I can illustrate that with the following (tongue-in-cheek) example: Imagine you see a bike but write, "I see a car". Somebody responds: "But this is a bike!". You reply, "Ok, you are right. This is a car-like thing", because you are big fan of cars. Somebody else says, "Now, that you say it, it looks a little bit like a car, and it is a vehicle after all". I hope that makes clear where the problem is.

I will not go into detail, why I would not call it either an "ozone hole" or "ozone hole-like feature". You got very detailed comments on this, which I do not need to repeat here.

Unfortunately, this problem is not restricted to the first sentence of the paper. Further occurrences are at **page 1, line 10, page, 1, line 14, page 6, line 29, page 7, line 22, page 7, line 31, page 11, line 7, page 13, line 5 and page 15, line 2**.

I would suggest writing "ozone minimum" or, if you prefer "pronounced ozone minimum". That would be fine and still get to the point in a clear statement. If you insist on calling it an "ozone hole-like feature", you unnecessarily provoke associations. As I already said in my original comments: Do not push the reader into a certain direction if you cannot back up this by facts.

Please understand that I am really not nitpicking here or trying to force my opinion on you. In my opinion, this is not something subjective, but really needs to be addressed. This was my only major comment, and I would be happy if this comment would be addressed not only reluctantly.

**From your reply: "(ii) the shape of the region of low TOC looks "ozone-hole-like""**

What do you mean by that? I think we agree that the area of the ozone minimum was only about 5% of the Antarctic ozone hole. Do you really refer to the geometrical shape here, i.e. the feature is roughly circular like the ozone hole? Does this have any relevance?

**Specific comments (I will refer to the original page and line numbers(!), so that you can quickly find the comment I am referring to)**

**Page 1, lines 11-12**

You say this has been changed. You did not really. See major comment.

**Page 1, line 14**

You say "In addition, … sentence is added to make clear … differences in the respective area and time". In contrast to your statement in the reply, you discuss area here but not time. Did you forget to include this?

**Page 1, line 16**

You say this has been changed. You did not really. See major comment again.

**Page 2, lines 5-10:**

This is certainly nothing I would insist on, but I have the reader in mind here. This is one of quite a few places which are either phrased quite awkwardly or where you will make the reader wonder why you write this.

**Page 3, lines 24-28:**

I think I have to make some serious comments to Farah here (only joking). But seriously, you make the reader wonder again why you write this. At least the part with the CDO tools makes the impression as if it would have been a challenge for you to calculate a simple mean. I am quite confident that this was not the case. Maybe you can keep the information that the daily mean was based on hourly values, but everything else is detail that is not needed. You can thank the authors of the CDO tools in the acknowledgments.

**Page 4, lines 26-28:**

The reason for my comment was that this is rather hard to read. May be I did not express this clearly enough.

**Page 4, line 28:**

Ok, you did change this, but not to the established terminology, as requested.

**Page 5, line 2**

Certainly, nothing I would insist on, but that does not change my opinion that the figure is not needed. You can see the circular shape easily in Figure 1.

**Page 5, line 3**

This is still misleading and ambiguous. The radiative cooling does not mainly result from the dynamical conditions. It is there because there is no sun in the polar night. It is present in every year. It is somewhat modulated by different amounts of radiatively active gases transported to the polar regions in different years, but I do not think you have this in mind here. Temperature variations are caused by adiabatic warming (and by the triggered additional(!) radiative cooling) by the BDC. I agree that you do not need to go into detail here, but the sentence should be correct. Suggestion: "The dynamical conditions in winter 2019/2020 with low planetary wave activity result in very low temperatures…" Only a slight change, but physically correct now.

**Figure 5 and 6: Please add…**

Why don't you add Antarctic data for Figure 7 (was Figure 6), too, as for the other figures (and as suggested by me)?

**Figure 5: Can you really learn…**

**(Either relevant information is missing or there is a problem with what is shown in Figure 5, now Figure 6)**

I think you have totally confused me here and I did not get it right in my original comment. Either, something must be wrong in the description in the figure caption or with the values shown the figure, or there is not enough information given to understand what you have done. This may carry over to the text on page 9 and 10 (new manuscript) and Table 1.

What exactly do the dots show in Figure 5 (now Figure 6)? You write "minima of monthly mean temperature". But thinking about it, this would only give one dot per month (you have the monthly means of polar cap temperature for 30 years, and then take the minimum, which is one value).

Or do you mean that you first take the monthly mean at all grid points for every year, and then look for the minimum in the monthly mean field in the longitude-latitude grid? That would be consistent with the data points in the plot, but would be a rather convoluted quantity, where I am not sure if it would make sense to look at the quantity. E.g., values would by definition be higher as the daily minima, and since the minimum is not always at the same position, values would smear out.

Or is it the monthly mean of the daily minimum values north of 50 degrees for each individual year? In this case, I have the impression that the dots are placed far above the position where I would have expected them. Take for example the blue dot for January. It seems that the daily blue values (small dots) for January are almost all placed below the big blue dot. But this can't be possible if the big blue dot is the mean of the small blue dots.

I think that some area with the daily range of values would be the quantity that the reader would expect here and would immediately understand (this is usually shown in the plots in the literature). If you first take the minimum of the monthly mean values, this is a rather convoluted quantity. It certainly does give you some impression about the range of the values, but is not as straightforward as it could be.

**Page 5, lines 11-12**

I do not think that you understood the problem here. I think it is mathematically not correct and therefore confusing. A sum like the sum here is always an approximation of an integral over time, and will always yield a value similar to an integral if done correctly. If you would take values every 12 h instead of every 24 h, you would get approximately twice the value by simply summing up. If you however do consider the units, you would divide by 2 in the case of 12-hourly values. That would give a similar value as before. The point is you are already implicitly assuming a value of "days" here, because you sum up values given every 24 hours. You probably do not want to depend your result on the number of values that you sum up in a given time period.

**Figure 1: Add contours for the 220 DU contour and the vortex edge. These are things that are really hard to see in a colored contour plot. And the 220 DU contour is really central for the discussion in your paper.**

You say this is not necessary in your reply. I would not have made this comment if I would not have large problems to discern things here, and I think many people would share this view. This is not only a continuous color scale, but it also lacks contrast in the relevant range. In this context, "dark purple" is subjective at the least. And it is easy to add a line contour to the plot (say, in white). This should be feasible and easy to do with whatever graphical package you are working with. I do not think I am asking for too much work here. Same applies for the vortex edge. The new Figure 2 helps, and this shows that it should be easy to add e.g. the 36 PVU contour to Figure 1 (in a color different from the 220 DU contour). You could omit the new Figure 2 then.

**Figure 6**

Sorry, but I do not see the qualitative difference to the other figures (Figures 3, 6, 8 in the new manuscript). I cannot follow your arguments. Again, I do not think I am asking for too much work here.

**Page 6, line 22**

No, they are not, see page 5, lines 11-12.

**Page 8, lines 11-15**

Mostly OK. Page 10, line 28 of the revised manuscript: "denitrification was much stronger than in 2011". That is not was the text says in Manney et al., 2020. My interpretation of the text is that 2011 was comparable in magnitude. Please change accordingly.

**Page 9, lines 12-14**

OK in the revised manuscript. But I do not understand your reply "The statement … was very clear that the spring 1997 was the coldest …" This was not clear at all in the original manuscript. There is no obvious and straightforward way to define the "coldness" of a complete winter. You can take minimum temperatures, VPSC, North Pole temperatures etc. and this will give different results.

**Page 10, line 10**

OK, but I do not think that 15% is "clearly higher". It is only 15%.  It would suffice to write "higher".

**Page 11, lines 11-14**

I certainly do not mind. It is your freedom as an author. My concern is just how the readers conceive this. This is all about readability and conciseness.

**Page 11, lines 17-19 etc.**

You say you deleted the expression "ozone hole". You did not. See major comment.

**Technical corrections**

**Page 8, line 1 and 3 (now page 10, lines 12 and 14)**

You did not change all occurrences. You can also just write "(PSC type 2; see" for "(PSC type 2, ICE-PSC; see"

**New comments (part 1: related to content, page and line numbers revised manuscript)**

**Page 3, lines 13-14:**

"However TOC below 220 DU have not been observed in these two years." This is not true. Have a look at your Figure 8. Please phrase correctly, e.g. add "for an extended time period in spring".

**Page 5, line 19:**

"In connection with Figure 1". Sorry, I cannot follow you. The sentence would read totally correct for me if you would delete this part. This seems to be superfluous and not correct.

**Page 6, line 33 to page 7, line 2:**

It is not clear what you compare to, add "than in the Arctic" or similar.

**Page 7, line 25:**

"not shown". This is not true anymore in the revised version.

**Page 9, lines 13-15:**

Looking at the plot, this is not really true. Daily minimum values for 2010/2011 and 2019/2020 are very similar in December and, to a lesser extent, they are also quite similar in January.

**Page 10, lines 6-7:**

That is possibly misleading. That depends on what quantity you are looking at. Vortex averaged loss was not so different, at least in my study. The maximum loss made the difference.

**Page 10, lines 19-20:**

Delete. There is an almost identical sentence only a few lines before (page 9, lines 13-15). In addition, the statement may not be quite correct (see there).

**Page 10, line 25:**

"due to heterogeneous reaction". I do not think this is true. Don't you just mean "by uptake of $HNO_3$"?

**Page 10, line 28:**

See "specific comments, page 8, line 11-15". I do not think this statement is correct.

**Page 11, lines 16-27:**

I found the information content of these lines to be small. This is well known from the literature and textbooks. Not that it would be wrong to mention this here, and it is all correct, but could you try and come a little bit more quickly to the point here?

**Page 12, lines 3-7:**

How do you compare 30 hPa North Pole temperatures from FU Berlin to the temperature metrics used in your study? These are not the same quantities, and 2019/2020 is not included in the FU time series. E.g., North Pole temperatures could be hampered by the fact that the polar vortex is not always situated over the North Pole. Wouldn't it be better to base your statement that "2019/2020 is outstanding … since … 1950s" on a comparison of the same quantity?

**Page 12, line 17 to Page 13, line 6:**

Again, it would help if you would come to the point here more quickly. It is totally ok to discuss this here, but this is a rather long and meandering discussion. The main topic of your paper is the 2019/2020 winter. While it is ok to compare to other winters, I do not really see the relevance of all the details given here. Where does this aim at? This seems like a repetition of facts from other sources.

**Page 13, line 11:**

"although the dynamic features are similar". What exactly do you mean by dynamic features here? I think this statement is too subjective and should be deleted.

**Page 13, lines 16-18:**

This has already been discussed. Delete. In addition, I think this statement is not really correct, see comments I have already made.

**Page 13, lines 20-24:**

Delete. You have already discussed that in preceding section in detail. You only repeat thinks here you have already said.

**Page 13, lines 24-26:**

Delete sentence starting with "It is obvious…" You have stated exactly the same a few lines above (lines 16-18). Lines 16-18 were already a repetition. In addition, the statement is not correct.

**Page 13, line 27-30:**

Delete. You have already said this and only repeat things here.

**Page 13, line 33:**

Probably you mean "in particular" and not "especially". However, that does not really fit either. You could delete "especially". However, here is a deeper problem. In fact, you are speculating here and cannot really back this up by facts. I would suggest either to cite a reference here or to replace "especially" by something like "possibly" or "probably".

**Page 14, line 1-2:**

This slipped through my attention in the first review. Again, this cannot really be backed up by facts. While it is plausible, you do not do an analysis of dynamical and chemical contributions to the observed ozone column. You need to phrase that more carefully.

**Page 14, line 7:**

Delete "only". I think I already made a comment that you can also learn something from the past here.

**Page 14, line 8:**

Delete "more or less". Either it is cooling or not. This is not really a scientific statement.

**Page 29, line 4:**

As said in the specific comments, something is wrong with the "minimum of the monthly mean temperatures", or there is not enough information given.

**New comments (part 2: language, grammar etc., page and line numbers revised manuscript)**

**Page 1, lines 23-24:**

"emphasizes the noteworthiness" does not sound like good English to me. "highlights" or "underlines" is probably better. "noteworthiness" sounds strange here. Maybe "highlights the unique evolution of…" or something similar.

**Page 2, line 4:**

Delete "especially". You probably mean, "In particular, unusually low ozone…"

**Page 2, line 7:**

Suggestion "…are found in the stratospheric ozone layer…"

**Page 2, line 21, Page 13, line 24, Page 29, line 11:**

Change "ICE-PSC" to "ice PSC"

**Page 2, line 23:**

Replace "for instance chlorine" by "chlorine and bromine". It is only these two species.

**Page 2, line 25:**

Would help to replace "ozone depletion begins" by "ozone is depleted by catalytic photochemical cycles" or something similar. Only a few words more, but some more relevant information.

**Page 2, line 27-30:**

This is a very long sentence and hard to read. Maybe you could just shorten "… in response to the Montreal protocol, 1987, and its amendments…"

**Page 2, line 31:**

"atmospheric content" seems not the right phrase for me. I suggest "atmospheric burden" or "atmospheric concentrations" etc.

**Page 3, line 5:**

Change "heavy ozone depletion" to "severe ozone depletion"

**Page 3, line 7:**

Delete "clearly"

**Page 3, line 10:**

You could delete "as we will see in the upcoming analysis"

**Page 3, lines 10-13:**

The sentence is somewhat convoluted and does not read well. Suggestion: "Comparable dynamical conditions in the Northern stratosphere in spring were noted in the literature for 1997 (references…) and 2011 (references…)"

**Page 3, lines 14-16:**

This is phrased awkwardly. Suggestion "Although the dynamical conditions in winter and spring 2019/2020 were unusual, they are in the expected range of stratospheric dynamical fluctuations in Arctic winter and early spring (e.g., Langematz et al., 2014)."

**Page 3, lines 28-29:**

Again, this is phrased awkwardly. Suggestion: Replace "It allows an evaluation of the current situation by the comparison with similar dynamical conditions in Arctic spring of other years" by "We compare the current winter to winters with similar dynamical conditions in Arctic spring…"

**Page 3, line 31:**

Replace "far away from the usually observed Antarctic ozone hole" by "far removed from the conditions usually observed in the Antarctic ozone hole".

**Page 4, line 10:**

Delete "For our investigations"

**Page 4, line 17:**

"The focus is laid on stratospheric zonal winds, polar temperatures and potential vorticity (PV)." I think you do not need that sentence.

**Page 5, line 18:**

"turned out to be persistent" is phrased awkwardly. Suggestion:  "Arctic winter and early spring 2019/2020 showed a persistent stratospheric polar vortex with strong zonal winds from mid-December until early April."

**Page 5, line 20-21:**

Phrased awkwardly. Suggestion: replace "representing the dynamic state of the lower stratosphere with respect to the position and strength of the polar vortex." by "and shows the position and strength of the polar vortex."

**Page 5, line 21:**

Change "PV-gradients" to "PV gradients"

**Page 5, line 24:**

Phrased awkwardly. Suggestion: "Figure 3 shows strong zonal mean zonal wind speeds at 60°N, 10 hPa (about 30 km altitude) in the ERA5 data (magenta line and dots in the figure), …"

**Page 5, line 25:**

Replace "which are high with respect to" by "which are higher than"

**Page 5, line 26-27:**

Delete "very much". Do not exaggerate.

**Page 5, line 27:**

Delete "the respective"

**Page 5, line 31 to page 6, line 3:**

I do not think that this sentence, which is very hard to read, does convey any information that is useful in the context of this paper. Please delete.

**Page 6, line 6:**

Delete "clearly"

**Page 6, line 10:**

Change "lower stratosphere temperatures" to "lower stratospheric temperatures"

**Page 6, line 11:**

The 50 hPa level is not a height range. It would be better to say, "which is inside the height range". I would also say "important for ozone depletion" and not "of vital importance for ozone depletion". Better English in my opinion.

**Page 6, line 14-15:**

You can delete "(i.e. the Cl activation threshold at this altitude, see for instance Figure 4-1 of Chapter 4 in WMO, 2018)" now. You now have that in the introduction. That was exactly the reason why I was asking to write it in the introduction. You can streamline the text here.

**Page 6, line 21-22:**

That could be shorter now: "This led to conditions allowing the formation of NAT-PSCs at 50 hPa for about 3.5 months (see Figures 6 and 7)." You introduce the abbreviation in the introduction.

**Page 6, lines 24-26:**

You can shorten this significantly now. Again, that is why I asked for this in the introduction.

**Page 6, lines 31-32:**

This is phrased awkwardly. I think you should delete "exemplarily" and start with "In Figure 3 and 6, corresponding…"

**Page 7, line 26:**

You could delete "indeed"

**Page 8, lines 6-7:**

Shorter: "The maximum area with TOC below 220 DU was 0.9 million $km^2$ (= $0.9 \cdot 10^{12}$ $m^2$) on March 12 (Figure 1)."

**Page 8, line 7:**

You very probably mean "polar vortex area" and not "polar vortex".

**Page 8, lines 7-8:**

Phrased awkwardly. Suggestion: "This is in the order of 4% of the polar vortex area at the 475 K isentropic surface inside the 36 PVU contour (e.g., Wohltmann et al., 2020)."

**Page 8, line 11:**

Change to "minimum TOC is clearly higher" (singular).

**Page 8, lines 18-19:**

"spring" is mentioned twice in the sentence. You could omit "in spring".

**Page 8, lines 24-26:**

This is an overly complicated sentence for a simple fact. Much shorter without loss of information: "The evolution of the polar vortex at 10 hPa and the minimum temperatures at 50 hPa are commonly used to examine the dynamical state of the stratosphere (see e.g. Lawrence et al., 2020)."

**Page 9, line 4:**

It is not clear what "see below" refers to.

**Page 9, line 5:**

Probably a "," would help: "…are similar, reaching…"

**Page 9, lines 6-9:**

This is a very long sentence. It would help to split it into two and to shorten it.

**Page 9, line 10:**

Delete "the respective"

**Page 9, line 11-12:**

Suggestion: "The temporal evolution of the observed daily minimum temperatures at 50 hPa is shown in Figure 6."

**Page 9, lines 12-13:**

Phrased awkwardly. Suggestion: "The minimum temperatures were below the threshold temperature for the formation of NAT PSCs (195 K) in February and March of all three years."

**Page 9, line 13-15:**

Phrased awkwardly. Suggestion: "Minimum temperatures at 50 hPa in December 2019 and January 2020 were lower than the minimum temperatures in December/January 1996/1997 and December/January 2010/2011 most of the time." (but see comments above that this not really true)

**Page 9, line 16:**

What do you mean by "indicating the characteristic of"? I am clueless.

**Page 10, line 1:**

It must be "of the two winters 1996/1997 and 2010/2011". "to" makes no sense here.

**Page 10, line 2:**

You probably mean "in particular" and not "especially"

**Page 10, line 3:**

Phrased awkwardly. Suggestion: "Severe chemical loss was observed in spring 2020 (Manney et al., 2020)."

**Page 10, line 4:**

Change "which was mentioned in" to "according to" (or just cite the studies at the end of sentence and skip this part of the sentence).

**Page 10, line 13:**

Change "was determined with" simply to "was"

**Page 10, line 17 and 18:**

You could easily delete "To summarize" and "Our analyses show"

**Page 10, line 20:**

You could delete "It is worth mentioning that"

**Page 10, line 22:**

Delete "in" at start of line. Seems to make no sense here.

**Page 10, line 22:**

You could delete "Having"

**Page 10, line 23:**

"more efficient". Compared to what? You could just delete "more".

**Page 10, line 23:**

Had to think a moment about what you mean by "they". I would just write "PSCs".

**Page 10, line 24:**

You probably mean "in particular" and not "especially"

**Page 10, line 26:**

Suggestion: "enabled a period of ozone depletion that was longer than usual"

**Page 10, line 27:**

Unnecessarily complicated sentence (and not phrased very well). Just start with "Manney et al. (2020) analyzed…"

**Page 10, line 30:**

I was confused here over reading "July" and "June" for the Northern hemisphere, until I realized that you are talking about the seasonal evolution of ozone over the period of a complete year. Can you try to phrase that a little bit differently?

**Page 10, lines 30-32:**

A very long sentence. Please split into two sentences.

**Page 10, line 32 to page 11, line 2:**

Unnecessarily long and complicated sentence.

Suggestion: "Typical features of a strong polar vortex can be observed in February 1997 and February 2011, with low TOC values in the vortex and relatively high TOC values in the collar region of the polar vortex (not shown)"

**Page 11, lines 3-6:**

Another very long sentence. You could delete "An important point to be mentioned is that". You could also delete "and which were then followed by a typical chemical ozone loss". What is a typical loss? Does this add any important information?

**Page 11, line 7:**

"shows" and not "is showing" is probably the correct tense.

**Page 11, line 8:**

Delete "clearly"

**Page 11, lines 11-14:**

Again, a very long sentence. Can you split this in two?

**Page 11, lines 28-30:**

Phrased awkwardly. Suggestion: "The winter 2019/2020 shows an extraordinary dynamical situation with a persistent strong and cold polar vortex over the complete winter season, when compared to the last four decades (the period of the ERA5 dataset)."

**Page 11, line 31:**

You could delete "our dynamical analyses based on"

**Page 11, line 31:**

It is not clear what "*the* historical data set" refers to. Better, "An analysis of historical data was…"

**Page 12, line 5:**

Phrased awkwardly. Suggestion: "Temperatures in January 1997 were near the climatological mean value."

**Page 12, lines 5-6:**

You could delete "In combination with our research results".

**Page 12, line 7:**

Delete "for instance"

**Page 12, line 10:**

Delete "their investigations of"

**Page 12, lines 11-12:**

Delete "(i.e. in the Berlin analysis it ranked second after 2019/2020)". You stated the same a few lines before.

**Page 12, lines 12:**

Phrased awkwardly. Change "(i.e. in the Berlin analysis it indicated as a cold winter, but not extraordinary)" to "(indicated as a moderate cold winter in the Berlin analysis)"

**Page 12, lines 12:**

Change "turned out to be" to "were"

**Page 12, lines 13-15:**

Phrased awkwardly. Suggestion: "The slightly different results for the record years indicate that results depend on the considered quantity" (or "considered meteorological variable").

**Page 12, line 15:**

Delete "Further it is clear that the"

**Page 12, lines 15-16:**

Phrased awkwardly. Suggestion: "Nevertheless, it is obvious that the winter 2019/2020 was one of the coldest winters in the last 65 years, and that it showed an exceptionally strong and stable polar vortex."

**Page 12, line 17:**

Delete "We note that"

**Page 12, lines 17-19:**

I would suggest splitting this long sentence into two.

**Page 12, line 23:**

Change "(around the long-term mean)" to "(similar to the long-term mean)"

**Page 12, line 25:**

Phrased awkwardly. Suggestion: "The Southern hemisphere spring seasons of 2002 and 2019 provide two additional examples for the importance of stratospheric dynamics in the development of low ozone columns"

**Page 12, line 28:**

"The other example happened in September 2019." Phrased awkwardly. Just delete and continue with "In 2019, the polar vortex was…"

**Page 12, line 29:**

Throughout the paper, you talk of "zonal mean zonal wind", but here you say "zonal mean west wind".

**Page 12, line 30:**

Change "happened" to "was observed"

**Page 12, line 32:**

Change "Afterwards" to "After this event,"

**Page 13, line 8-9:**

Change "lower stratosphere temperatures" to "lower stratospheric temperatures"

**Page 13, line 10:**

Delete "that". Grammatical error.

**Page 13, line 11-12:**

You could split this sentence into two.

**Page 13, line 13-14:**

I would just say, "…were similar to the conditions in early spring…"

**Page 13, line 14:**

Delete "Further". Or write at least "Furthermore"

**Page 13, line 15:**

I would suggest "Minimum TOC values were below 220 DU for several days in March 2020, although…"

**Page 13, line 16:**

Delete "clearly"

**Page 13, lines 16-18:**

Delete "Our comparisons show that especially"

**Page 13, lines 18-19:**

Change "all the time below 195 K" to "below 195 K most of the time"

**Page 13, line 19:**

Add "," following "In this context"

**Page 13, line 20:**

Delete "Our analyses show that"

**Page 13, line 30:**

It is not clear what "However" does refer to. Delete.

**Page 14, line 3:**

Phrased awkwardly. Suggestion: "Record low stratospheric ozone values over the Arctic in 2020 are not an unequivocal result of climate change."

**Page 14, line 4:**

Phrased awkwardly. Suggestion: "The dynamical situations in February and March of 1997, 2011 and 2020 were similar."

**Page 14, line 4:**

Delete "Beyond that". It is not clear what you refer to and it is confusing to read.

**Page 14, lines 6-7:**

Awkward phrasing. "is possible" sounds like that would be a surprise, but we expect cold winters from time to time. Suggestion: "The NH winter 2019/2020 is a perfect showcase for a Northern winter with low planetary wave activity, a strong and stable vortex and low temperatures." (replaced "less" by "low", since you do not compare anything here).

**Page 14, line 12:**

Phrased awkwardly. Suggestion "… showed that cold Arctic winters may possibly get colder in the future."

**Page 14, line 18:**

Delete "respective"

**Page 14, lines 23-25:**

I think that I have already made a comment that this seems strangely out of context. I would really suggest deleting this.

**Page 14, line 27:**

Why did you add "consistent" to my suggestion? That sounds strange. I would delete that. Nobody would expect an inconsistent description here.

**Page 14, line 29:**

"in the vicinity or below 220 DU". Phrased awkwardly. Suggestion: "Record low TOC values of 220 DU and less were detected…"

**Page 14, line 29:**

It has to be "large" for "larger" and "extended" for "longer". You do not compare anything here.

**Page 14, line 30:**

Phrased awkwardly. Suggestion: "2019/2020 is compared…" or "The situation in 2019/2020 is compared…"

**Page 14, line 31 to Page 15, line 2:**

Awkward phrasing. Suggestion: "We have used recent meteorological data from ERA5 and recent total ozone column data of GTO-ECV (based on…) in combination with TROPOMI onboard Sentinel-5P."

**Page 15, lines 2-5:**

Extremely long sentence. Please split into two.

**Page 15, line 4:**

It seems the hyphen is not at the correct position. It should be "…mid-October) – and TOC…"

**Page 15, line 5:**

"were not observed before over a period of 5 weeks". It should become clearer that you refer to past years and not to the same winter here.

**Page 15, line 5-6:**

You could delete "The results of our study pointed out that"

**Page 15, Line 7:**

"supporting". Is this really what you want to say? You probably want to express that ozone depletion was more severe than in other years. Maybe "leading to enhanced ozone depletion compared to other years" or similar.

**Page 15, line 8-9:**

Shorter and not so convoluted: "The special dynamical situation in winter 2019/2020 is the cause for the significant reduction of the TOC in spring 2020 …". This removes also the ambiguity what "despite" refers to.

**Page 15, lines 11-14:**

Very long and complicated sentence. Suggestion: "Numerous studies of the 2019/2020 winter season can be found in a special issue of Geophysical Research Letters and Journal of Geophysical Research – Atmosphere (e.g., Manney et al., 2020; Wohltmann et al., 2020, Lawrence et al., 2020; Grooß and Müller, 2020)." Note that I removed statements that are only true at the moment of your writing. Soon, nobody will care that some of the studies were not published at the time of writing.

**Page 15, lines 18-20:**

Phrased so awkwardly that I had to read it several times to get the meaning. Suggestion: "However, in winters with a cold and stable polar vortex, a persistent region of low TOC might also develop in the Northern hemisphere in the future again."

**Page 15, line 21:**

You probably do not mean "documented" but "considered"

**Page 15, line 22:**

"enable well founded scientific explanations of special ozone features" is phrased awkwardly. Suggestion: "Continued monitoring of ozone with a suite of instruments will be key to understand the future development of Arctic ozone" or similar…

However, this rephrasing would also mean to change the following sentences a bit.

**Page 25, lines 6-7:**

Suggestion: "The grey line is highlighting the 36 PVU contour."

**Page 26, line 6, Page 29, line 7, Page 31, line 14:**

I would move "(attention: the respective data are shifted by six months)" to a separate sentence. Delete "attention" and write "Southern hemisphere data are shifted by six months."

**Page 29, line 10:**

Change "broken" to "dashed"

**Page 31, line 11:**

"Note the …." I would not write that in the figure caption without more context. I think this belongs into the main text.

---

## Referee Report (RR2)

**Review of "Evaluation of record low ozone values over the Arctic in boreal spring 2020" (revised from "First description and classification of the ozone hole over the Arctic in boreal spring 2020") by Dameris et al.**

Gloria L Manney, manney@nwra.com

**Overview Comments:**

This revised paper is, for the most part, scientifically sound, and provides a different view using new datasets of the evolution of total ozone column (TOC) in 2019/2020; the material is thus ultimately appropriate for publication in ACP. In the revision, the authors have gone a long way towards addressing several serious concerns in the reviews of and SCs on the initial ACPD version, but they have not entirely succeeded in some cases, as detailed below. In addition, there are some changes that should be made to the figures and text that should not be difficult or consuming of time or other resources but would have great value in making the main points of the paper more clear. I recommend publication in ACP if these concerns are addressed.

I do find the overall focus (what you might call "balance") of material in this paper somewhat problematic, because two already published papers (the Lawrence et al JGR paper and the Wohltmann et al GRL paper; these are augmented by complementary information in the Manney et al 2020 GRL paper) describe the meteorological situation in the lower stratosphere (where most relevant to chemical ozone loss) and its relationship to other Arctic winter/spring seasons with extremely strong and/or cold polar vortices much more completely that does this paper. All of these papers detail the comparison with 2011; in addition, Lawrence et al compare with 1997, and all of these papers also compare with 2016, which is notable for having an extended period, primarily in Jan/Feb, with the record cold of any Arctic winter in the past approximately 70 years (e.g., Manney and Lawrence, 2016, ACP; Matthias et al, 2016, GRL), the most denitrification and dehydration on record, and chemical ozone loss as rapid as or more rapid than that in 2011 (or 2020) until the early vortex breakup in March (Manney and Lawrence, 2016, ACP; Khosrawi et al., 2017, ACP; Johansson et al., 2019, ACP). In contrast, while there is some material on TOC in the papers published so far on the 2019/2020 Arctic winter (including comparisons of OMI and ground-based data with climatology in Bernhard et al, 2020, revised and resubmitted with very minor revisions for GRL, original ESSOAr link: https://doi.org/10.1002/essoar.10504414.1, a paper focusing primarily on the associated UV anomalies; and analysis including TOC comparisons with other winters using the CAMS reanalysis in Inness et al, 2020, https://agupubs.onlinelibrary.wiley.com/doi/10.1029/2020JD033563), the current manuscript does offer a different view of this with long-term comparisons using different, and relatively new, datasets. Thus I would strongly encourage the authors to rebalance the paper so as to focus less on the description of the meteorology (which could, for the most part, be described sufficiently by citing published material that was readily available well before this paper was initially submitted) and more on the impacts of that on TOC and clearly detailing how those impacts are seen in the TOC data that they show. I do, of course, realize that, with the special

issue, previously published material is a "moving target" and with a special issue (which I like to think of publications in other journals as informally contributing to), a certain amount of approximate duplication is inevitable -- so I hope it is clear that I am not asking the authors to remove all of the dynamical material (even where I do point out below in specific comments that a paper has covered the point already), just to cite the published work appropriately (which is already much improved, though not perfect, in the revised paper) and to focus less on that and more on the parts of this manuscript that are unique.

In the next section, I make comments on some points raised in the initial reviews and SCs that I don't think the current revision completely addresses (where I don't make comments, I either think the authors' responses are adequate or that I do not have anything to add that cannot be better evaluated by the authors of the initial reviews). Following that are some more specific comments based on the revised manuscript.

**Comments Related to Author Responses To Initial Reviews/SCs:**

All of the reviews and SCs questioned the suitability of using the term "Arctic ozone hole". While the revised manuscript is improved in this respect, in particular in presenting the TOC in the Arctic in the context of that in the SH, which helps greatly in conveying the ways in which the TOC values/patterns in the Arctic in 2020 did and did not resemble those in the Antarctic. However, there are several places (including in the abstract) where the authors have simply replaced "ozone hole" with "ozone hole-like pattern", which in my opinion can still be misleading and does not really address the fact that dynamics (e.g., very low temperatures in the vortex in cases where the vortex and the temperatures are very concentric) could in principle produce a pattern that looks superficially like an ozone hole even in the absence of any chemical processing. Of course, in practice what happens is that dynamical and chemical mechanisms reinforce each other when there are widespread low temperatures in the vortex. I think "ozone hole-like" (which, if you were going to use it should be "ozone-hole-like" since all together it is one compound adjective) really does not much change how the reader sees it, and thus the term should be avoided, especially in the abstract.

Regarding the focus on 10 hPa winds (now Figures 3 and 4), in relation to the comment in my SC about their lack of relevance to this paper, and also Ingo's comment that Figure 3 (now Figure 4) was redundant, I do not think the authors' response is adequate. They note that Lawrence et al (2020) also show zonal mean winds; however, Lawrence et al are giving an overview of the polar vortex throughout the stratosphere and its relationship to numerous other phenomena, and are using zonal mean winds in relation to common definitions of "strong vortex" and "weak vortex" events that are used for examining stratosphere / troposphere dynamical coupling. In contrast to this, the focus of this paper is on TOC and the chemical and dynamical processes in the lower stratosphere (e.g., near 50hPa) that lead to low TOC via chemical ozone depletion. In the section in Lawrence et al that focuses on lower stratospheric polar processing and ozone loss, they (because 10hPa winds in any view, as well as zonal

mean winds at any level, give little information relevant to that) use diagnostics that are vortex-centered and that are at levels that are in the altitude region where these processes take place. Figures 3 and 4 (new numbering) do not add any information that is relevant to the focus of this paper, but simply serve as a distraction or misdirection. I think they should be deleted. If the authors believe it is necessary to include interannual comparisons of diagnostics of vortex strength, those should be diagnostics that capture vortex strength at the levels where the vortex strength is relevant to the evolution of TOC (e.g., centered near 50 hPa if on isobaric levels, near 450--550K if on isentropic levels). In fact the authors' response to Ingo's comment (that what is now Figure 4 illustrates the nearly circular shape of the vortex) is true only for the middle stratospheric vortex, since the shape of the vortex commonly varies greatly with height and how it varies is different every year -- thus this tells us little, if anything, about the shape of the vortex in the lower stratosphere. (Per my overview comment above, such diagnostics have been more thoroughly covered by Lawrence et al, and thus referring to that paper for this material may be sufficient to make the points about the vortex strength in relation to other years that are important for this paper.)

With regard to the authors' response to Ingo's comment re Page 4, lines 26--28 (regarding wave activity) in the original manuscript, a better response to this would be to make this point by referring to Lawrence et al (2020), who show and discuss wave activity / fluxes / propagation / reflection and its variations throughout the winter in more detail (note that the GSFC people who produce the information on the website currently mentioned in this manuscript are co-authors on the Lawrence et al. paper, meaning that discussion has been vetted by them and is consistent with what they post on their website).

I agree with Ingo regarding Figure 5; it is common to show the years that are not highlighted as a grey envelope (with an indication of the range and standard deviation) because it conveys more complete information about the interannual variability of the daily values. Diagnostics based on the monthly mean values do not do that. The authors have used a format like that commonly used in Figure 8 (new numbering), and this would be a more informative way to show the comparisons with years that are not highlighted in other figures.

In relation to my concern about showing the lower stratospheric vortex structure, and Ingo's request to add a 220DU contour and vortex edge contour to the plots in Figure 1, I think the new Figure 2 does help with the vortex definition, but also think Figures 1 and 2 should be combined -- the PV and column ozone maps could be shown side-by-side, if desired, but whether or not the PV maps are shown, the 220DU contour (the color scale is not appropriate for the reader to be able to distinguish a particular color as the authors suggest) and a vortex edge contour (e.g., an appropriate PV value) should be overlaid on the TOC maps. It would also be helpful to add a 50-hPa 195K temperature contour to illustrate the relationship of the vortex to the cold region (as I recall, it was particularly concentric in this past winter, which is relevant to TOC morphology, especially before much chemical loss has occurred).

I agree with Ingo that Figure 6 should show the comparison with the other years. And, per my comments above, think it would be much more useful if the other years were represented in a manner similar to that in Figure 8 (new numbering).

Regarding the authors' response to the question by referee #2 about P5L3 (re radiative cooling and dynamical processes), the authors' response is ambiguous and could be misleading. In absence of dynamical heat fluxes, lower temperatures lead to less radiative cooling because they are closer to radiative equilibrium. So the question here is really the balance of that with the reduction in warming because of reduced planetary wave activity (i.e., dynamical heat fluxes). The wording of the statement in the revised paper is likewise ambiguous and should be modified to clarify this point.

Regarding my comment about page 9, lines 4--8 (relation of cold and strong vortices), I still believe this is misleading and should be modified. Yes, you can have these conditions. But you can also have weak vortex / strong mixing / substantial ozone loss, as was the case in 2004/2005.... And in 1997, even after the temperatures became unusually low, the vortex was never remarkably strong (and was remarkably weak -- but only in the lower stratosphere -- earlier in the winter) (Manney et al 2011, Nature; Lawrence et al 2020).

**Other Comments / Questions on Revised Manuscript (in order of appearance in paper, not importance):**

Page 1, line 21, and abstract in general: The point about the different mechanisms for the low TOC in 1997 vs the other two years compared (that is, the much smaller chemical loss in 1997) should be made in the abstract.

Page 1, line 26--27: "larger" than what? Presumably than in other Arctic winters in the first usage and in the Antarctic than in the Arctic in the latter -- but since the same wording is used in two different ways, you need to be explicit about what you are comparing to in each case.

Page 2, lines 1-2: It seems odd to me to make a general statement like this and give only a reference that discusses three instruments measuring column ozone. What about the numerous instruments with vertically-resolved measurements of multiple species (which are also critical to fully monitoring ozone loss/recovery and the processes involved)?

Page 2, lines 16--18: There are also direct effects of lower temperatures, and a relationship to higher, colder tropopauses, that work in the same direction (see SI in Manney et al, 2011, Nature, and references therein, in addition to references you already give later on tropopause heights).

Page 2, lines 20--21: This should be reworded to make clear that the threshold temperatures are approximate values that depend on $HNO_3$ and $H_2O$ concentrations, and that there are

several other types of particles (e.g., STS, etc) that form at temperatures similar to those of the NAT particles.

Page 2, lines 21--22: This sentence (contrasting NH and SH) should be moved to the end of the paragraph, after the description of the chemistry, so that it doesn't interrupt the description of the steps leading to ozone loss.

Page 3, lines 6--8: At this point in the paper, no evidence has been presented as to whether this is due to chemical ozone loss. Therefore, it is premature to make this statement assuming it is related to chlorine-catalyzed chemistry.

Page 3, line 9: "stable" is not an appropriate word here, as it has a specific formal meaning in relation to the dynamical stability (e.g., barotropic or baroclinic instability) of the flow; "quiescent" or "undisturbed" would be appropriate terms.

Page 3, lines 16--19: The papers cited here are all, with the exception of Tegtmeier et al, primarily chemistry papers, that is, they discuss the links of particular dynamical conditions to chemical loss. It would be worth citing some of the papers that discuss direct dynamical mechanisms in addition to those focused on in Tegtmeier et al (see, e.g., references in Manney et al, 2011, Nature, SI).

Page 3, lines 22--24: Instead of this detail / URL, and in addition to Wohltmann et al, please cite Bernhard et al (2020), submitted to GRL; this paper details column ozone anomalies in 2020 from OMI and from ground-based measurements and the corresponding UV anomalies. (Since this paper details TOC anomalies in different datasets than the ones used here, there are probably a few other places in this paper it could be cited and the consistency of their results with this paper mentioned. The same is true for comparison of TOC results with those in Inness et al. (2020).)

Page 3, line 27, and page 4, lines 2--4: As noted above, a comprehensive (much more so than in this paper) description of the dynamical situation in 2019/2020 winter (also compared with 1996/1997, 2010/2011, and 2015/2016) is already published in Lawrence et al (2020).

Page 4, lines 13--14: From "using the CDO" to the end of the sentence should be deleted, or, if you feel it is very important to give this detail, moved to the "Data Availability" section.

Page 4, line 25: Using "less than" and "up to" with signed values is a bit imprecise, technically it should say, for example, "less than +1% or more than -1%". It would be best to rephrase this so you talk about the magnitude of the bias and standard deviation (which isn't a signed value to begin with) rather than stating a signed value. I also fail to see why you need to give a range when it is prefaced by "up to" -- just say "up to 2.5%".

Page 6, lines 17--18:  The results of Lawrence et al and Wohltmann et al are more comprehensive than those shown here, so it might be sufficient to replace the minimum temperature plot (Figure 6) by a brief description of their results with the citations.

Page 6, line 25:  Dameris 2010 is a rather obscure reference to cite for what is textbook material.  In addition to Solomon 1999 (or instead of in this case), I would suggest Chapter 7 of the 2000 textbook "Chemistry and Physics of Stratospheric Ozone" by Andrew Dessler.

Page 6, line 33 to page 7, line 2:  Should cite Wargan et al (2020) here.

Page 7, lines 13--15: Manney et al 2011, Nature, also show the impact of tropopause height variations on column ozone, comparing 1997 to 2011.

Page 7, line 19: The statement "...the polar vortex existed already in late November and early December 2019" should be compared / contrasted to the other years considered here (this could be done very briefly by citing Lawrence et al 2020, who contrast the early development of the vortex in fall 2019 with other years.

Page 7, lines 22--23:  This is a good example of a place where it is particularly inappropriate to say "an ozone hole-like pattern".  In January, there has been little chemical ozone loss (almost none in most years) so the pattern of low ozone inside the vortex is primarily related directly (dynamically) to the low temperatures and concentricity of the cold region with the vortex. Even in July (+6mo) in the SH, the "ozone hole-like pattern" is mostly due to dynamical effects of low temperatures -- it is generally mid-July before the chemical loss signature overwhelms the dynamical ones.  It is not appropriate to call every large low ozone region within the polar vortex an "ozone hole-like pattern".

Page 7, lines 24--25:  As discussed above, the 10hPa winds provide no information about the strength, size, or shape of the lower stratospheric vortex.  In addition, rather than saying "(not shown") you could cite Lawrence et al (2020) for strong PV gradients.

Page 7, lines 27--30: This could be replaced by citing Wohltmann et al (2020) and Bernhard et al (2020).

Page 7, lines 31--32:  This ("strong horizontal gradient in the vicinity of the polar jet with strongest zonal winds") is not shown in any of your figures.

Page 8, lines 22--26:  Other dynamical effects that vary interannually (direct effects of low T, tropopause variations) could also be mentioned here, with appropriate references as already suggested above.

Page 8, line 27 through page 9, line 9: This paragraph is again discussing middle-stratospheric fields as if they were (1) relevant to the lower stratosphere and (2) had the same relationship to the conditions in the lower stratosphere in each year.  Neither of these is true.

Page 10, lines 6--7:  This is not true.  Manney et al and Wohltmann et al found that ozone loss was very similar in the two years. Ozone values were lower in 2020 because chemical loss started early and possibly because of less replenishment by descent and/or less mixing.

Page 10, lines 8--29:  The first paragraph here is an example where examining V_psc (or V_psc / V_vort) would provide a more complete picture.  Both Lawrence et al (2020) and Wohltmann et al (2020) do this.  These paragraphs could be condensed in light of that published information.

Page 11, lines 5--6:  Please clarify what you mean by "typical" here.  The ozone loss in 2011 was not typical, rather it was "unprecedented".

Page 11, lines 21--22:  This is too oversimplified (see previous comment on radiative heating vs dynamical heat fluxes).

Page 11, line 23 through page 12, line 16:  Could be condensed, since this information content is already in published papers.

Page 12, line 17 through page 13, line 6:  These two paragraphs seem tangential to the focus of the paper, and, since they are entirely discussing results shown in already published papers without making any cogent point about the relevance to this paper, seem more of a distraction than anything else.

Page 13, lines 7--12:  This has already been discussed in Wohltmann et al (2020) and thus could be condensed or removed.

Page 13, lines 13--26:  This has already been discussed in Lawrence et al (2020) and thus could be condensed or removed.

Page 13, line 30:  "However" is not appropriate here -- the "extended phase of active stratospheric chlorine" leads to the "substantial ozone depletion", whereas with "However" you are saying that the latter is in contrast to the former.  (Note also that neither of these is a result of this paper, though both are shown by Manney et al, 2020, and the latter by Wohltmann et al, 2020.)

Page 14, line 3: No one has suggested that it was demonstrably due to climate change. Wohltmann et al (2020; and to a lesser degree Manney et al, 2020) have also already discussed this.

Page 14, lines 23--25:  This statement does not appear to be related to anything else in the paper and seems completely out of place.  It also doesn't follow from anything shown in this paper.  I suggest deleting it.

Page 15, line 4:  It isn't clear what "and TOC values below 220 DU are seen for up to about four months" is in relation to here.  Is this for the Antarctic?  For the Arctic in extreme winters?

Page 15, lines 11--14: Add Bernhard et al (2020), DeLand et al (2020; https://agupubs.onlinelibrary.wiley.com/doi/abs/10.1029/2020JD033271; this discusses OMPs PSC measurements in 2020 and compares them to Antarctic PSCs), and Inness et al. (2020). There are also two other papers published for this special section, on aspects of strat/trop coupling and S2S forecasting, but I don't think these need to be cited specifically here, since they are not directly related to the topics of the current manuscript.  However, "a couple" should probably be changed to something like "several".

**Typos / Grammar / Minor Wording:**

Page 1, line 27: should be "...(on the order…"

Page 2, line 5: "allow" should be "allows" and "hamper" should be "hampers"

Page 2, line 8: "an altitude" should be "altitudes"

Page 2, line 19: "lower polar" should be "polar lower"

Page 3, line 5: "heavy" should be "large" or "strong".

Page 3, line 13: "have" should be "has".

Page 3, line 21: delete comma after "noteworthy".

Page 3, line 31: "far away" could just as easily mean "far below" as "far above"!

Page 4, line 15:  "to" should be "as for".

Page 4, line 17: delete "laid", and "data is" should be "data are".

Page 6, line 11:  "50 hPa" isn't really a height "range".

Page 6, line 15: Please say "approximate activation threshold".

Page 6, line 31:  I have no idea what you mean by "exemplarily corresponding" -- this is certainly incorrect English usage in this sentence, but I can't suggest a correction because I don't know what you mean to say.

Page 5, line 8:  This sentence is not very clear.  What does "They" refer to?  It would also be better to say "using a correction" rather than "in terms of a correction".

Page 7, line 8: Suggest adding "and references therein" to the Millán and Manney reference.

Page 11, line 7: "is showing" should be "shows".

Page 11, line 16: "is in large part reflecting" should be something like "reflects in large part" or "to a large degree reflects".

Page 13, line 29: "five weeks" should be "five-week".

Page 15, line 3:  "in" should be "on".

---

## Editor Decision (ED1)

**Editor Corrections for ACP Manuscipt No. ACP-2020-746 2nd revision**

**Dameris et al., Record low ozone values over the Arctic in boreal spring 2020**

P1, L15: Abbreviations ERA5 and ECMWF need to be introduced

P1, L18: change to …., namely the 1997 and 2011 seasons that were also showing……

P1, L23: in Northern hemisphere spring 2020 → in the Northern hemispheric spring 2020

P1, L34: (here 2016) → (as e.g. 2016)

P2, L1: can occur in polar regions → can occur in the polar regions

P2, L2-3: Sentence not clear. Does the ozone depletion hampers the transport or the chemical and dynamical processes? Sentence should be split in two or rephrased to be more clear.

P2, L8: The abbreviation SH should be introduced here.

P2, L23: in the polar spring → in polar spring

P3, L2: Introducing SH here is too late. Should already be done on P2, L8.

P3, L4: is known → was known

P3, L5: P3, L5: in NH spring → in the NH spring

P3, L6: as we will see → as will be seen

P3, L9: Here I would suggest to write that these were low, but not as low as 220 DU. Otherwise it feels a bit contradicting to what is written on P3, L30.

P3, L24: delete "such low"

P3, L26: "far removed" is also not the right expression. I would say it is even worse than far away. Please rephrase. I think just writing "far from" could be correct experession.

P3, L28: delete "it is".

P3, L29: I would rather write "season" here than "period".

P3, L32: in Section 5 and Section 6 → in Section 5 and 6

P4, L3 and 4: ERA5 and ECMWF need to be introduced here once again.

P4, L5: delete "was downloaded" and add at the end of the sentence "were used".

P4, L6: prepared → calculated

P4, L9: produced → "derived" or "calculated"

P4, l9-10: Not clear what you actually derive from ERA5 the PV or the data on isentropes (interpolation from pressure to theta levels)?

P4, L13: The abbreviation TROPOMI needs to be introduced.

P4, L17: total ozone → total ozone columns?

P4, L20: Abbreviation OMI needs to be introduced.

P4, L22: What do you mean with "first"? For the first time?

P4, L25: Also here, the instrument abbreviations that have not been introduced before need to be introduced here.

P4, L29: Sentence not clear. It may be that you need to add a comma after "for".

P5, L17: which was using → who used

P5, L26: as hinted → as can be seen from

P5, L26: further supported → better to write "in agreement with"

P5, L27: March in 2020 → March 2020

P6, L1: replace either "which is" by "that is" or just write without so that it reads…..inside the height range important for ….

P6, L7: in both occasions delete "which were"

P6, L8-9:  which is found end of→ found at the end of January (delete "which is" and add "at the" before end.

P6, L10: Add here that this is the vortex size.

P6, L10: allowing the formation → allowing for the formation

P6, L11: Our results were → Our results are

P6, L24: GTO-ECV has already been introduced. Obsolete here.

P6, L27: Dec → December

P6, L28: Also here, in all occasions Jan should be changed to January.

P7, L2: Connected is not the correct word here. It should rather read "coincides" or "accompanied".

P7, L3: created → develop

P7, L4: they → these

P7, L7: Abbreviation UK not introduced

P7, L9: Add "Arctic" before winter

P7, L10: was → were

P7, L16-17: I don't understand why you add here the web links to the data. The references to the studies by van Geffen et al. (2017) and Wohltmann et al. (2020) are sufficient.

P8, L4: periods → seasons

P8, L11: in (see e.g. Lawrence et al., 2020) → in Lawrence e.g. et al. (2020)

P9, L20 and 21: What areas? Be more precise.

P10, L1–2: Same here. Cumulative areas of what?

P10, L11: Abbreviation MLS needs to be introduced.

P11, L1-2: NH and SH has already been introduced. Therefore, here it is obsolete and can be removed.

P11, L8: we saw → we found

P13, L2: caused by reduced → caused by a reduced

P13, L28: Add "hemispheric" after Northern

P14, L2: data of → data from

P14, L11: This paragraph stands a bit alone here. Something should be added what one gets from these studies, thus a detailed overview. description or analyses of the Arctic winter 2019/2020.

P14, L19: add "stratospheric" before ozone

Figure 1: this is a very weird solution to make these areas white. This could easily be misread as missing data.

P30, L10: cycle → season

---

## Author Response (AR2)

acp-2020-746

Reply to the second review (revised version) of Ingo Wohltmann
by Dameris et al.

Thank you very much again for your very detailed remarks and the specific comments regarding our revised manuscript! We greatly acknowledge your work! Your statements and suggestions are very much appreciated. We have considered them in the second revised version of the paper.

In the following the points raised by you, the referee, are displayed in black and our responses are given in blue:

Unfortunately, I think that this manuscript needs another revision. The "Major comment" and the comments under "General" were not fully addressed. In addition, the number of specific comments (new ones and old ones) adds up to more than 6 pages. I have also added 10 pages of suggestions to improve language, grammar etc.

We appreciate the reviewer's effort in providing comments and suggestions to improve the readability of the manuscript and will address them point by point in the following.

**Title:**
I am much happier with the title now. However, while I am not a native speaker, I have the impression a native speaker would not have phrased it like this. Just "Record low ozone values over the Arctic in boreal spring 2020" sounds better to me. Or may be "assessment of"? Or "Record low […] spring 2020 compared to other winters"?

 The title has been changed accordingly.

**General:**
**"I have to admit that I had more of a problem with the wording and readability in some places. Wording was quite awkward in some places, and sometimes the text could have been less confusing and better organized, it felt a little bit rushed in places. I have the impression that asking a native speaker to go through the text would help in many places."**
I have the impression that you misunderstood my first comment in the "General" section or confused that with my comment under the section "Major comment". This comment was not about phrasing things carefully and the use of the term "ozone hole". It was about readability, conciseness, correct English language and grammar, awkward phrasing etc. Unfortunately, that means that this comment was not addressed adequately.
Maybe I did not express myself clearly enough. But in fact, your manuscript is very hard to read. I would try to improve language, grammar etc. in the manuscript in your own interest.
Even though I am not a native speaker, I can tell that there are many places of awkward writing, or not using the correct English phrases and words. In addition, the text is often very convoluted and confusing, and contains many repetitions and filler words. You probably could shorten the manuscript considerably without loss of information. As a result, you would get a more concise and readable text.
I have read the manuscript again and have compiled a long list of concrete suggestions how you could change the text (see last section **"New comments part 2: language"**). But this list is certainly not exhaustive.

Thank you very much for all your hard work on our manuscript. We very much hope that we got now all your points correctly. The manuscript has been revised according to your comments (see below our answers and the revised manuscript).

**Reply to my major comment:**
I appreciate your effort to phrase the script more carefully and I am happy that you acknowledge my

major comment in your reply. The manuscript has certainly improved.

Unfortunately, you contradict what you write in your reply as early as in the first sentence of your revised manuscript. You have rephrased "atypical ozone hole feature" just to "exceptional ozone hole-like feature". I am sorry to say that, but for the majority of the readers (and certainly for me) that will mean too much the same as before.

May be I can illustrate that with the following (tongue-in-cheek) example: Imagine you see a bike but write, "I see a car". Somebody responds: "But this is a bike!". You reply, "Ok, you are right. This is a car-like thing", because you are big fan of cars. Somebody else says, "Now, that you say it, it looks a little bit like a car, and it is a vehicle after all". I hope that makes clear where the problem is.

I will not go into detail, why I would not call it either an "ozone hole" or "ozone hole-like feature". You got very detailed comments on this, which I do not need to repeat here.

Unfortunately, this problem is not restricted to the first sentence of the paper. Further occurrences are at **page 1, line 10, page, 1, line 14, page 6, line 29, page 7, line 22, page 7, line 31, page 11, line 7, page 13, line 5 and page 15, line 2**.

I would suggest writing "ozone minimum" or, if you prefer "pronounced ozone minimum". That would be fine and still get to the point in a clear statement. If you insist on calling it an "ozone holelike feature", you unnecessarily provoke associations. As I already said in my original comments: Do not push the reader into a certain direction if you cannot back up this by facts.

Please understand that I am really not nitpicking here or trying to force my opinion on you. In my opinion, this is not something subjective, but really needs to be addressed. This was my only major comment, and I would be happy if this comment would be addressed not only reluctantly.

**From your reply: "(ii) the shape of the region of low TOC looks "ozone-hole-like""**

What do you mean by that? I think we agree that the area of the ozone minimum was only about 5% of the Antarctic ozone hole. Do you really refer to the geometrical shape here, i.e. the feature is roughly circular like the ozone hole? Does this have any relevance?

We got your major criticism and critical points (again) and revised the manuscript accordingly. Sorry for some of the misunderstandings. We hope that this version of our manuscript considers all your points adequately and that critical wording is now avoided. Furthermore, as the reviewer and another reviewer were not satisfied with some of the rephrasing in the revised manuscript (see upcoming Specific comments that are related to previous reviewer comments), we have completely deleted phrases such as "ozone hole-like" to avoid any misunderstandings.

**Specific comments (I will refer to the original page and line numbers(!), so that you can quickly find the comment I am referring to)**

**Page 1, lines 11-12**

You say this has been changed. You did not really. See major comment.

Has been changed to "show exceptionally low total ozone columns in the polar region…"

**Page 1, line 14**

You say "In addition, … sentence is added to make clear … differences in the respective area and time". In contrast to your statement in the reply, you discuss area here but not time. Did you forget to include this?

You are right. A sentence (at the end of the abstract) has been added.

**Page 1, line 16**

You say this has been changed. You did not really. See major comment again.

It has been changed now to "The record low total ozone columns were caused by a particularly stable polar vortex …". See also our reply to your major comment.

**Page 2, lines 5-10:**
This is certainly nothing I would insist on, but I have the reader in mind here. This is one of quite a few places which are either phrased quite awkwardly or where you will make the reader wonder why you write this.

Thank you! We decided to keep this paragraph as is. We do not think that the written text is misleading or inaccurate … it is kind of a motivation for the study.

**Page 3, lines 24-28:**
I think I have to make some serious comments to Farah here (only joking). But seriously, you make the reader wonder again why you write this. At least the part with the CDO tools makes the impression as if it would have been a challenge for you to calculate a simple mean. I am quite confident that this was not the case. Maybe you can keep the information that the daily mean was based on hourly values, but everything else is detail that is not needed. You can thank the authors of the CDO tools in the acknowledgments.

As this was also noted in another review, we have shifted the detailed discussion to the data availability section.

**Page 4, lines 26-28:**
The reason for my comment was that this is rather hard to read. May be I did not express this clearly enough.

We tried to improve this paragraph; we slightly changed it now.

**Page 4, line 28:**
Ok, you did change this, but not to the established terminology, as requested.

It is changed now, as requested.

**Page 5, line 2**
Certainly, nothing I would insist on, but that does not change my opinion that the figure is not needed. You can see the circular shape easily in Figure 1.

Ok. We like the figure (Figure 4) since it shows the persistence of the polar vortex; and that the wind speeds are high. It is now somehow related to the new Figure 2, which shows PV (475 K).

**Page 5, line 3**
This is still misleading and ambiguous. The radiative cooling does not mainly result from the dynamical conditions. It is there because there is no sun in the polar night. It is present in every year. It is somewhat modulated by different amounts of radiatively active gases transported to the polar regions in different years, but I do not think you have this in mind here. Temperature variations are caused by adiabatic warming (and by the triggered additional(!) radiative cooling) by the BDC. I agree that you do not need to go into detail here, but the sentence should be correct. Suggestion: "The dynamical conditions in winter 2019/2020 with low planetary wave activity result in very low temperatures…" Only a slight change, but physically correct now.

Changed.

**Figure 5 and 6: Please add…**
Why don't you add Antarctic data for Figure 7 (was Figure 6), too, as for the other figures (and as suggested by me)?

See also the comment below (related to Page 5, lines 11-12). The used proxy (cumulative area) is used to point out the differences of the three Northern winters. From our point of view, there is no necessity to add the SH also in this figure as we focus on Arctic conditions. Furthermore, we have added and discussed SH conditions in detail in the text and the other figures to avoid any possible misunderstandings. Therefore, we decided not to show respective values for the Southern polar region.  Furthermore, adding the SH in Figure 7 (old Fig. 6) would also expand the written text.

**Figure 5: Can you really learn…**
**(Either relevant information is missing or there is a problem with what is shown in Figure 5, now Figure 6)**
I think you have totally confused me here and I did not get it right in my original comment. Either, something must be wrong in the description in the figure caption or with the values shown the figure, or there is not enough information given to understand what you have done. This may carry over to the text on page 9 and 10 (new manuscript) and Table 1.

The lines in Figure 6 are indicating the minimum values in the polar cap region of each day. The dots show the minimum values in the polar cap region in the monthly mean field.  The description of data (dots) presented in Figure 6 should be consistent with the description of values presented in Table 1.

What exactly do the dots show in Figure 5 (now Figure 6)? You write "minima of monthly mean temperature". But thinking about it, this would only give one dot per month (you have the monthly means of polar cap temperature for 30 years, and then take the minimum, which is one value).
Or do you mean that you first take the monthly mean at all grid points for every year, and then look for the minimum in the monthly mean field in the longitude-latitude grid? That would be consistent with the data points in the plot, but would be a rather convoluted quantity, where I am not sure if it would make sense to look at the quantity.

Yes, the latter is the case. You get one point per month for each of the years. Furthermore, the colored dots in Figure 6 (new) are showing the same temperature values, which are given in the table. This is also in line with the temperature data presented in Figure 5 (monthly means). The quantity of minimum monthly mean temperature is helpful because it shows first that the stratospheric temperatures in winter 2019/22020 were low. Second, we use this quantity because it indicates that the winter 2019/2020 was unusually cold over the complete winter season including early spring. In addition, with the help of the grey dots, it is demonstrated that 2019/2020 is in each winter month at the lower end of observed minimum temperatures. Furthermore, to look at the minimum monthly mean temperatures is useful because it indicates the differences between the winters with respect to the conditions for the formation of PSCs. Also, short time-scale and small-scale events are not overly highlighted in the monthly mean data.

E.g., values would by definition be higher as the daily
minima, and since the minimum is not always at the same position, values would smear out.

Correct. This smearing out however, is also useful a small-scale and temporally limited events will not be noted so much.

Or is it the monthly mean of the daily minimum values north of 50 degrees for each individual year?
In this case, I have the impression that the dots are placed far above the position where I would have expected them. Take for example the blue dot for January. It seems that the daily blue values (small

dots) for January are almost all placed below the big blue dot. But this can't be possible if the big blue dot is the mean of the small blue dots.

See comments above and below.

I think that some area with the daily range of values would be the quantity that the reader would expect here and would immediately understand (this is usually shown in the plots in the literature). If you first take the minimum of the monthly mean values, this is a rather convoluted quantity. It certainly does give you some impression about the range of the values, but is not as straightforward as it could be.

We very much hope that after our revision of the text it the description of the used data is better understandable now. Thanks for raising this point. However, we think that showing the variability of the minima of monthly mean data is not some "strange" quantity. Furthermore, these values relate to Fig. 5 (new) and Table 1.

**Page 5, lines 11-12**
I do not think that you understood the problem here. I think it is mathematically not correct and therefore confusing. A sum like the sum here is always an approximation of an integral over time, and will always yield a value similar to an integral if done correctly. If you would take values every 12 h instead of every 24 h, you would get approximately twice the value by simply summing up. If you however do consider the units, you would divide by 2 in the case of 12-hourly values. That would give a similar value as before. The point is you are already implicitly assuming a value of "days" here, because you sum up values given every 24 hours. You probably do not want to depend your result on the number of values that you sum up in a given time period.

See also the comment below (Page 6, line 22): We have understood the problem (i.e. your point!). It is mathematically correct and also the unit for the cumulative area, i.e. "$10^{12}$ m$^2$". We have only calculated the daily (mean) area below 195 K at 50 hPa for each day, and finally we have added the daily values of the complete winter season (simply an addition of daily areas below 195 K). It is a simple proxy created for the comparison of the discussed three winters. The numbers for the SH are significantly larger and for instance scale of the cumulative values would be difficult to put them into relation with the "real world".

But we have expanded the text to provide a precise definition of what this quantity is supposed to show. Hence, there is no mathematical inconsistency. We agree that using 12h data would roughly double the values, but if defined correctly (as sum over the 12h values) and applied correctly this would be the case for all of the years shown, which would in turn allow for an intercomparison again as the relation of the quantities for the various years would not change.

**Figure 1: Add contours for the 220 DU contour and the vortex edge. These are things that are really hard to see in a colored contour plot. And the 220 DU contour is really central for the discussion in your paper.**
You say this is not necessary in your reply. I would not have made this comment if I would not have large problems to discern things here, and I think many people would share this view. This is not only a continuous color scale, but it also lacks contrast in the relevant range. In this context, "dark purple" is subjective at the least. And it is easy to add a line contour to the plot (say, in white). This should be feasible and easy to do with whatever graphical package you are working with. I do not think I am asking for too much work here. Same applies for the vortex edge. The new Figure 2 helps, and this shows that it should be easy to add e.g. the 36 PVU contour to Figure 1 (in a color different from the 220 DU contour). You could omit the new Figure 2 then.

We have highlighted data below 220 DU in the revised Figure 1. The area below 220 DU is now indicated in white. (To have consistent depictions, Figure 9 has been also updated.) We do not want to combine the Figures 1 and 2, to avoid overloading the figures with too much information. With respect to Figure 2, we would like to

keep it as it, because it also contains temperature contours (in red).

**Figure 6**

Sorry, but I do not see the qualitative difference to the other figures (Figures 3, 6, 8 in the new manuscript). I cannot follow your arguments. Again, I do not think I am asking for too much work here.

As said in our reply to your first review, this figure has been created only to demonstrate the differences of the discussed three winter seasons with low TOC. Added are also the values of the "warm" winter 2018/2019, which indicates that the "cold" areas (below 195 K) are very small (in this case only some view days around mid-December 2018). Our Figure 7 (new) indicates that the most obvious difference (for instance differences between the first half of December and first half of January between 2010/11 and 2019/20), which support the results of the other analyses discussed in the paper. We would like the figure as is (without showing the "range of variability") and hope that you will agree.

**Page 6, line 22**

No, they are not, see page 5, lines 11-12.

In your first review you said that "The numbers for the vortex area are a little bit unintuitive." We think that the given number are correct and also that the unit is correct. The cumulative value should be taken as a proxy to document the differences between the three NH winter seasons. We have given a clear definition in the text. As per that definition, we calculated the daily (mean) areas below 195 K, and we have summed the daily areas (total sum of all daily areas), which at the end provide clear differences between these three winters. And therefore, the correct unit is "$10^{12}$ m$^2$". We agree that using another definition as integration over time, which would yield units of area X time would be possible as well, however the units might sound strange as we talk about an area.

**Page 8, lines 11-15**

Mostly OK. Page 10, line 28 of the revised manuscript: "denitrification was much stronger than in 2011". That is not was the text says in Manney et al., 2020. My interpretation of the text is that 2011 was comparable in magnitude. Please change accordingly.

Has been changed ("much stronger" now reads "stronger"); see also below. Please also note, that we talk here about 2011 and not 2016.

**Page 9, lines 12-14**

OK in the revised manuscript. But I do not understand your reply "The statement … was very clear that the spring 1997 was the coldest …" This was not clear at all in the original manuscript. There is no obvious and straightforward way to define the "coldness" of a complete winter. You can take minimum temperatures, VPSC, North Pole temperatures etc. and this will give different results.

Ok. Maybe our reply was misleading, sorry. Thank you for your comment.

**Page 10, line 10**

OK, but I do not think that 15% is "clearly higher". It is only 15%. It would suffice to write "higher".

Deleted.

**Page 11, lines 11-14**

I certainly do not mind. It is your freedom as an author. My concern is just how the readers conceive this. This is all about readability and conciseness.

Ok, but we think that the given statement is correct and that it will not lead to wrong interpretation. Therefore, we would keep it as is.

**Page 11, lines 17-19 etc.**
You say you deleted the expression "ozone hole". You did not. See major comment.

It was changed to ozone hole-like in the revised version. But as stated in our reply to the major comment, we have now deleted the use of the phrases "ozone hole" and "ozone hole-like" entirely when referring to the NH.

**Technical corrections**

**Page 8, line 1 and 3 (now page 10, lines 12 and 14)**
You did not change all occurrences. You can also just write "(PSC type 2; see" for "(PSC type 2, ICEPSC; see".

Changed now in the complete manuscript.

**New comments (part 1: related to content, page and line numbers revised manuscript)**

**Page 3, lines 13-14:**
"However TOC below 220 DU have not been observed in these two years." This is not true. Have a look at your Figure 8. Please phrase correctly, e.g. add "for an extended time period in spring".

Changed.

**Page 5, line 19:**
"In connection with Figure 1". Sorry, I cannot follow you. The sentence would read totally correct for me if you would delete this part. This seems to be superfluous and not correct.

Deleted.

**Page 6, line 33 to page 7, line 2:**
It is not clear what you compare to, add "than in the Arctic" or similar.

Changed.

**Page 7, line 25:**
"not shown". This is not true anymore in the revised version.

"(not shown)" has been deleted.

**Page 9, lines 13-15:**
Looking at the plot, this is not really true. Daily minimum values for 2010/2011 and 2019/2020 are very similar in December and, to a lesser extent, they are also quite similar in January.

Statement has been slightly changed. It should be correct now.

**Page 10, lines 6-7:**

That is possibly misleading. That depends on what quantity you are looking at. Vortex averaged loss was not so different, at least in my study. The maximum loss made the difference.

Wording has been changed ("maximum" instead of "chemical" ozone loss).

**Page 10, lines 19-20:**

Delete. There is an almost identical sentence only a few lines before (page 9, lines 13-15). In addition, the statement may not be quite correct (see there).

Correct. Sentence has been deleted here.

**Page 10, line 25:**

"due to heterogeneous reaction". I do not think this is true. Don't you just mean "by uptake of HNO3"?

You are right! Changed.

**Page 10, line 28:**

See "specific comments, page 8, line 11-15". I do not think this statement is correct.

This statement has been slightly changed. It should be correct now. See previous comment.

**Page 11, lines 16-27:**

I found the information content of these lines to be small. This is well known from the literature and textbooks. Not that it would be wrong to mention this here, and it is all correct, but could you try and come a little bit more quickly to the point here?

This paragraph has been slightly shortened.

**Page 12, lines 3-7:**

How do you compare 30 hPa North Pole temperatures from FU Berlin to the temperature metrics used in your study? These are not the same quantities, and 2019/2020 is not included in the FU time series. E.g., North Pole temperatures could be hampered by the fact that the polar vortex is not always situated over the North Pole. Wouldn't it be better to base your statement that "2019/2020 is outstanding … since … 1950s" on a comparison of the same quantity?

Your point is correct! The discussed quantities are not the same. Therefore, we have mentioned very clearly that a range of uncertainties is given with respect to the classification of the winter 2019/2020. The given statements are vague. In this case, the results of Lawrence et al. (2020) also helped, which are based on JRA-55. This paragraph has been slightly changed (wording).

**Page 12, line 17 to Page 13, line 6:**

Again, it would help if you would come to the point here more quickly. It is totally ok to discuss this here, but this is a rather long and meandering discussion. The main topic of your paper is the 2019/2020 winter. While it is ok to compare to other winters, I do not really see the relevance of all the details given here. Where does this aim at? This seems like a repetition of facts from other sources.

These two paragraphs have been significantly shortened.

**Page 13, line 11:**
"although the dynamic features are similar". What exactly do you mean by dynamic features here? I think this statement is too subjective and should be deleted.

Deleted.

**Page 13, lines 16-18:**
This has already been discussed. Delete. In addition, I think this statement is not really correct, see comments I have already made.

We have not deleted this part; we have modified the statement, so that it is also correct.

**Page 13, lines 20-24:**
Delete. You have already discussed that in preceding section in detail. You only repeat thinks here you have already said.

We would like to keep this part in the discussion section. The sentence has been slightly changed (according to your suggestion given below).

**Page 13, lines 24-26:**
Delete sentence starting with "It is obvious…" You have stated exactly the same a few lines above (lines 16-18). Lines 16-18 were already a repetition. In addition, the statement is not correct.

Deleted.

**Page 13, line 27-30:**
Delete. You have already said this and only repeat things here.

Parts of the sentence have been deleted.

**Page 13, line 33:**
Probably you mean "in particular" and not "especially". However, that does not really fit either. You could delete "especially". However, here is a deeper problem. In fact, you are speculating here and cannot really back this up by facts. I would suggest either to cite a reference here or to replace "especially" by something like "possibly" or "probably".

Changed ("was especially caused" to "might have been caused").

**Page 14, line 1-2:**
This slipped through my attention in the first review. Again, this cannot really be backed up by facts. While it is plausible, you do not do an analysis of dynamical and chemical contributions to the observed ozone column. You need to phrase that more carefully.

We have added "might have" to keep the sentence.

**Page 14, line 7:**
Delete "only". I think I already made a comment that you can also learn something from the past here. Deleted.

**Page 14, line 8:**
Delete "more or less". Either it is cooling or not. This is not really a scientific statement.

Deleted.

**Page 29, line 4:**
As said in the specific comments, something is wrong with the "minimum of the monthly mean temperatures", or there is not enough information given.

As said already above, the lines in Figure 6 are indicating the minimum values in the polar cap region of each day. The dots are given the minimum values in the polar cap region in the monthly mean field. It should be now clear in the text and caption. See our previous comments regarding minima of monthly mean temperature data.

**New comments (part 2: language, grammar etc., page and line numbers revised manuscript)**

**Page 1, lines 23-24:**
"emphasizes the noteworthiness" does not sound like good English to me. "highlights" or "underlines" is probably better. "noteworthiness" sounds strange here. Maybe "highlights the unique evolution of…" or something similar.

Changed.

**Page 2, line 4:**
Delete "especially". You probably mean, "In particular, unusually low ozone…".

Changed.

**Page 2, line 7:**
Suggestion "…are found in the stratospheric ozone layer…".

Changed.

**Page 2, line 21, Page 13, line 24, Page 29, line 11:**
Change "ICE-PSC" to "ice PSC".

Changed all.

**Page 2, line 23:**
Replace "for instance chlorine" by "chlorine and bromine". It is only these two species.

Changed.

**Page 2, line 25:**
Would help to replace "ozone depletion begins" by "ozone is depleted by catalytic photochemical cycles" or something similar. Only a few words more, but some more relevant information.

Changed.

**Page 2, line 27-30:**
This is a very long sentence and hard to read. Maybe you could just shorten "… in response to the Montreal protocol, 1987, and its amendments…"

Slightly shortened.

**Page 2, line 31:**
"atmospheric content" seems not the right phrase for me. I suggest "atmospheric burden" or "atmospheric concentrations" etc.

Changed to "burden".

**Page 3, line 5:**
Change "heavy ozone depletion" to "severe ozone depletion".

Changed.

**Page 3, line 7:**
Delete "clearly".

Deleted.

**Page 3, line 10:**
You could delete "as we will see in the upcoming analysis".

We would like to keep this sentence as otherwise this statement seems is not backed up.

**Page 3, lines 10-13:**
The sentence is somewhat convoluted and does not read well. Suggestion: "Comparable dynamical conditions in the Northern stratosphere in spring were noted in the literature for 1997 (references…) and 2011 (references…)".

Changed.

**Page 3, lines 14-16:**
This is phrased awkwardly. Suggestion "Although the dynamical conditions in winter and spring 2019/2020 were unusual, they are in the expected range of stratospheric dynamical fluctuations in Arctic winter and early spring (e.g., Langematz et al., 2014).".

Changed ("expected" to "natural").

**Page 3, lines 28-29:**
Again, this is phrased awkwardly. Suggestion: Replace "It allows an evaluation of the current situation by the comparison with similar dynamical conditions in Arctic spring of other years" by "We compare the current winter to winters with similar dynamical conditions in Arctic spring…".

Changed.

**Page 3, line 31:**
Replace "far away from the usually observed Antarctic ozone hole" by "far removed from the conditions usually observed in the Antarctic ozone hole".

Changed.

**Page 4, line 10:**
Delete "For our investigations".

Deleted.

**Page 4, line 17:**
"The focus is laid on stratospheric zonal winds, polar temperatures and potential vorticity (PV)." I think you do not need that sentence.

Sentence has been deleted.

**Page 5, line 18:**
"turned out to be persistent" is phrased awkwardly. Suggestion: "Arctic winter and early spring 2019/2020 showed a persistent stratospheric polar vortex with strong zonal winds from midDecember until early April.".

Changed.

**Page 5, line 20-21:**
Phrased awkwardly. Suggestion: replace "representing the dynamic state of the lower stratosphere with respect to the position and strength of the polar vortex." by "and shows the position and strength of the polar vortex.".

Changed.

**Page 5, line 21:**
Change "PV-gradients" to "PV gradients".

Changed.

**Page 5, line 24:**
Phrased awkwardly. Suggestion: "Figure 3 shows strong zonal mean zonal wind speeds at 60°N, 10 hPa (about 30 km altitude) in the ERA5 data (magenta line and dots in the figure), …".

Changed.

**Page 5, line 25:**
Replace "which are high with respect to" by "which are higher than".

Changed.

**Page 5, line 26-27:**
Delete "very much". Do not exaggerate.

Deleted.

**Page 5, line 27:**
Delete "the respective".

Deleted.

**Page 5, line 31 to page 6, line 3:**
I do not think that this sentence, which is very hard to read, does convey any information that is useful in the context of this paper. Please delete.

This sentence (with the explanations) has been shortened. We think that a reference is needed here.

**Page 6, line 6:**
Delete "clearly".

Deleted.

**Page 6, line 10:**
Change "lower stratosphere temperatures" to "lower stratospheric temperatures".

Changed.

**Page 6, line 11:**
The 50 hPa level is not a height range. It would be better to say, "which is inside the height range". I would also say "important for ozone depletion" and not "of vital importance for ozone depletion". Better English in my opinion.

Changed.

**Page 6, line 14-15:**
You can delete "(i.e. the Cl activation threshold at this altitude, see for instance Figure 4-1 of Chapter 4 in WMO, 2018)" now. You now have that in the introduction. That was exactly the reason why I was asking to write it in the introduction. You can streamline the text here.

Deleted.

**Page 6, line 21-22:**
That could be shorter now: "This led to conditions allowing the formation of NAT-PSCs at 50 hPa for about 3.5 months (see Figures 6 and 7)." You introduce the abbreviation in the introduction.

Changed.

**Page 6, lines 24-26:**
You can shorten this significantly now. Again, that is why I asked for this in the introduction.

A sentence has been deleted.

**Page 6, lines 31-32:**
This is phrased awkwardly. I think you should delete "exemplarily" and start with "In Figure 3 and 6, corresponding…".

This has been changed to: "In Figures 3 and 6, corresponding…"

**Page 7, line 26:**
You could delete "indeed".

Deleted.

**Page 8, lines 6-7:**
Shorter: "The maximum area with TOC below 220 DU was 0.9 million km2 (= 0.9·1012 m2) on March 12 (Figure 1).".

Changed.

**Page 8, line 7:**
You very probably mean "polar vortex area" and not "polar vortex".

Changed.

**Page 8, lines 7-8:**
Phrased awkwardly. Suggestion: "This is in the order of 4% of the polar vortex area at the 475 K isentropic surface inside the 36 PVU contour (e.g., Wohltmann et al., 2020).".

Changed.

**Page 8, line 11:**
Change to "minimum TOC is clearly higher" (singular).

Changed.

**Page 8, lines 18-19:**
"spring" is mentioned twice in the sentence. You could omit "in spring".

Changed.

**Page 8, lines 24-26:**
This is an overly complicated sentence for a simple fact. Much shorter without loss of information: "The evolution of the polar vortex at 10 hPa and the minimum temperatures at 50 hPa are commonly used to examine the dynamical state of the stratosphere (see e.g. Lawrence et al., 2020).".

Changed.

**Page 9, line 4:**
It is not clear what "see below" refers to.

Changed.

**Page 9, line 5:**
Probably a "," would help: "…are similar, reaching…".

"," added.

**Page 9, lines 6-9:**
This is a very long sentence. It would help to split it into two and to shorten it.

Changed.

**Page 9, line 10:**
Delete "the respective".

Deleted.

**Page 9, line 11-12:**
Suggestion: "The temporal evolution of the observed daily minimum temperatures at 50 hPa is shown in Figure 6.".

Changed.

**Page 9, lines 12-13:**
Phrased awkwardly. Suggestion: "The minimum temperatures were below the threshold temperature for the formation of NAT PSCs (195 K) in February and March of all three years.".

Changed.

**Page 9, line 13-15:**
Phrased awkwardly. Suggestion: "Minimum temperatures at 50 hPa in December 2019 and January 2020 were lower than the minimum temperatures in December/January 1996/1997 and December/January 2010/2011 most of the time." (but see comments above that this not really true).

The sentence has been rephrased.

**Page 9, line 16:**
What do you mean by "indicating the characteristic of"? I am clueless.

"the characteristic of" deleted.

**Page 10, line 1:**
It must be "of the two winters 1996/1997 and 2010/2011". "to" makes no sense here.

Changed.

**Page 10, line 2:**
You probably mean "in particular" and not "especially".

Changed.

**Page 10, line 3:**
Phrased awkwardly. Suggestion: "Severe chemical loss was observed in spring 2020 (Manney et al., 2020).".

Changed to "Severe chemical ozone loss was observed in spring 2011 (Manney et al., 2011)."

**Page 10, line 4:**
Change "which was mentioned in" to "according to" (or just cite the studies at the end of sentence and skip this part of the sentence).

Changed.

**Page 10, line 13:**
Change "was determined with" simply to "was".

Changed.

**Page 10, line 17 and 18:**
You could easily delete "To summarize" and "Our analyses show".

We think the introductory phrase "To summarize" helps the reader to understand why some results are repeated here. The second phrase has been deleted.

**Page 10, line 20:**
You could delete "It is worth mentioning that".

Deleted.

**Page 10, line 22:**
Delete "in" at start of line. Seems to make no sense here.

Deleted.

**Page 10, line 22:**
You could delete "Having".

Changed.

**Page 10, line 23:**
"more efficient". Compared to what? You could just delete "more".

Changed.

**Page 10, line 23:**
Had to think a moment about what you mean by "they". I would just write "PSCs".

Changed.

**Page 10, line 24:**
You probably mean "in particular" and not "especially".

Changed.

**Page 10, line 26:**
Suggestion: "enabled a period of ozone depletion that was longer than usual".

Changed.

**Page 10, line 27:**
Unnecessarily complicated sentence (and not phrased very well). Just start with "Manney et al. (2020) analyzed…".

Sentence has been shortened.

**Page 10, line 30:**
I was confused here over reading "July" and "June" for the Northern hemisphere, until I realized that you are talking about the seasonal evolution of ozone over the period of a complete year. Can you try to phrase that a little bit differently?

"temporal" has been changed to "seasonal".

**Page 10, lines 30-32:**
A very long sentence. Please split into two sentences.

Structure of the sentence has been slightly changed, i.e. split the sentence into two.

**Page 10, line 32 to page 11, line 2:**
Unnecessarily long and complicated sentence.
Suggestion: "Typical features of a strong polar vortex can be observed in February 1997 and February 2011, with low TOC values in the vortex and relatively high TOC values in the collar region of the polar vortex (not shown)".

Changed.

**Page 11, lines 3-6:**
Another very long sentence. You could delete "An important point to be mentioned is that". You could also delete "and which were then followed by a typical chemical ozone loss". What is a typical loss? Does this add any important information?

The sentence has been shortened as suggested.

**Page 11, line 7:**
"shows" and not "is showing" is probably the correct tense.

Changed.

**Page 11, line 8:**
Delete "clearly".  Deleted.

**Page 11, lines 11-14:**
Again, a very long sentence. Can you split this in two?

Structure of the sentence has been slightly changed, but it has not been split in two parts.

**Page 11, lines 28-30:**
Phrased awkwardly. Suggestion: "The winter 2019/2020 shows an extraordinary dynamical situation with a persistent strong and cold polar vortex over the complete winter season, when compared to the last four decades (the period of the ERA5 dataset)."

Changed.

**Page 11, line 31:**
You could delete "our dynamical analyses based on".

Not deleted, see comment above.

**Page 11, line 31:**
It is not clear what "*the* historical data set" refers to. Better, "An analysis of historical data was…".

Changed.

**Page 12, line 5:**
Phrased awkwardly. Suggestion: "Temperatures in January 1997 were near the climatological mean value.".

Changed.

**Page 12, lines 5-6:**
You could delete "In combination with our research results".

Deleted.

**Page 12, line 7:**
Delete "for instance".

Deleted.

**Page 12, line 10:**
Delete "their investigations of".

Changed.

**Page 12, lines 11-12:**
Delete "(i.e. in the Berlin analysis it ranked second after 2019/2020)". You stated the same a few lines before.

Changed.

**Page 12, lines 12:**
Phrased awkwardly. Change "(i.e. in the Berlin analysis it indicated as a cold winter, but not extraordinary)" to "(indicated as a moderately cold winter in the Berlin analysis)".

Changed.

**Page 12, lines 12:**
Change "turned out to be" to "were".

Changed.

**Page 12, lines 13-15:**
Phrased awkwardly. Suggestion: "The slightly different results for the record years indicate that results depend on the considered quantity" (or "considered meteorological variable").

Changed.

**Page 12, line 15:**
Delete "Further it is clear that the".

"Further" was deleted.

**Page 12, lines 15-16:**
Phrased awkwardly. Suggestion: "Nevertheless, it is obvious that the winter 2019/2020 was one of the coldest winters in the last 65 years, and that it showed an exceptionally strong and stable polar vortex.".

Changed.

**Page 12, line 17:**
Delete "We note that".

Changed.

**Page 12, lines 17-19:**
I would suggest splitting this long sentence into two.

Rephrased.

**Page 12, line 23:**
Change "(around the long-term mean)" to "(similar to the long-term mean)".

Changed.

**Page 12, line 25:**
Phrased awkwardly. Suggestion: "The Southern hemisphere spring seasons of 2002 and 2019 provide two additional examples for the importance of stratospheric dynamics in the development of low ozone columns".

Changed.

**Page 12, line 28:**
"The other example happened in September 2019." Phrased awkwardly. Just delete and continue with "In 2019, the polar vortex was…".

Changed.

**Page 12, line 29:**
Throughout the paper, you talk of "zonal mean zonal wind", but here you say "zonal mean west wind".

Changed.

**Page 12, line 30:**
Change "happened" to "was observed".

Deleted.

**Page 12, line 32:**
Change "Afterwards" to "After this event,".

Deleted.

**Page 13, line 8-9:**
Change "lower stratosphere temperatures" to "lower stratospheric temperatures".

Changed.

**Page 13, line 10:**
Delete "that". Grammatical error.

Deleted.

**Page 13, line 11-12:**
You could split this sentence into two.

Changed.

**Page 13, line 13-14:**
I would just say, "…were similar to the conditions in early spring…".

Changed.

**Page 13, line 14:**
Delete "Further". Or write at least "Furthermore".

"Further" has been deleted.

**Page 13, line 15:**
I would suggest "Minimum TOC values were below 220 DU for several days in March 2020, although…"

Changed. But we also added that 220 DU were not reached in March 1997 and 2011.

**Page 13, line 16:**
Delete "clearly".

Deleted.

**Page 13, lines 16-18:**
Delete "Our comparisons show that especially".

Deleted.

**Page 13, lines 18-19:**
Change "all the time below 195 K" to "below 195 K most of the time".

The sentence has been rephrased.

**Page 13, line 19:**
Add "," following "In this context".

Added.

**Page 13, line 20:**
Delete "Our analyses show that".

Deleted.

**Page 13, line 30:**
It is not clear what "However" does refer to. Delete.

Deleted.

**Page 14, line 3:**
Phrased awkwardly. Suggestion: "Record low stratospheric ozone values over the Arctic in 2020 are not an unequivocal result of climate change."

Changed.

**Page 14, line 4:**
Phrased awkwardly. Suggestion: "The dynamical situations in February and March of 1997, 2011 and 2020 were similar."

Changed.

**Page 14, line 4:**
Delete "Beyond that". It is not clear what you refer to and it is confusing to read.

Deleted.

**Page 14, lines 6-7:**
Awkward phrasing. "is possible" sounds like that would be a surprise, but we expect cold winters from time to time. Suggestion: "The NH winter 2019/2020 is a perfect showcase for a Northern winter with low planetary wave activity, a strong and stable vortex and low temperatures." (replaced "less" by "low", since you do not compare anything here).

Changed.

**Page 14, line 12:**
Phrased awkwardly. Suggestion "… showed that cold Arctic winters may possibly get colder in the future."

Changed.

**Page 14, line 18:**
Delete "respective".

Deleted.

**Page 14, lines 23-25:**
I think that I have already made a comment that this seems strangely out of context. I would really suggest deleting this.

The question of the possible role of the unexpected CFC-11 emissions was raised (e.g., by colleagues and journalists) with respect to the record low Arctic TOC in spring 2020. For this reason, we would like to keep this statement in the manuscript.

**Page 14, line 27:**
Why did you add "consistent" to my suggestion? That sounds strange. I would delete that. Nobody would expect an inconsistent description here.

Word deleted.

**Page 14, line 29:**
"in the vicinity or below 220 DU". Phrased awkwardly. Suggestion: "Record low TOC values of 220 DU and less were detected…"

Changed (around 220 DU or less).

**Page 14, line 29:**
It has to be "large" for "larger" and "extended" for "longer". You do not compare anything here.

Changed.

**Page 14, line 30:**
Phrased awkwardly. Suggestion: "2019/2020 is compared…" or "The situation in 2019/2020 is compared…"

Changed.

**Page 14, line 31 to Page 15, line 2:**
Awkward phrasing. Suggestion: "We have used recent meteorological data from ERA5 and recent total ozone column data of GTO-ECV (based on…) in combination with TROPOMI onboard Sentinel-5P."

Sentences has been changed.

**Page 15, lines 2-5:**
Extremely long sentence. Please split into two.

We keep it as is. The text between the hyphens should be seen separately. See next comment: the brackets have been deleted.

**Page 15, line 4:**
It seems the hyphen is not at the correct position. It should be "…mid-October) – and TOC…"

The brackets have been deleted and the sentence was slightly rephrased

**Page 15, line 5:**
"were not observed before over a period of 5 weeks". It should become clearer that you refer to past years and not to the same winter here.

Change "before" to "in previous years".

**Page 15, line 5-6:**
You could delete "The results of our study pointed out that".

Changed.

**Page 15, Line 7:**
"supporting". Is this really what you want to say? You probably want to express that ozone depletion was more severe than in other years. Maybe "leading to enhanced ozone depletion compared to other years" or similar.

Changed.

**Page 15, line 8-9:**
Shorter and not so convoluted: "The special dynamical situation in winter 2019/2020 is the cause for the significant reduction of the TOC in spring 2020 …". This removes also the ambiguity what "despite" refers to.

Changed.

**Page 15, lines 11-14:**
Very long and complicated sentence. Suggestion: "Numerous studies of the 2019/2020 winter season can be found in a special issue of Geophysical Research Letters and Journal of Geophysical Research – Atmosphere (e.g., Manney et al., 2020; Wohltmann et al., 2020, Lawrence et al., 2020; Grooß and Müller, 2020)." Note that I removed statements that are only true at the moment of your writing. Soon, nobody will care that some of the studies were not published at the time of writing.

Changed.

**Page 15, lines 18-20:**
Phrased so awkwardly that I had to read it several times to get the meaning. Suggestion: "However, in winters with a cold and stable polar vortex, a persistent region of low TOC might also develop in the Northern hemisphere in the future again."

Changed.

**Page 15, line 21:**
You probably do not mean "documented" but "considered".

Wording is changed ("watched carefully")

**Page 15, line 22:**
"enable well founded scientific explanations of special ozone features" is phrased awkwardly. Suggestion: "Continued monitoring of ozone with a suite of instruments will be key to understand the future development of Arctic ozone" or similar…
However, this rephrasing would also mean to change the following sentences a bit.

Changed.

**Page 25, lines 6-7:**
Suggestion: "The grey line is highlighting the 36 PVU contour."

Changed.

**Page 26, line 6, Page 29, line 7, Page 31, line 14:**
I would move "(attention: the respective data are shifted by six months)" to a separate sentence. Delete "attention" and write "Southern hemisphere data are shifted by six months."

Changed.

**Page 29, line 10:**
Change "broken" to "dashed".

Changed.

**Page 31, line 11:**
"Note the …." I would not write that in the figure caption without more context. I think this belongs into the main text.

Changed.

acp-2020-746

Reply to the review of the revised manuscript of Gloria L. Manney
by Dameris et al.

Thank you very much for your remarks and the specific comments regarding our revised manuscript. Your statements and suggestions are very much appreciated. We have considered most of them in the secondly revised version of the paper.

In the following the points, which you raised, are displayed in black and our responses are given in blue.

**Overview Comments:**
This revised paper is, for the most part, scientifically sound, and provides a different view using new datasets of the evolution of total ozone column (TOC) in 2019/2020; the material is thus ultimately appropriate for publication in ACP. In the revision, the authors have gone a long way towards addressing several serious concerns in the reviews of and SCs on the initial ACPD version, but they have not entirely succeeded in some cases, as detailed below. In addition, there are some changes that should be made to the figures and text that should not be difficult or consuming of time or other resources but would have great value in making the main points of the paper more clear. I recommend publication in ACP if these concerns are addressed.
I do find the overall focus (what you might call "balance") of material in this paper somewhat problematic, because two already published papers (the Lawrence et al JGR paper and the Wohltmann et al GRL paper; these are augmented by complementary information in the Manney et al 2020 GRL paper) describe the meteorological situation in the lower stratosphere (where most relevant to chemical ozone loss) and its relationship to other Arctic winter/spring seasons with extremely strong and/or cold polar vortices much more completely that does this paper. All of these papers detail the comparison with 2011; in addition, Lawrence et al compare with 1997, and all of these papers also compare with 2016, which is notable for having an extended period, primarily in Jan/Feb, with the record cold of any Arctic winter in the past approximately 70 years (e.g., Manney and Lawrence, 2016, ACP; Matthias et al, 2016, GRL), the most denitrification and dehydration on record, and chemical ozone loss as rapid as or more rapid than that in 2011 (or 2020) until the early vortex breakup in March (Manney and Lawrence, 2016, ACP; Khosrawi et al., 2017, ACP; Johansson et al., 2019, ACP). In contrast, while there is some material on TOC in the papers published so far on the 2019/2020 Arctic winter (including comparisons of OMI and ground-based data with climatology in Bernhard et al, 2020, revised and resubmitted with very minor revisions for GRL, original ESSOAr link: https://doi.org/10.1002/essoar.10504414.1, a paper focusing primarily on the associated UV anomalies; and analysis including TOC comparisons with other winters using the CAMS reanalysis in Inness et al, 2020, https://agupubs.onlinelibrary.wiley.com/doi/10.1029/2020JD033563), the current manuscript does offer a different view of this with long-term comparisons using different, and relatively new, datasets. Thus I would strongly encourage the authors to rebalance the paper so as to focus less on the description of the meteorology (which could, for the most part, be described sufficiently by citing published material that was readily available well before this paper was initially submitted) and more on the impacts of that on TOC and clearly detailing how those impacts are seen in the TOC data that they show. I do, of course, realize that, with the special issue, previously published material is a "moving target" and with a special issue (which I like to think of publications in other journals as informally contributing to), a certain amount of approximate duplication is inevitable -- so I hope it is clear that I am not asking the authors to remove all of the dynamical material (even where I do point out below in specific comments that a paper has covered the point already), just to cite the published work appropriately (which is already much improved, though not perfect, in the revised paper) and to focus less on that and more on the parts of this manuscript that are unique.

At the beginning of this reply to your review, we would like to explain again the circumstances for the preparation of our paper. Please also take notice of our recent answers to the different comments and reviews.

When we started to write this paper (beginning of April 2020) we did not have notice of the mentioned papers and the special issue. Our paper was submitted first on June 9 (acp-2020-573) This version was rejected due to a misunderstanding with respect to the used data material. After this misunderstanding was resolved, a re-submission of our manuscript was published in ACPD on July, 21 (acp-2020-746). As we were trying to retain the initial publication record (i.e. keep the version as acp-2020-573) it took quite some time between the initial submission and the re-submission (e.g. because of absences). Unfortunately, in the end we were told, that this is technically not possible and hence the re-submission has a new version record (acp-2020-746). In the meantime, we realized that there are many other papers on the way with respect to the winter 2019/2020. We have replied to the referees and to the authors of the given short comments, where we did describe our situation. In the revised version of our paper, we have discussed in detail the results of the newly published and submitted papers, in particular the mentioned studies of Manney et al. (2020), Wohltmann et al. (2020) and Lawrence et al. (2020). In addition, several other papers have been included in the revision to acknowledge the work of others in this field of research.

Thank you for raising the point that " previously published material is a 'moving target' ". This is the reason, why we would not like to change the focus of our study. As described above, the structure and focus of this paper had been chosen before other material regarding this topic was published. The study is concentrating on the dynamics of the stratosphere and its role for the reduction of the ozone layer. In combination with our ozone data sets (GTO-ECV in combination with TROPOMI), our study provides additional information. Based on the comments by Ingo Wohltmann (2 reviews), by another anonymous referee and your comments and this review, we think that in the secondly revised version a good "balance" is found with respect to the previously published papers and our investigations. We are not only giving the respective references, but also discuss the results of these paper. Our analyses provide some additional information, which should be discussed together with the previously published paper. For instance, it is also useful to demonstrate that independent analyses of the same atmospheric situation provide very similar results. On the other hand, we have reduced some passages of our manuscript, which were presented in the ACPD version (and the first revision). We hope that you can agree with our changes (see below and in the 2nd revised version – attached), although not all points and suggestions raised by you are fully implemented.

In the next section, I make comments on some points raised in the initial reviews and SCs that I don't think the current revision completely addresses (where I don't make comments, I either think the authors' responses are adequate or that I do not have anything to add that cannot be better evaluated by the authors of the initial reviews). Following that are some more specific comments based on the revised manuscript.

**Comments Related to Author Responses To Initial Reviews/SCs:**

All of the reviews and SCs questioned the suitability of using the term "Arctic ozone hole". While the revised manuscript is improved in this respect, in particular in presenting the TOC in the Arctic in the context of that in the SH, which helps greatly in conveying the ways in which the TOC values/patterns in the Arctic in 2020 did and did not resemble those in the Antarctic. However, there are several places (including in the abstract) where the authors have simply replaced "ozone hole" with "ozone hole-like pattern", which in my opinion can still be misleading and does not really address the fact that dynamics (e.g., very low temperatures in the vortex in cases where the vortex and the temperatures are very concentric) could in principle produce a pattern that looks superficially like an ozone hole even in the absence of any chemical processing. Of course, in practice what happens is that dynamical and chemical mechanisms reinforce each other when there are widespread low temperatures in the vortex. I think "ozone hole-like" (which, if you were going to use it should be "ozone-hole-like" since all together it is one compound adjective) really does not much change how the reader sees it, and thus the term should be avoided, especially in the abstract.

In the revised manuscript, we have changed or deleted all critical text passages (i.e. with respect to "ozone hole" or "ozone hole-like"). We got your point and the critical phrases have been eliminated. No misleading information or statement should be given now.

Regarding the focus on 10 hPa winds (now Figures 3 and 4), in relation to the comment in my SC about their lack of relevance to this paper, and also Ingo's comment that Figure 3 (now Figure 4) was redundant, I do not think the authors' response is adequate. They note that Lawrence et al (2020) also show zonal mean winds; however, Lawrence et al are giving an overview of the polar vortex throughout the stratosphere and its relationship to numerous other phenomena, and are using zonal mean winds in relation to common definitions of "strong vortex" and "weak vortex" events that are used for examining stratosphere / troposphere dynamical coupling. In contrast to this, the focus of this paper is on TOC and the chemical and dynamical processes in the lower stratosphere (e.g., near 50hPa) that lead to low TOC via chemical ozone depletion. In the section in Lawrence et al that focuses on lower stratospheric polar processing and ozone loss, they (because 10hPa winds in any view, as well as zonal mean winds at any level, give little information relevant to that) use diagnostics that are vortex-centered and that are at levels that are in the altitude region where these processes take place. Figures 3 and 4 (new numbering) do not add any information that is relevant to the focus of this paper, but simply serve as a distraction or misdirection. I think they should be deleted. If the authors believe it is necessary to include interannual comparisons of diagnostics of vortex strength, those should be diagnostics that capture vortex strength at the levels where the vortex strength is relevant to the evolution of TOC (e.g., centered near 50 hPa if on isobaric levels, near 450--550K if on isentropic levels). In fact the authors' response to Ingo's comment (that what is now Figure 4 illustrates the nearly circular shape of the vortex) is true only for the middle stratospheric vortex, since the shape of the vortex commonly varies greatly with height and how it varies is different every year -- thus this tells us little, if anything, about the shape of the vortex in the lower stratosphere. (Per my overview comment above, such diagnostics have been more thoroughly covered by Lawrence et al, and thus referring to that paper for this material may be sufficient to make the points about the vortex strength in relation to other years that are important for this paper.)

As mentioned in the manuscript, the zonal wind at 10 hPa is a common measure for the strength of the polar vortex. Of course, the shape (and strength) of the polar vortex can be height dependent. This was mentioned in the revised manuscript. It was also stated that with respect to 2019/2020 lower levels (altitudes) are showing qualitatively similar structures (i.e. a stable vortex). [See the added figures below.] You are right, the focus should be laid on the altitude range of 20 km (50 hPa), the height range of importance for ozone changes. Therefore, we have added Figure 2 (new) in our paper, which shows the PV field at 475 K. We also checked the PV field at 530 K, and it looks qualitatively similar (not shown). Figures 3 and 4 (new) are both containing information, which is quite relevant for the discussion of stratospheric dynamics and its variations. And, in this connection, we are discussing the results of Lawrence et al. (2020), to provide a consistent picture. We would like to keep those two figures in the paper. With respect to the discussion of the TOC changes, we are focusing on the minimum temperature in the polar cap at 50 hPa (Figure 6) and the potential area for PSC at 50 hPa (Figure 7), which is the altitude of importance for TOC changes.

[Figure]

**Same as Figure 3 in the manuscript, but for 30 hPa (left) and 50 hPa (right). This work contains modified Copernicus Climate Change Service information (Hersbach et al., 2018; 2019a).**

With regard to the authors' response to Ingo's comment re Page 4, lines 26--28 (regarding wave activity) in the original manuscript, a better response to this would be to make this point by referring to Lawrence et al (2020), who show and discuss wave activity / fluxes / propagation / reflection and its variations throughout the winter in more detail (note that the GSFC people who produce the information on the website currently mentioned in this manuscript are co-authors on the Lawrence et al. paper, meaning that discussion has been vetted by them and is consistent with what they post on their website).

At this place of the manuscript, we have added now also the paper by Lawrence et al. (2020) and changed the text a bit.

I agree with Ingo regarding Figure 5; it is common to show the years that are not highlighted as a grey envelope (with an indication of the range and standard deviation) because it conveys more complete information about the interannual variability of the daily values. Diagnostics based on the monthly mean values do not do that. The authors have used a format like that commonly used in Figure 8 (new numbering), and this would be a more informative way to show the comparisons with years that are not highlighted in other figures.

We have commented this also in the reply to Ingo. You get one point per months (i.e. the minimum in the monthly mean field) and the colored dots in Figure 6 (new) are showing the same temperature values, which are given in the table. It is also in line with the temperature data presented in Figure 5 (monthly mean field over the polar cap). The quantity of minimum monthly mean temperature is helpful because it shows firstly that the stratospheric temperatures in winter 2019/22020 were low. Secondly, we use this quantity because it indicates that the winter 2019/2020 was unusually cold over the complete winter season including early spring. In addition, with the help of the grey dots, it is demonstrated that 2019/2020 is in each winter month at the lower end of observed minimum temperatures. This is another kind of presenting the differences (the range of variability). Finally, showing the minima of the monthly means has the benefit, that small-scale or temporally limited events do not impact this analysis too much … We think that the description of the used data should be correct now.

In relation to my concern about showing the lower stratospheric vortex structure, and Ingo's request to add a 220DU contour and vortex edge contour to the plots in Figure 1, I think the new Figure 2 does help with the vortex definition, but also think Figures 1 and 2 should be combined -- the PV and column ozone maps could be shown side-by-side, if desired, but whether or not

the PV maps are shown, the 220DU contour (the color scale is not appropriate for the reader to be able to distinguish a particular color as the authors suggest) and a vortex edge contour (e.g., an appropriate PV value) should be overlaid on the TOC maps. It would also be helpful to add a 50-hPa 195K temperature contour to illustrate the relationship of the vortex to the cold region (as I recall, it was particularly concentric in this past winter, which is relevant to TOC morphology, especially before much chemical loss has occurred).

We have highlighted TOC data below 220 DU in the revised Figure 1. The area below 220 DU is now indicated in white. (To have consistent depictions, Figure 9 has been also updated.) We do not want to combine the Figures 1 and 2, to avoid overloading the figures with too much information. With respect to Figure 2, we would like to keep it as it, because it also contains temperature contours (in red).

I agree with Ingo that Figure 6 should show the comparison with the other years. And, per my comments above, think it would be much more useful if the other years were represented in a manner similar to that in Figure 8 (new numbering).

We have commented this also in the reply to Ingo. As said in our reply to the first review of Ingo, this figure (now Figure 7) has been created only to demonstrate the differences of the discussed three Arctic winter seasons. Added are also the values of the "warm" winter 2018/2019, which indicates that the "cold" areas (below 195 K) are very small (in this case only some few days around mid-December 2018). Our Figure 7 (new) indicates that the most obvious differences (for instance differences between the first half of December and first half of January between 2010/11 and 2019/20), which support the results also of the other analyses discussed in the paper. Further, the focus of this manuscript lies on Arctic conditions. Nevertheless, as the reviewers requested, we have added and discussed SH values at many instances in the manuscript to avoid any misunderstanding regarding the relation of ozone holes in the SH to low Arctic ozone values e.g. in 2020. We are thankful that the reviewers have raised this issue of relating SH to NH values as it helped to improve the manuscript. However, we think that this issue has been dealt with sufficiently and raising it here again is unnecessary and would make the manuscript longer.

The cumulative value should be taken as a proxy to document the differences between the four NH winter seasons... We would like to keep the figure as is (without showing the "range of variability" and respective values for the Antarctic) and hope that you will agree.

Regarding the authors' response to the question by referee #2 about P5L3 (re radiative cooling and dynamical processes), the authors' response is ambiguous and could be misleading. In absence of dynamical heat fluxes, lower temperatures lead to less radiative cooling because they are closer to radiative equilibrium. So the question here is really the balance of that with the reduction in warming because of reduced planetary wave activity (i.e., dynamical heat fluxes). The wording of the statement in the revised paper is likewise ambiguous and should be modified to clarify this point.

This statement has been modified in the 2$^{nd}$ revised paper version.

Regarding my comment about page 9, lines 4--8 (relation of cold and strong vortices), I still believe this is misleading and should be modified. Yes, you can have these conditions. But you can also have weak vortex / strong mixing / substantial ozone loss, as was the case in 2004/2005.... And in 1997, even after the temperatures became unusually low, the vortex was never remarkably strong (and was remarkably weak -- but only in the lower stratosphere -- earlier in the winter) (Manney et al 2011, Nature; Lawrence et al 2020).

To bring up here in addition the situation of 2004/2005 would make the things more complicated. Therefore, we decided to shorten this part of the text. The general statement in the beginning (starting with line 4) has been

deleted. Please have a look at the 2nd revised manuscript.

**Other Comments / Questions on Revised Manuscript (in order of appearance in paper, not importance):**

Page 1, line 21, and abstract in general: The point about the different mechanisms for the low TOC in 1997 vs the other two years compared (that is, the much smaller chemical loss in 1997) should be made in the abstract.

From our point of view such specific comments in the abstract about 1997 would not be helpful. Since we are focusing on the winter season 2019/2020, the abstract should highlight the results, which are related to the exceptional low TOC values in spring 2020.

Page 1, line 26--27: "larger" than what? Presumably than in other Arctic winters in the first usage and in the Antarctic than in the Arctic in the latter -- but since the same wording is used in two different ways, you need to be explicit about what you are comparing to in each case.

We said "larger" to make clear that it is not a single point, but that low TOC values are found in a considerable area. Therefore, we changed it to "considerable".

Page 2, lines 1-2: It seems odd to me to make a general statement like this and give only a reference that discusses three instruments measuring column ozone. What about the numerous instruments with vertically-resolved measurements of multiple species (which are also critical to fully monitoring ozone loss/recovery and the processes involved)?

Therefore, we said "for example". Since we are focusing in this paper on TOC (derived from GTO-ECV and TROPOMI), we are only referring to the paper of Loyola et al. Certainly, we could mention also other papers highlighting other measurements or measurement techniques, in particular vertically-resolved observations. But from our point of view they would be not directly related to the presented results …

Page 2, lines 16--18: There are also direct effects of lower temperatures, and a relationship to higher, colder tropopauses, that work in the same direction (see SI in Manney et al, 2011, Nature, and references therein, in addition to references you already give later on tropopause heights).

We added a short sentence about the possible role of tropopause changes and refer to Manney et al. (2011).

Page 2, lines 20--21: This should be reworded to make clear that the threshold temperatures are approximate values that depend on HNO3 and H2O concentrations, and that there are several other types of particles (e.g., STS, etc) that form at temperatures similar to those of the NAT particles.

The required information regarding $HNO_3$ and $H_2O$ have been added.

Page 2, lines 21--22: This sentence (contrasting NH and SH) should be moved to the end of the paragraph, after the description of the chemistry, so that it doesn't interrupt the description of the steps leading to ozone loss.

Changed.

Page 3, lines 6--8: At this point in the paper, no evidence has been presented as to whether

this is due to chemical ozone loss. Therefore, it is premature to make this statement assuming it is related to chlorine-catalyzed chemistry.

We have tried to adjust the corresponding text.

Page 3, line 9: "stable" is not an appropriate word here, as it has a specific formal meaning in relation to the dynamical stability (e.g., barotropic or baroclinic instability) of the flow; "quiescent" or "undisturbed" would be appropriate terms.

Changed from "stable" to "undisturbed".

Page 3, lines 16--19: The papers cited here are all, with the exception of Tegtmeier et al, primarily chemistry papers, that is, they discuss the links of particular dynamical conditions to chemical loss. It would be worth citing some of the papers that discuss direct dynamical mechanisms in addition to those focused on in Tegtmeier et al (see, e.g., references in Manney et al, 2011, Nature, SI).

We have added the reference of Petzoldt (1999).

Page 3, lines 22--24: Instead of this detail / URL, and in addition to Wohltmann et al, please cite Bernhard et al (2020), submitted to GRL; this paper details column ozone anomalies in 2020 from OMI and from ground-based measurements and the corresponding UV anomalies. (Since this paper details TOC anomalies in different datasets than the ones used here, there are probably a few other places in this paper it could be cited and the consistency of their results with this paper mentioned. The same is true for comparison of TOC results with those in Inness et al. (2020).)

During the phase of the record low TOC (late February until mid-April), we looked at this URL daily for comparison with the TROPOMI data. Therefore, we mention the website here. It provides a complete overview. And a corresponding reference (van Geffen et al., 2017) is given. Of course, we could cite other papers, but I think we should not cite papers, which are still not accepted. At this place, we could live with the reference of Ingo's paper (in addition to the "URL").

Page 3, line 27, and page 4, lines 2--4: As noted above, a comprehensive (much more so than in this paper) description of the dynamical situation in 2019/2020 winter (also compared with 1996/1997, 2010/2011, and 2015/2016) is already published in Lawrence et al (2020).

Ok, therefore we are referring to the Lawrence et al. (2020) several times including a discussion of the results presented in his paper. At the end of the Introduction we are saying what we have done and what will be presented in this paper.

Page 4, lines 13--14: From "using the CDO" to the end of the sentence should be deleted, or, if you feel it is very important to give this detail, moved to the "Data Availability" section.

Deleted as suggested and this part of the sentence has been moved to the "data availability" section.

Page 4, line 25: Using "less than" and "up to" with signed values is a bit imprecise, technically it should say, for example, "less than +1% or more than -1%". It would be best to rephrase this so you talk about the magnitude of the bias and standard deviation (which isn't a signed value to begin with) rather than stating a signed value. I also fail to see why you need to give a range when it is prefaced by "up to" -- just say "up to 2.5%".

Slightly changed, as suggested.

Page 6, lines 17--18: The results of Lawrence et al and Wohltmann et al are more comprehensive than those shown here, so it might be sufficient to replace the minimum temperature plot (Figure 6) by a brief description of their results with the citations.

As said, a first version of Figure 6 was prepared in April 2020 without the knowledge of the other studies and figure therein. Among other, now this figure contains other (SH) information. We suggest to show this figure as is.

Page 6, line 25: Dameris 2010 is a rather obscure reference to cite for what is textbook material. In addition to Solomon 1999 (or instead of in this case), I would suggest Chapter 7 of the 2000 textbook "Chemistry and Physics of Stratospheric Ozone" by Andrew Dessler.

Dameris (2010) has been deleted. The citation of Solomon (1999) should be sufficient.

Page 6, line 33 to page 7, line 2: Should cite Wargan et al (2020) here.

The reference of Wargan et al. (2020) has been added here.

Page 7, lines 13--15: Manney et al 2011, Nature, also show the impact of tropopause height variations on column ozone, comparing 1997 to 2011.

The reference of Manney et al. (2011) has been included here.

Page 7, line 19: The statement "...the polar vortex existed already in late November and early December 2019" should be compared / contrasted to the other years considered here (this could be done very briefly by citing Lawrence et al 2020, who contrast the early development of the vortex in fall 2019 with other years.

Lawrence et al. (2020) has been added here.

Page 7, lines 22--23: This is a good example of a place where it is particularly inappropriate to say "an ozone hole-like pattern". In January, there has been little chemical ozone loss (almost none in most years) so the pattern of low ozone inside the vortex is primarily related directly (dynamically) to the low temperatures and concentricity of the cold region with the vortex. Even in July (+6mo) in the SH, the "ozone hole-like pattern" is mostly due to dynamical effects of low temperatures -- it is generally mid-July before the chemical loss signature overwhelms the dynamical ones. It is not appropriate to call every large low ozone region within the polar vortex an "ozone hole-like pattern".

As said above, such phrases have been eliminated all in the manuscript. The text has been changed accordingly.

Page 7, lines 24--25: As discussed above, the 10hPa winds provide no information about the strength, size, or shape of the lower stratospheric vortex. In addition, rather than saying "(not shown)" you could cite Lawrence et al (2020) for strong PV gradients.

In this context, Lawrence et al. (2020) was cited already (see beginning of next sentence).

Page 7, lines 27--30: This could be replaced by citing Wohltmann et al (2020) and Bernhard et al (2020).

The reference of Wohltmann et al. (2020) is already included here.

Page 7, lines 31--32: This ("strong horizontal gradient in the vicinity of the polar jet with strongest zonal winds") is not shown in any of your figures.

Text has slightly changed and "not shown" has been added.

Page 8, lines 22--26: Other dynamical effects that vary interannually (direct effects of low T, tropopause variations) could also be mentioned here, with appropriate references as already suggested above.

Respective literature has been added.

Page 8, line 27 through page 9, line 9: This paragraph is again discussing middle-stratospheric fields as if they were (1) relevant to the lower stratosphere and (2) had the same relationship to the conditions in the lower stratosphere in each year. Neither of these is true.

The aim of the figure is to show dynamical differences of the stratosphere with respect to the three winter seasons. Although we are only looking at 10 hPa, the results give a consistent picture in comparison with the results presented in other related studies, for instance Lawrence et al. (2020) and Lee and Butler (2020). Therefore, both papers have been cited. As said already, we also looked at other lower altitudes (i.e. 30 and 50 hPa) and there the zonal wind data showed qualitative similar results with respect to the temporal evolution of the mean zonal wind at 60°N. In addition, we have added the new Figure 2, showing the PV at 475K, representing the dynamics of the lower stratosphere in spring 2020. In combination with the results of Figure 6 (minimum temperature in the polar cap region at 50 hPa) we think that the overall message should be (more or less) clear. Our results (and conclusions) do not show (present) a different (or wrong) picture, in comparison with the results presented in the other published studies.

Page 10, lines 6--7: This is not true. Manney et al and Wohltmann et al found that ozone loss was very similar in the two years. Ozone values were lower in 2020 because chemical loss started early and possibly because of less replenishment by descent and/or less mixing.

Correct! You are right! We changed it and another sentence has been added.

Page 10, lines 8--29: The first paragraph here is an example where examining V_psc (or V_psc / V_vort) would provide a more complete picture. Both Lawrence et al (2020) and Wohltmann et al (2020) do this. These paragraphs could be condensed in light of that published information.

We have slightly reduced these two paragraphs. The mentioned papers have been cited many times, also at other place. The discussion of "our proxy" (T-area below 195K at 50 hPa) provide another view of the differences between these three winter seasons. The given numbers help to distinguish between the years of interest. The results are in line with the other investigations. And here we would give an interim conclusion, before the discussion of results start. We would like to keep it …

Page 11, lines 5--6: Please clarify what you mean by "typical" here. The ozone loss in 2011 was not typical, rather it was "unprecedented".

This was referring to 1997. However, the last part of the sentence has been deleted.

Page 11, lines 21--22: This is too oversimplified (see previous comment on radiative heating vs dynamical heat fluxes).

We have slightly changed this paragraph. The formulation is now more vague pointing to possible conditions and relationships.

Page 11, line 23 through page 12, line 16: Could be condensed, since this information content is already in published papers.

Again, we have slightly reduced the text. Nevertheless, as you said (previously published material is a 'moving target'), it is difficult for us to reduce our statements in a way, that it looks like that "everything" was already published before. The outline of our paper was created at a time, when we did not know the scientific content of the other papers.

Page 12, line 17 through page 13, line 6: These two paragraphs seem tangential to the focus of the paper, and, since they are entirely discussing results shown in already published papers without making any cogent about the relevance to this paper, seem more of a distraction than anything else.

The reason for having these two paragraphs is that one of our focusses is the role of stratospheric dynamics for the causing situation with reduced TOC in polar regions. Here we would like to point out that the dynamical behavior in both hemispheres yield to (more or less) equal circumstances. From our point of view, it makes sense to put together the available information (in particular based on recent years) and discuss it in the context (see the figures, which contain the data of other years of relevance). Nevertheless, we have again tried to shorten this part of the manuscript.

Page 13, lines 7--12: This has already been discussed in Wohltmann et al (2020) and thus could be condensed or removed.

Same reply again. We think that our Figure 6 contains an interesting comparison showing not only NH conditions, but also those of the SH. A direct comparison of NH and SH conditions helps to get the main message (i.e. the role of stratospheric dynamics). We would like to keep this paragraph …

Page 13, lines 13--26: This has already been discussed in Lawrence et al (2020) and thus could be condensed or removed.

Same reply again. Yes, Lawrence and colleagues discussed it already (in parts) … and therefore Lawrence et al. (2020) has been cited (again) in this context.

Page 13, line 30: "However" is not appropriate here -- the "extended phase of active stratospheric chlorine" leads to the "substantial ozone depletion", whereas with "However" you are saying that the latter is in contrast to the former. (Note also that neither of these is a result of this paper, though both are shown by Manney et al, 2020, and the latter by Wohltmann et al, 2020.)

The word "however" has been deleted. We did not include the references before as we thought the reasoning is typical textbook knowledge and we made vague statements ("were pointing to"). Nevertheless, we added the references as suggested and rephrased the whole part.

Page 14, line 3: No one has suggested that it was demonstrably due to climate change. Wohltmann et al (2020; and to a lesser degree Manney et al, 2020) have also already discussed this.

Ok, we also would like to raise this point. Such a question is lying on the hand and therefore is of general interest. And therefore, we have added the following paragraph, putting together the conflicting results and interpretations.

Page 14, lines 23--25: This statement does not appear to be related to anything else in the paper and seems completely out of place. It also doesn't follow from anything shown in this paper. I suggest deleting it.

The question of the possible role of the unexpected CFC-11 emissions was raised (e.g., by colleagues and journalists) with respect to the record low Arctic TOC in spring 2020. For this reason, we would like to keep this statement in the manuscript to make clear that the low ozone values in spring 2020 were not affected by enhanced CFC-11 emissions.

Page 15, line 4: It isn't clear what "and TOC values below 220 DU are seen for up to about four months" is in relation to here. Is this for the Antarctic? For the Arctic in extreme winters?

The text has been revised. It should be much clearer now.

Page 15, lines 11--14: Add Bernhard et al (2020), DeLand et al (2020; https://agupubs.onlinelibrary.wiley.com/doi/abs/10.1029/2020JD033271; this discusses OMPs PSC measurements in 2020 and compares them to Antarctic PSCs), and Inness et al. (2020). There are also two other papers published for this special section, on aspects of strat/trop coupling and S2S forecasting, but I don't think these need to be cited specifically here, since they are not directly related to the topics of the current manuscript. However, "a couple" should probably be changed to something like "several".

This paragraph has been changed / shortened. We would like to avoid citations of papers, which have not been discussed in our paper. We now are saying that "numerous papers" can be found in this special issue.

**Typos / Grammar / Minor Wording:**
Page 1, line 27: should be "...(on the order…"   Changed!
Page 2, line 5: "allow" should be "allows" and "hamper" should be "hampers"   Changed!
Page 2, line 8: "an altitude" should be "altitudes"   Changed!
Page 2, line 19: "lower polar" should be "polar lower"   Changed!
Page 3, line 5: "heavy" should be "large" or "strong".   Changed to "severe"!
Page 3, line 13: "have" should be "has".   Changed!
Page 3, line 21: delete comma after "noteworthy".   Changed!
Page 3, line 31: "far away" could just as easily mean "far below" as "far above"!   Changed!
Page 4, line 15: "to" should be "as for".   Changed!
Page 4, line 17: delete "laid", and "data is" should be "data are".   The first sentences in line 17 (page 4) has been deleted; "data is" has been changed to "data are!
Page 6, line 11: "50 hPa" isn't really a height "range".   Wording has been changed!
Page 6, line 15: Please say "approximate activation threshold".   This part of the sentence has been deleted.
Page 6, line 31: I have no idea what you mean by "exemplarily corresponding" -- this is certainly incorrect English usage in this sentence, but I can't suggest a correction because I don't know what you mean to say.   We have deleted the word "exemplarily". Now it should be clear.
Page 5, line 8: This sentence is not very clear. What does "They" refer to? It would also be

better to say "using a correction" rather than "in terms of a correction".   Both points have been changed!

Page 7, line 8: Suggest adding "and references therein" to the Millán and Manney reference.  Added!

Page 11, line 7: "is showing" should be "shows".   Changed!

Page 11, line 16: "is in large part reflecting" should be something like "reflects in large part" or "to a large degree reflects".   Changed!

Page 13, line 29: "five weeks" should be "five-week".   Changed!

Page 15, line 3: "in" should be "on".   Changed!

Thank you very much for your support!